# LARGE LANGUAGE MODELS AS MARKOV CHAINS

## ABSTRACT

Large language models (LLMs) have proven to be remarkably efficient, both across a wide range of natural language processing tasks and well beyond them. However, a comprehensive theoretical analysis of the origins of their impressive performance remains elusive. In this paper, we approach this challenging task by drawing an equivalence between generic autoregressive language models with vocabulary of size $T$ and context window of size $K$ and Markov chains defined on a finite state space of size $\mathcal{O}(T^K)$. We derive several surprising findings related to the existence of a stationary distribution of Markov chains that capture the inference power of LLMs, their speed of convergence to it, and the influence of the temperature on the latter. We then prove pre-training and in-context generalization bounds and show how the drawn equivalence allows us to enrich their interpretation. Finally, we illustrate our theoretical guarantees with experiments on several recent LLMs to highlight how they capture the behavior observed in practice.

## 1 INTRODUCTION

The fields of machine learning and artificial intelligence have recently seen significant progress with the introduction of large language models (LLMs) (Brown et al., 2020; Touvron et al., 2023a), built on the transformer architecture (Vaswani et al., 2017). These models, trained on vast amounts of data, have been applied in many natural language processing tasks, including machine translation (Brown et al., 2020), text generation, question answering (Roberts et al., 2020), and sentiment analysis (Zhang et al., 2023a). Although successful in practice, the origins of the impressive performance of LLMs remain elusive, as there is no widely accepted agreement in the scientific community on how they achieve remarkable reasoning capabilities that go far beyond their training data (Brown et al., 2020).

This work takes a step towards bridging the knowledge gap mentioned above by providing an *explicit* characterization of the LLM's inference capabilities. For this, we adopt an intuitive, yet overlooked, approach that interprets LLMs as Markov chains operating on a finite state space of sequences and tokens (see Fig. 1). A key insight is that despite the seeming infinity of LLMs generating capacity, they have a limited vocabulary and context window making all their possible input and output sequences enumerable. We show that despite the prohibitively large size of the latter set, it exhibits a structure that makes it amenable to theoretical analysis. We further generalize recent theoretical advances on the generalization of LLMs and leverage our proposed point of view to provide a more insightful interpretation of them.

**Markov chains and Large Language Models.** While none of the prior works considered the equivalence between LLMs and Markov chains presented in this work[1], some used the Markovian generative process to better understand the intrinsic capabilities of the transformer architecture. Makkuva et al. (2024) assume that the input data is generated by an unknown high-order Markov chain to analyze the learning dynamics of the self-attention mechanism in a single-layer single-head transformer. Similarly, Edelman et al. (2024) study in-context learning of a transformer model trained on samples drawn from a bi-gram Markov chain. Ildiz et al. (2024) establish an equivalence between context-conditioned Markov chains and the self-attention mechanism in transformers to show that self-attention weights can be learned, under certain conditions, by prompting the model. In contrast to this prior work, we seek to model any transformer-based LLM as a Markov chain. Hence, our

---

[1]We note, however, a recent blog post (Nardo, 2023) discussing a similar idea on a high level without analyzing it at the level of details of this work.

Figure 1: **LLM as a Markov chain**. A large language model with vocabulary size $T$ and context window $K$ is equivalent to a Markov chain with a sparse and block-structured transition matrix of size $\sum_{i \leq K} T^i \sim \mathcal{O}(T^K)$. The latter captures all possible outputs of a given LLM for all possible input sequences allowed by its vocabulary and context window.

analysis provides insights into LLMs beyond understanding the self-attention mechanism in simplified transformers.

**In-Context Learning (ICL).** ICL is the ability of LLMs to adapt their predictions during inference by leveraging examples or prompts directly without updating their parameters. Xie et al. (2022) provide a theoretical guarantee for ICL by showing its equivalence to implicit Bayesian inference, while Jeon et al. (2024) study ICL in the Bayesian setup by adopting a point of view that relates it to meta-learning from non-independent and identically distributed (i.i.d.) data. Their guarantees introduce a desirable dependence on the length of the input sequence and the number of sequences seen, yet their data generative process assumes that each sequence is outputted by a transformer model drawn from a prior distribution. A follow-up work by Wies et al. (2024) generalizes this analysis within a broader Probably Approximately Correct (PAC) framework and sheds light on the few-shot nature of ICL. Li et al. (2023) studied the generalization error of trained ICL transformers from a stability viewpoint and provided generalization guarantees for temporally dependent prompts that can be seen as Markov chains of different orders. The closest work on ICL to our analysis is (Zhang et al., 2023b) upon which we improve by relaxing many assumptions and, most importantly, by providing stronger interpretations and experimental observations.

**Generalization bounds for LLMs.** Deriving generalization bounds for neural networks is inherently difficult due to the complexity of the operations performed by the model. This can require expressing it as a continuous process (Marion, 2023). For LLMs, an avenue to obtain such bounds is to rely on the PAC-Bayes framework. Lotfi et al. (2023; 2024) leverage compression techniques in combination with PAC-Bayes type bounds to obtain tight generalization bounds both at the document and token level. These works are connected to the existing literature on compressibility and the intrinsic dimension of neural networks (Aghajanyan et al., 2020; Yaras et al., 2024) and typically focus on fine-tuning LLMs with LoRA (Hu et al., 2022) inspired adapters. The work of Zhang et al. (2023b) considers the Bayesian framework to derive generalization bounds for pre-training where the data is assumed to be a Markov chain. Our theoretical analysis of pre-training is done *without* relying on Bayesian modeling and under realistic data generating assumption covering many common types of data used to train LLMs.

**Summary of our contributions.** Our main contributions are summarized as follows.

1) We provide an explicit characterization of LLM's inference mechanism by showing its equivalence to a finite-state Markov chain. We analyze the transition matrix of the latter and prove the existence and uniqueness of its stationary distribution. We give a rate of convergence to this distribution that depends on the vocabulary and context window sizes, and the model's temperature.

2) By leveraging concentration inequalities for *dependent* random variables, we obtain generalization bounds for LLMs in both pretraining and in-context inference. Our bounds are proved under minimal assumptions on the model and data and depend on the model's depth, dictionary, and dataset sizes, as well as the intrinsic properties of the temporally-dependent sequences it was trained on. We highlight the insights that stem from these bounds by relating them to the minimax bounds of Markov chain learning.

3) We experimentally show that the most recent LLMs dating from 2023-2024 obey the in-context scaling laws predicted by our theoretical results. One highlight is that LLMs are better Markov chains learners than the minimax optimal frequentist approach (Wolfer & Kontorovich, 2019).

We underline that, in this work, the term LLM refers to a deep transformer-based model trained on non-iid data whose inference is based on the next-token prediction principle in an autoregressive fashion. The latter implies that such a model transitions between a sequence of tokens to a sequence of tokens. Hence, in Section 3, the **Markov chain** formalization transitions between states that are **sequences of tokens** (instead of single tokens). The vast majority of existing LLMs fall into our definition suggesting that our results apply to them.

**Organization of the paper.** Section 2 provides background material on autoregressive models and Markov chains. We formalize an equivalence between these two models in Section 3 and illustrate it on a toy example. In Section 4, we derive generalization bounds for LLMs trained on *non-iid* data and prompted on Markov chains. Our results are empirically verified in Section 5.

## 2 BACKGROUND KNOWLEDGE

We recall some elementary facts about Markov chains (Paulin, 2015; Roberts & Rosenthal, 2004) and LLMs. More notations and background materials are available in Appendices A to C.

**Markov chains.** Let $\Omega$ be a discrete finite set of size $|\Omega|$. A discrete-time, time-homogeneous Markov chain $\text{MC}(\Omega, \mathbf{Q})$ defined on a state space $\Omega = \{x_i\}_{i=1}^{|\Omega|}$ with transition matrix $\mathbf{Q} \in \mathbb{R}^{|\Omega| \times |\Omega|}$ with entries $\mathbf{Q}_{ij} = \mathbf{Q}(x_i, x_j) \in [0, 1]$ is a sequence of random variables $(\mathbf{X}_1, \mathbf{X}_2, \dots)$ taking values in $\Omega$ such that for any $n \in \mathbb{N}$ and $(x_1, \dots, x_{n+1}) \in \Omega^{n+1}$, we have

$$\mathbb{P}(\mathbf{X}_{n+1} = x_{n+1} \mid \mathbf{X}_n = x_n, \dots, \mathbf{X}_1 = x_1) = \mathbb{P}(\mathbf{X}_{n+1} = x_{n+1} \mid \mathbf{X}_n = x_n) =: \mathbf{Q}(x_n, x_{n+1}).$$

A distribution $\pi$ on $\Omega$ is said to be a stationary distribution if $\mathbf{Q}\pi = \pi$. Under mild conditions on $\mathbf{Q}$, $\text{MC}(\Omega, \mathbf{Q})$ has a unique stationary distribution to which it converges, i.e., for any $x \in \Omega$, $\lim_{n \to \infty} d_{\text{TV}}(\mathbf{Q}^n(x, \cdot), \pi) = 0$, where $\mathbf{Q}^n(x, \cdot)$ denotes the probability of $\mathbf{X}_n$ conditioned on $\mathbf{X}_1 = x$ and the total variation between two distributions $\mathbb{P}$ and $\mathbb{Q}$, defined on $(\Omega, \mathcal{F})$, is

$$d_{\text{TV}}(\mathbb{P}, \mathbb{Q}) := \sup_{A \in \mathcal{F}} |\mathbb{P}(A) - \mathbb{Q}(A)|.$$

We recall that the mixing time $t_{\text{mix}}(\varepsilon)$ of a Markov chain is the minimal time needed to be $\varepsilon$-close to its stationary distribution (see Definition C.8). Intuitively, a Markov chain mixes slowly when it remains close to the initial state after a given number of steps and doesn't explore its state space. A Markov chain that exhibits a fast mixing time on the contrary quickly forgets its initial state and transitions more easily to a wider set of states.

**Large language models.** Let $\mathcal{V}$ denote a dictionary of size $T$ used to encode an arbitrary sequence into a sequence of predefined tokens belonging to $\mathcal{V}$. We assume that our model admits a maximum of $K$ tokens as input, referred to as the *context window* of the model. The domain of the autoregressive LLM is the set of all sequences consisting of elements from $\mathcal{V}$ with up to $K$ elements. We denote this by $\mathcal{V}_K^*$, which represents a restriction of Kleene closure of $\mathcal{V}$, i.e., $\mathcal{V}_K^* := \{v \in \mathcal{V}^* , |v| \leq K\}$ with $|v|$ the length of $v$. We define an LLM with trainable parameters $\mathbf{\Theta}$ as a function $f_{\mathbf{\Theta}}^{T,K} : \mathcal{V}_K^* \to \Delta(\mathcal{V})$, where $\Delta(\mathcal{V})$ is the probability simplex over $\mathcal{V}$, that given a sequence of tokens $v$ outputs a probability distribution over the whole state space indicating the likelihood for each of its elements to appear after $v$ (see Appendix B for more details). We consider a setting where the learner's objective is to approximate the probabilities of sequences over an input vocabulary given by some reference distribution $\mathbb{P}_{\mathcal{L}} : \mathcal{P}(\mathcal{V}_K^*) \to [0, 1]^2$.

## 3 LARGE LANGUAGE MODELS AS MARKOV CHAINS

We formally define a Markov chain that explicitly captures the full inference capacity of a given LLM $f_{\mathbf{\Theta}}$. We build upon a high-level idea that associates a tokenized input prompt with a state $v_i$, from

---

[2]$\mathcal{P}(\mathcal{V}_K^*)$ denotes the powerset of $\mathcal{V}_K^*$.

which we transition to a new state $v_j = [v_i, v]$ by concatenating the token $v$ predicted by an LLM to it. We then provide a theoretical characterization of this Markov chain highlighting its intriguing properties and asymptotic behaviour.

## 3.1 MARKOV CHAIN FORMALIZATION

We begin by defining the transition matrix associated with a large language model $f_{\Theta}^{T,K}$.

**Proposition 3.1.** *Any large language model $f_{\Theta}^{T,K}$ can be equivalently represented by a Markov chain $MC(\mathcal{V}_K^*, \mathbf{Q}_f)$, with a sparse transition matrix $\mathbf{Q}_f \in \mathbb{R}^{|\mathcal{V}_K^*| \times |\mathcal{V}_K^*|}$ defined as:*

$$\forall v_i, v_j \in \mathcal{V}_K^*, \ \mathbf{Q}_f(v_i, v_j) = \begin{cases} 0, & \text{if } \exists l \in \{1, \ldots, |v_i| - 1\}, \text{s.t. } (v_i)_{l+1} \neq (v_j)_l, \\ \{f_{\Theta}^{T,K}(v_i)\}_j, & \text{otherwise,} \end{cases}$$

*where $|\mathcal{V}_K^*| = T(T^K - 1)/(T - 1)$. The proportion of non-zero elements in $\mathbf{Q}_f$ is $(T - 1)/(T^K - 1)$.*

We discuss the intuition behind the definition of $\mathbf{Q}_f$ provided above and illustrate it in Figure 2 for a case of $T = 2$ and $K = 3$. For this, we first note that given an input sequence $v_i$ of size $|v_i| < K$, a transition to any state $v_j$ has a probability of 0 if $v_j \neq [v_i, v]$ for some $v \in \mathcal{V}$, i.e., if the state we transition to is not a concatenation of the input sequence with an additional token from the vocabulary (for instance, a state $\{0\}$ cannot transition to $\{1, 0\}$ in one step). Applying this reasoning for different values of $k < K$ defines green rectangular blocks of size $T^k \times T^{k+1}$ in the transition matrix portrayed in Figure 2. When one reaches the blue *square* block in the transition matrix, the input sequence reaches the maximum context window length $v_i$: the model can no longer append tokens to the input sequence and has to delete

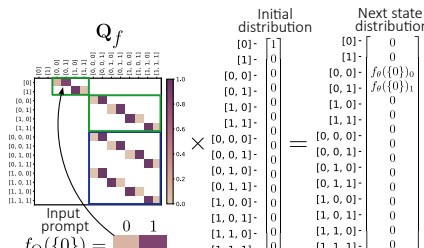

Figure 2: Illustration of Proposition 3.1 with $T = 2$ and $K = 3$.

the first token from it to proceed. This blue block is of size $T^K \times T^K$: it captures transitions between all possible sequences of the maximum admissible length. We define similarly the reference transition matrix $\mathbf{Q}^*$ of the language where the probability of transitions $\{f_{\Theta}^{T,K}(v_i)\}_j$ are replaced by ground-truth probabilities $\mathbb{P}_{\mathcal{L}}(v_j \mid v_i)$. In order to use $\mathbf{Q}_f$ as $f_{\Theta}$, it is now sufficient to define an input distribution $\delta_0$ of the Markov chain based on input prompt $v$ as a one-hot encoding vector of size $|\mathcal{V}_K^*|$ with 1 at the position of the state corresponding to $v$. Then, the transition to the next state simply writes as $\delta_1 = \mathbf{Q}_f \delta_0$. The output of $f_{\Theta}(v)$ for individual tokens in $\mathcal{V}$ would then correspond exactly to the probabilities in $\delta_1$ for states that are concatenations of $v$ with $T$ tokens from $\mathcal{V}$. This process is illustrated in Figure 2.

We now characterize this Markov chain and note that, since $\mathcal{V}_K^*$ is finite, $MC(\mathcal{V}_K^*, \mathbf{Q}_f)$ admits a stationary distribution. This stationary distribution is unique given the structure of the transition matrix $\mathbf{Q}_f$, as established in the following result.

**Proposition 3.2.** *Let $MC(\mathcal{V}_K^*, \mathbf{Q}_f)$ be a Markov chain defined in Proposition 3.1. Then $MC(\mathcal{V}_K^*, \mathbf{Q}_f)$ is an ergodic unichain and has a unique stationary distribution.*

A unichain is a chain that has at most one recurrent class plus some additional transient states. From Proposition 3.1, we note immediately that green blocks in Fig. 1 represent transient classes, meaning that applying $\mathbf{Q}_f$ to the input prompt, represented by a one-hot encoding of size $|\mathcal{V}_K^*|$, will transition to a state that corresponds to a sequence of length increased by one with an additional, most likely, token appended to it. This process is repeated if the model is called further on: we append tokens until we reach the context window limit $K$. At this point, we reach the recurrent class, represented in blue, in which the chain stays until it reaches its unique stationary distribution. We now characterize how many times one should apply $\mathbf{Q}_f$ to the input to reach the stationary distribution.

**Proposition 3.3.** *Given an ergodic finite-state unichain $MC(\mathcal{V}_K^*, \mathbf{Q}_f)$ and $e = (1, 1, \ldots, 1)^\top$, then $\lim_{n\to\infty} \mathbf{Q}_f^n = e\boldsymbol{\pi}$ where $\boldsymbol{\pi}$ is the stationary distribution of the recurrent class $\mathcal{R}$ of states, expanded by $0$'s for each transient state of the unichain. Moreover, for all $n \geq K$,*

$$|(\mathbf{Q}_f^n)_{i,j} - (e\boldsymbol{\pi})_{i,j}| \leq (1 - 2\varepsilon)^{\lfloor \frac{n}{K} \rfloor - 1},$$

*where $\varepsilon = \min_{i,j \in \mathcal{R}^2} \{(\mathbf{Q}_f^K)_{i,j}\} > 0$.*

The stationary distribution is the long-term equilibrium of the Markov chain defined by the LLM and can be interpreted as a proxy of its understanding of natural language in its token space. It is independent of the initial state (i.e., input prompt) but rather captures the absolute frequencies of occurrences of certain tokens seen during pre-training. For a well-performing model, it is hence likely to be heavy-tailed, meaning that rare states have a non-zero probability of occurring due to language's ambiguity and complexity. Proposition 3.3 shows that reaching the stationary distribution requires more generation steps for models with larger context window $K$. Additionally, convergence depends on $\varepsilon$ (that is, the smallest element of the $K^{\text{th}}$ power of the transition matrix), which is related to the ability of the chain to explore the state space after having forgotten the input prompt.

### 3.2 ILLUSTRATION ON A TOY MODEL

We illustrate the results of Section 3 on a toy model trained on a sequence of $0$s and $1$s. Here, each subsequent token is $0$ if the sum of three previous tokens is even and $0$ otherwise. Therefore, $T = 2$ and $K = 3$. We generate a sequence of $40$ digits, resulting in $37$ distinct supervised examples, and train a small "GPT-like" model (Karpathy, 2023) on it. We extract the logits from the model by prompting it with all possible combinations of $0$s and $1$s of length less than three to obtain the transition matrix $\mathbf{Q}_f \in \mathbb{R}^{14\times14}$ depicted in Fig. 3(a). The transition matrix's structure (e.g., presence of transient and recurrent classes) matches the one presented in Fig. 1. Fig. 3(b) displays the stationary distribution of the trained model obtained by raising $\mathbf{Q}_f$ to power $10^5$. We note that it has a strong bias toward seen training samples in accordance with our intuition behind the stationary distribution presented earlier. Finally, Fig. 3(c) illustrates the convergence rate of the toy model, predicted by Proposition 3.3, and compares it to models with larger dictionary size $T$ and context window $K$. In Fig. 3(c), we set $\varepsilon = 10^{-6}$ and note that this parameter reflects the ability of the LLM to explore state space.

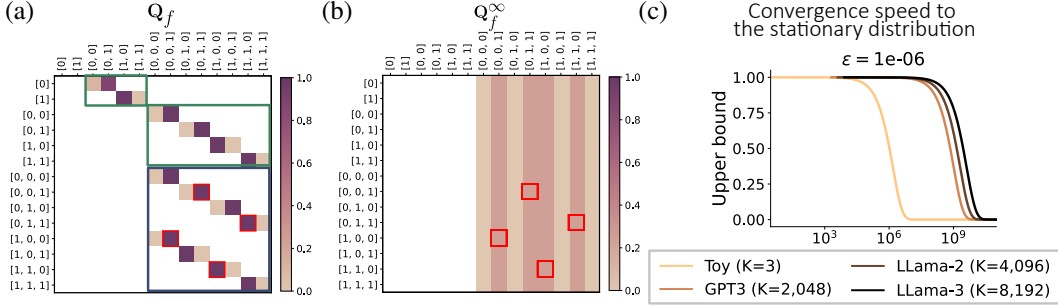

Figure 3: Markov chain with a small GPT-like model. (a) Transition matrix $\mathbf{Q}_f$ of the model where $\square$ denotes the examples from the training set. (b) Stationary distribution of the trained model assigning almost uniform probabilities to the states seen during training. (c) Convergence rate to the stationary distribution for the considered toy model along with three LLMs, highlighting the dependence on $K$. The y-axis is the upper bound in Proposition 3.3.

**Role of the temperature.** To better illustrate the role of $\varepsilon$, we now plot the transition matrix of the studied Markov chain obtained when applying different temperature scaling to the logits returned by the trained model. As the temperature is commonly linked to the ability of LLMs to transition more freely to a large set of states (Chen & Ding, 2023), we expect that lower temperatures should impact negatively the speed of the convergence to the stationary distribution. In Fig. 4(a), we show that for a low temperature (0.2), the Markov chain mixes slowly and is unable to reach its stationary

distribution (same line in the transition matrix as in Fig. 3(c)) even after $10^6$ steps. In the case of a more commonly used temperature equal to 1 (Fig. 4(b)), the model requires only 300 steps to converge. Finally, setting the model's temperature to 2 (Fig. 4(c)) makes the convergence extremely fast, reaching the stationary distribution after only 30 steps. The interplay between $\varepsilon$ and the model's temperature is displayed in Fig. 4(d), increasing the temperature leads to a drastic improvement in the convergence speed.

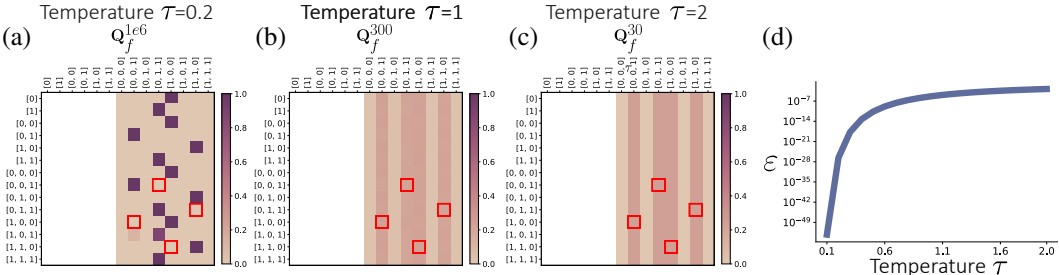

Figure 4: Dependence of $\varepsilon$ on the temperature of the model. (a) For low temperatures, $\varepsilon$ becomes too small to achieve convergence to the stationary distribution. (b)-(c) Increasing the temperature from 1 to 2 leads to a $\times 10$ faster convergence. (d) $\varepsilon$ (log-scale) increase for temperature values in $[0.1, 2]$.

## 4 GENERALIZATION BOUNDS FOR LARGE LANGUAGE MODELS

The inference of any large language model $f_{\Theta}$ can be fully captured by a Markov chain with a finite transition kernel $\mathbf{Q}_f$ defined as above. The formalization of Section 3.1 allows us to see and study the generalization of $f_{\Theta}$ as its capacity to infer correctly all the elements of $\mathbf{Q}_f$ that approximate the true reference matrix of transition probabilities $\mathbf{Q}^*$. The hardness of this task lies in achieving precise inference having observed a negligible amount of $\mathbf{Q}^*$'s elements during its pre-training. For GPT-3 (Brown et al., 2020), this represents $5 \times 10^{11}$ training tokens, which pales in comparison with the number of non-zero elements in $\mathbf{Q}_f$, given by $T^{K+1} \approx 10^{9632}$.

**Risk definition.** We denote by $X = (\mathbf{X}_1, \ldots, \mathbf{X}_N)$ the tokens in $\mathcal{V}$ that $f_{\Theta}$ observes (e.g., during pre-training or at inference time). The training sequences of tokens can be written as $\mathbf{S}_n = (\mathbf{X}_1, \ldots, \mathbf{X}_n)$ if $n \leq K$ and $\mathbf{S}_n = (\mathbf{X}_{n-K+1}, \ldots, \mathbf{X}_n)$ otherwise due to the deletion process (see Definition B.2). In particular, the $\mathbf{S}_n$ are elements of $\mathcal{V}_K^*$. For any $n \in [N]$, the true probability of next token $\mathbf{X}_{n+1}$ given a past sequence $\mathbf{S}_n$ is defined as $\mathbb{P}_{\mathcal{L}}(\cdot \mid \mathbf{S}_n) \in \Delta_T$ and the probability estimated by the model writes $\mathbb{P}_{\Theta}(\cdot \mid \mathbf{S}_n)$. We assume the existence of a constant $c_0 > 0$ such that for any $n \in [N]$ and $(x_1, \ldots, x_{n+1}) \in \Omega^{n+1}$,

$$\mathbb{P}_{\mathcal{L}}(\mathbf{X}_{n+1} = x_{n+1} \mid \mathbf{X}_n = x_n, \ldots, \mathbf{X}_1 = x_1) \geq c_0 > 0. \tag{1}$$

This is a common assumption used previously in (Hu et al., 2024; Wies et al., 2024; Xie et al., 2022; Zhang et al., 2023b). Following the Markov chain formalization introduced in Section 3.1, we define the theoretical and empirical risks for any $\Theta \in \mathcal{W}$ as[3]

$$\mathcal{R}(\Theta) := \mathbb{E}_{\mathbf{S} \sim \mathbb{P}_{\mathcal{L}}}[d_{\mathrm{TV}}(\mathbf{Q}^*(\mathbf{S}, \cdot), \mathbf{Q}_f(\mathbf{S}, \cdot))], \ \widehat{\mathcal{R}}(\Theta) := \frac{1}{N} \sum_{n=1}^{N} d_{\mathrm{TV}}(\mathbb{P}_{\mathcal{L}}(\cdot \mid \mathbf{S}_n), \mathbb{P}_{\Theta}(\cdot \mid \mathbf{S}_n)). \tag{2}$$

The generalization problem consists of bounding the difference $\mathcal{R}(\Theta) - \widehat{\mathcal{R}}(\Theta)$.

**Remark 4.1** (Choice of risk). *Our risk definition departs from usual generalization bounds in statistical learning where risks are mostly derived from empirical risk minimization (Bach, 2024; Marion, 2023; Redko et al., 2019; Vapnik, 1999). As we want to assess how well the model estimates the probability* distribution *of the next token, we rather follow (Hu et al., 2024; Zhang et al., 2023b) and the learning and identity testing of Markov chains literature (Wolfer & Kontorovich, 2019; 2023) and use the total variation distance.*

---

[3]$\mathcal{R}(\Theta) = \mathbb{E}[\widehat{\mathcal{R}}(\Theta)] = \mathbb{E}_{\mathbf{S} \sim \mathbb{P}_{\mathcal{L}}}[d_{\mathrm{TV}}(\mathbb{P}_{\mathcal{L}}(\cdot \mid \mathbf{S}), \mathbb{P}_{\Theta}(\cdot \mid \mathbf{S}))] = \mathbb{E}_{\mathbf{S} \sim \mathbb{P}_{\mathcal{L}}}[d_{\mathrm{TV}}(\mathbf{Q}^*(\mathbf{S}, \cdot), \mathbf{Q}_f(\mathbf{S}, \cdot))]$.

**Transformer model.** Without loss of generality, $f_{\Theta}$ is assumed to be a transformer model with $L$ layers and $H$ heads, consisting of alternating multi-head attention (MHA) and feed-forward blocks (more details in Appendix B). The first layer receives an input $\mathbf{S}^{(0)} = \mathbf{S}$ embedded in $r$-dimensional space. To obtain a probability distribution on the vocabulary $\mathcal{V}$, the output $\mathbf{S}^{(L)} \in \mathbb{R}^{r \times T}$ of the final layer is projected back to the vocabulary size by an "unembedding layer" $\mathbf{W}_U \in \mathbb{R}^{T \times r}$ and averaged over the columns to obtain a vector in $\mathbb{R}^T$. A softmax layer is finally applied to obtain the probability distribution of the next token $\mathbb{P}_{\Theta}(\cdot \mid \mathbf{S}) := \mathrm{softmax}\left(\frac{1}{n\tau}\mathbf{W}_U\mathbf{S}^{(L)}\mathbb{1}_n\right) \in \Delta_T$, where $\Theta$ denotes the parameters of the entire network and $\tau$ is the softmax temperature (Hinton, 2015). Unless otherwise specified, we assume that the unembedding layer is bounded. The classes of parameters and neural networks it generates respectively write $\mathcal{W} = \{\Theta \text{ s.t. } \|\mathbf{W}_U^{\top}\|_{2,1} \leq B_U\}$ and $\mathcal{F} = \{f_{\Theta} \text{ s.t. } \Theta \in \mathcal{W}\}$.

## 4.1 PRE-TRAINING THEORETICAL ANALYSIS

We now significantly extend the scope of our theoretical contributions by assuming that the pre-training data $S = (\mathbf{S}_1, \ldots, \mathbf{S}_{N_{\mathrm{train}}})$ is a sequence of *dependent* random variables with a mild coupling structure, namely that a Marton coupling with mixing matrix $\mathbf{\Gamma}$ exists for $S = (\mathbf{S}_1, \ldots, \mathbf{S}_{N_{\mathrm{train}}})$.[4] This ensures our setting remains very broad as it subsumes the case of independent variables, $m$-dependent variables, language bigrams (Bietti et al., 2023), and the Markov chain setting considered in state-of-the-art ICL analysis of LLMs (Hu et al., 2024; Zhang et al., 2023b).

**Generalization bound.** We denote the risks by $\mathcal{R}_{\mathrm{pre}}(\Theta)$ and $\widehat{\mathcal{R}}_{\mathrm{pre}}(\Theta)$ to indicate that we take $N = N_{\mathrm{train}}$ in Eq. (2). Below, we state our main result, whose proof is deferred to Appendix D.4, which provides a generalization bound on the estimation risk of pre-training.

> **Theorem 4.1** (Pre-training generalization bound). *Consider an LLM $f_{\Theta} \in \mathcal{F}$. We denote by $\mathbf{\Gamma}$ the mixing matrix of the pre-training sequences of tokens $(\mathbf{S}_1, \ldots, \mathbf{S}_{N_{\mathrm{train}}})$. Let $0 < \delta < 1$, then with probability at least $1 - \delta$,*
>
> $$\mathcal{R}_{\mathrm{pre}}(\Theta) \leq \widehat{\mathcal{R}}_{\mathrm{pre}}(\Theta) + \frac{\bar{B}}{\sqrt{N_{\mathrm{train}}}}\sqrt{\log\left(\frac{2}{\delta}\right)},$$
>
> *where $\bar{B} = 2\|\mathbf{\Gamma}\|\max\{\log(T) + 2B_U/\tau, \log(1/c_0)\}^{1/2}$ is a constant depending on the parameters of the problem.*

The bound in Theorem 4.1 depends on the intrinsic structure of the pre-training data through the norm of the mixing matrix $\|\mathbf{\Gamma}\|$. If the pre-training data $S$ is a Markov chain with state space $\Omega$, this norm captures exactly the mixing time of the latter, making sequences that mix at a slower pace harder to learn. Secondly, and perhaps most surprisingly, this bound becomes model-independent when $\max\{\log(T) + 2B_U/\tau, \log(1/c_0)\}$ is dominated by $\log(1/c_0)$ term. Hence, if $B_U \approx \mathcal{O}(T\sqrt{r})$, which happens in practice due to the common normalization of the unembedding layer, then the model's hidden dimension $r$ and vocabulary size $T$ should be large enough to ensure $\log(T) + 2B_U/\tau \geq \log(1/c_0)$ for some unknown reference constant $c_0$. Below this threshold, the architecture of $f_{\Theta}$ is not expressive enough to have any tangible impact on its generalization, although it may affect the training error $\widehat{\mathcal{R}}_{\mathrm{pre}}(\Theta)$.

**Depth-dependent variation.** We extend Theorem 4.1 to make its dependency on $f_{\Theta}$ more fine-grained. Rather than assuming that only the norm of the embedding layer's matrix is bounded, we follow the setting of prior work (Edelman et al., 2022; Furuya et al., 2024; Marion, 2023; Zhang et al., 2023b) and consider the parameter space defined as follows:

$$\widetilde{\mathcal{W}} = \{\Theta \in \mathcal{W} \mid \forall \ell \in [L], \|\mathbf{W}_V^{(\ell)}\|_{\infty} \leq B_V, \|\mathbf{W}_O^{(\ell)}\|_{\infty} \leq B_O, \|\mathbf{W}_1^{(\ell)}\|_{\infty} \leq B_1, \|\mathbf{W}_2^{(\ell)}\|_{\infty} \leq B_2\}.$$

The definition of $\widetilde{\mathcal{W}}$ concerns the query, key, and value matrices of all layers and heads. Similarly to Zhang et al. (2023b, Assumption 5.1), we assume that each token has an $\ell_1$-norm bounded by $B_{\mathrm{tok}}$. We have the following generalization bound, whose proof is deferred to Appendix D.5.

---

[4] $\|\mathbf{\Gamma}\| = 1$ for independent variables and more details on Marton coupling can be found in Appendix C.3.

**Corollary 4.2** (Depth-dependent bound). *Consider an LLM $f_{\boldsymbol{\Theta}} \in \tilde{\mathcal{F}} := \{f_{\boldsymbol{\Theta}} \mid \boldsymbol{\Theta} \in \tilde{\mathcal{W}}\}$. With the same assumptions as in Theorem 4.1, we have*

$$\mathcal{R}_{\mathrm{pre}}(\boldsymbol{\Theta}) \leq \widehat{\mathcal{R}}_{\mathrm{pre}}(\boldsymbol{\Theta}) + \frac{\bar{B}}{\sqrt{N_{\mathrm{train}}}} \sqrt{\log\left(\frac{2}{\delta}\right)},$$

*where $\bar{B} = 2\|\boldsymbol{\Gamma}\| \max\{\log(T) + 2(B_{\boldsymbol{\Theta}})^L/\tau, \log(1/c_0)\}^{1/2}$ is a constant depending on the parameters of the problem, and $B_{\boldsymbol{\Theta}} = [(1 + rmB_1 B_2)(1 + \frac{r^3}{H} B_O B_V)](B_{\mathrm{tok}} B_U)^{1/L}$.*

We note that $\bar{B}$ exhibits an exponential dependence on the depth of the transformer, which also amplifies the hidden dimensionality (width) of the embedding layer $r$. This contrasts with the dependency in $m$, the hidden dimensionality of the MLP block, which is linear. All these factors are commonly associated with higher expressive power of transformers suggesting that they should contribute to a better minimization of $\widehat{\mathcal{R}}_{\mathrm{pre}}(\boldsymbol{\Theta})$ at the expense of requiring more training data. The number of heads $H$ can be used as a counterbalance to increasing the width in the cubic term $r^3$, suggesting that a good balance between these parameters may lead to more data-efficient models.

**Sample complexity of LLMs.** Our goal is to show the asymptotic dependence on the number of sequences that an LLM requires such that $\mathbf{Q}_f$ is $\varepsilon$-close to the reference transition matrix $\mathbf{Q}^*$. We then derive a sample complexity bound. The proof is deferred to Appendix D.6.

**Corollary 4.3** (Sample complexity). *Let $\bar{B}$ be the parameter-dependent constant of Theorem 4.1 or Corollary 4.2. Let $\delta \in [0, 1]$ and let $\epsilon > 0$. If $N_{\mathrm{train}} \geq N^* := \lceil \frac{4\bar{B}^2}{\epsilon^2} \log\left(\frac{2}{\delta}\right) \rceil$ and if we assume a perfect pre-training error for $f_{\boldsymbol{\Theta}}$, then we have with probability at least $1 - \delta$,*

$$\mathbb{E}_{\mathbf{S} \sim \mathbb{P}_{\mathcal{L}}} \|\mathbf{Q}^*(\mathbf{S}, \cdot) - \mathbf{Q}_f(\mathbf{S}, \cdot)\|_1 \leq \epsilon.$$

This result allows us to contextualize LLMs' ability to learn Markov chains with respect to the existing literature. To the best of our knowledge, the only existing approach with theoretical guarantees for learning Markov chains is the frequentist method: counting the number of occurrences of different states to fill in the matrix $\mathbf{Q}_f$. Wolfer & Kontorovich (2019) show that the sample complexity of approximating $\mathbf{Q}^*$ up to $\epsilon$ with such approach requires at most $\mathcal{O}(\max\{|\mathcal{V}_K^*|/\epsilon^2 \gamma_s, 1/\gamma_s \pi^*\})$ samples, where $\gamma_s$ is a (pseudo) spectral gap of the Markov chain and $\pi^*$ is the smallest element of its stationary distribution. The authors state that the frequentist approach is minimax optimal (up to logarithmic factors). Our bound has a dependence that behaves as $\bar{B}^2 = \mathcal{O}(\max\{\log T + \frac{2T\sqrt{r}}{\tau}, \log(1/c_0)\})$. Given that in practice $T > r$, it then simplifies to $\mathcal{O}(\max\{T/\epsilon^2 \tau, 1/\epsilon^2\})$. Note that the LLMs' sample complexity is linear in the vocabulary size $T$, which is remarkable compared to the sample complexity of the frequentist approach, which scales as $\mathcal{O}(T^K)$. We show in Section 5 that this is confirmed experimentally: LLM's ability to learn Markov chains exceeds the frequentist approach for Markov chains with a large state space.

### 4.2 IN-CONTEXT LEARNING OF MARKOV CHAINS

Although insightful, the analysis presented above is related to the pre-training of LLMs – a process that is hard and extremely costly to reproduce in practice. Similarly, we do not have access to the ground-truth matrix $\mathbf{Q}^*$ to reason about LLM's ability to infer it in practice. To provide theoretical results that can be confirmed experimentally, we now turn our attention to in-context learning of Markov chains: a setup where one provides an LLM with an input sequence formed by a Markov chain of size $N_{\mathrm{icl}}$ defined over a state space $\Omega$ of size $d$[5]. Different from the setting of Section 4.1, we now can explicitly use a transition kernel $\mathbb{P}$ of this Markov chain for the theoretical analysis by replacing $\mathbb{P}_{\mathcal{L}}$ with it in the definition of $\mathcal{R}_{\mathrm{icl}}(\boldsymbol{\Theta})$ and $\widehat{\mathcal{R}}_{\mathrm{icl}}(\boldsymbol{\Theta})$ in Eq. (2) (see Appendix D.7 for details on the problem setup). To relate the generalization error to the pre-training error, we quantify

---

[5]This is different from another variation of ICL where supervised $(x, y)$ pairs are provided in-context. Rather, the supervision is provided from observing transitions between states $(x_i, x_{i+1} = f(x_i))$ as discussed in (Li et al., 2023, Fig.1).

the discrepancy between an LLM pre-trained mostly on textual data, and a hypothetical LLM with parameters in $\mathcal{W}_{\mathrm{mc}}$ that is pre-trained on a dataset of Markov chains with the same data distribution as the Markov chain used as an input during in-context inference. We define the divergence between two estimated transition matrices $\mathbb{P}_{\Theta_1}, \mathbb{P}_{\Theta_2}$ as

$$\mathcal{K}(\Theta_1, \Theta_2) \coloneqq \frac{1}{N} \sum_{n=1}^{N} \mathbb{E}_{\mathbf{S}_n}[d_{\mathrm{TV}}(\mathbb{P}_{\Theta_1}(\cdot \mid \mathbf{S}_n), \mathbb{P}_{\Theta_2}(\cdot \mid \mathbf{S}_n))]. \tag{3}$$

The operator $\mathcal{K}$ is akin to a distance (the separation property is only verified almost surely, see Appendix C.4 for more details). The next result, whose proof is deferred to Appendix D.7, provides a generalization bound on the in-context learning phase.

> **Theorem 4.4** (In-Context Learning generalization bound). *Consider an LLM $f_{\Theta} \in \mathcal{F}$. We provide as input of $f_{\Theta}$ a $d-$state Markov chain $X = (\mathbf{X}_1, \dots, \mathbf{X}_{N_{\mathrm{icl}}})$. The sequence of subsequences of the first $n$ terms is denoted by $S = (\mathbf{S}_1, \dots, \mathbf{S}_n)$. $S$ is also a Markov chain, and we denote by $t_{\mathrm{mix}}(\varepsilon)$ its mixing time. Let $t_{\min} \coloneqq \inf_{0 \leq \varepsilon < 1} t_{\mathrm{mix}}\left(\frac{\varepsilon}{2}\right)\left(\frac{2-\varepsilon}{1-\varepsilon}\right)^2$. Let $\delta > 0$. Then, with probability at least $1 - \delta$,*
>
> $$\mathcal{R}_{\mathrm{icl}}(\Theta) \leq \inf_{\vartheta \in \mathcal{W}_{\mathrm{mc}}} \{\widehat{\mathcal{R}}_{\mathrm{icl}}(\vartheta) + \mathcal{K}(\vartheta, \Theta)\} + \bar{B}\sqrt{\frac{t_{\min}}{N_{\mathrm{icl}}}}\sqrt{\log\left(\frac{2}{\delta}\right)}, \tag{4}$$
>
> *where $\bar{B} = 2\max\{\log(d) + 2B_U/\tau, \log(1/p_{\min})\}^{1/2}$.*

We first note that instead of the norm of the mixing matrix $\Gamma$ seen before, we now have an explicit dependency on $t_{\min}$, which is related to the mixing time of the input Markov chain. This, together with the availability of the ground-truth transition matrix, allows us to use Theorem 4.4 to derive and verify experimentally the scaling laws of ICL for popular LLMs. Theorem 4.4 also suggests that an LLM pre-trained on diverse data sequences different from Markov chains should exhibit a certain degree of invariance to correctly infer the transition probabilities of the latter. This is reminiscent of the domain adaptation bounds (Redko et al., 2019) that also commonly involve a distribution shift (i.e., a distance or a divergence) term that vanishes if the model is invariant to classes of transformations linking the distribution of the input data with that on which it is applied during inference. A recent success of applying LLMs to time series data (Gruver et al., 2023), for instance, suggests that this term is indeed small for certain types of data not used during pre-training.

## 5 NUMERICAL EXPERIMENTS

Theorem 4.4 provides a practically verifiable result which naturally stems from our analysis in Section 4. We then evaluate the ability of recent LLMs, namely `Mistral 7Bv0.1` (Jiang et al., 2023), `Llama2 7B & 13B` (Touvron et al., 2023b), and `Gemma 2B` (Team et al., 2024) to infer transition probabilities of Markov chains in-context. We associate each state in the $d$-state Markov chain with a token from the set $\{0, \dots, d-1\}$, concatenated to obtain a prompt of length $N_{\mathrm{icl}}$. Bearing in mind the differences in the tokenization mechanisms of the different models, we add comas whenever necessary to ensure that each state is tokenized separately. More details on the experimental setup and additional experiments with more Markov chains and with `Llama3.2` (Dubey et al., 2024) are available in Appendix E.1.

**Dependence on $N_{\mathrm{icl}}$.** We first analyze the effect of $N_{\mathrm{icl}}$ on the risk calculated for a randomly generated 3-state Markov transition matrix. From the results presented in Fig. 5(left), we note that `Llama2` models deviate from our $\mathcal{O}(N_{\mathrm{icl}}^{-1/2})$ theoretical scaling law, while most recent models (`Mistral` and `Gemma`) stay much closer to Theorem 4.4, similarly to what was observed by Cabannes et al. (2024). Being randomly generated, the Markov chains provided to the models have not been seen during training, and older (weaker) models naturally struggle to generalize.

**Dependence on $t_{\min}$.** Theorem 4.4 states that Markov chains with slow mixing (higher $t_{\min}$) are slower to learn. We now plot the true risk for a single model with different values of $t_{\min}$ highlighting in Fig. 5(right) a two-stage regime of ICL. In a first stage, the bound in Eq. (4) is dominated by

$\sqrt{t_{\min}/N_{\mathrm{icl}}}$ for small $N_{\mathrm{icl}}$, and depends strongly on $t_{\min}$, while the scaling law $\mathcal{O}(N_{\mathrm{icl}}^{-1/2})$ dominates as $N_{\mathrm{icl}}$ increases beyond $N_{\mathrm{icl}} \approx 20$.

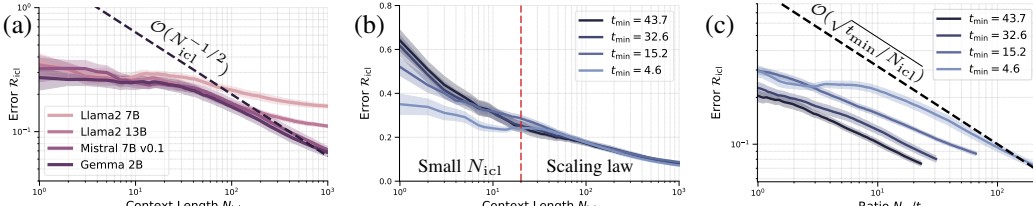

Figure 5: **In-context scaling laws.** The risk $\mathcal{R}_{\mathrm{icl}}$ as functions of $N_{\mathrm{icl}}$, with 95% confidence intervals. (a) Risks for different LLMs along with the scaling law of Theorem 4.4. (b)-(c) Risks with Mistral 7B v0.1 for random 3-state transition matrices and different $t_{\min}$ as functions of $N_{\mathrm{icl}}$ and $N_{\mathrm{icl}}/t_{\min}$.

**Dependence on $d$.** We now verify Theorem 4.4 for Markov chains with a different state space size (previously $d = 3$). We also consider a baseline given by the frequentist method mentioned before. We recall that, for the latter, its dependence on $d$ behaves like $\mathcal{O}(\sqrt{d/N_{\mathrm{icl}}})$, while Theorem 4.4 gives $\mathcal{O}(\sqrt{\log(d)/N_{\mathrm{icl}}})$. For Markov chains with a small number of states $d$, there is no clear difference between the frequentist estimator and a LLM. However, as $d$ grows the frequentist estimator struggles to estimate the transition matrix due to the $\mathcal{O}(\sqrt{d})$ scaling factor. This is verified experimentally in Fig. 6, where we vary the parameter $d$ from 3 (left) to 700 (right). We observe that the LLM follows the theoretical neural scaling law $\mathcal{O}(N_{\mathrm{icl}}^{-1/2})$ and outperforms the frequentist method for $d = 700$, while being close to it for $d = 3$. We conclude that our analysis gives theoretical insights on the ICL neural scaling law observed empirically in (Liu et al., 2024). The additional experiments conducted in Appendix E.5 show that our bounds remain valid for large values of $d$.

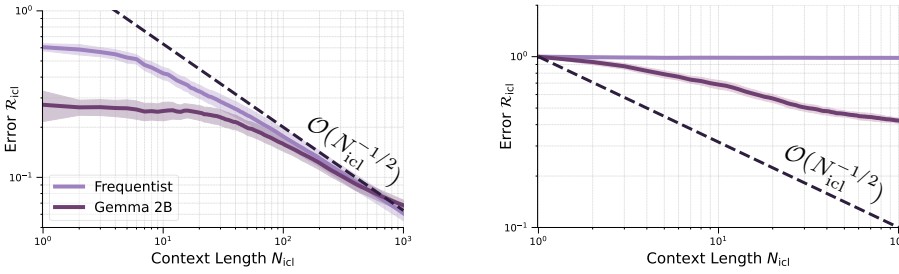

Figure 6: **Impact of the number of states.** We plot the risks $\mathcal{R}_{\mathrm{icl}}$ as functions of $N_{\mathrm{icl}}$ for Gemma 2B and the frequentist approach (Wolfer & Kontorovich, 2019) with 95% confidence intervals. **Left.** The input sequence is a random 3-state Markov chain. **Right.** The input sequence is a Brownian motion discretized as a 700-state Markov chain, similarly to Liu et al. (2024).

## 6 CONCLUSION

This paper proposed an explicit characterization of the inference mechanism in large language models through an equivalent finite-state Markov chain. We provided an insightful theoretical analysis based on the established characterization and the ability of the LLM to infer the transition kernel approximating the true transition probabilities of language. We adapted our results to in-context learning where experiments confirm our theoretical insights. In the future, we hope that the proposed equivalence will have far-reaching implications on our understanding of LLMs and allow for a more fine-grained understanding of their expressiveness.

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

# Appendix

**Roadmap.** In Appendix A, we first recall our notations. We provide additional details on large language models and transformers in Appendix B. Important notions and definitions related to Markov chains and Marton couplings are given in Appendix C. The detailed proofs of our theoretical results are given in Appendix D. Finally, we provide additional experiments in Appendix E.

## TABLE OF CONTENTS

## A  NOTATIONS

We denote $\{1, \cdots, N\}$ as $[N]$. We represent scalar values with regular letters (e.g., parameter $\lambda$), vectors with bold lowercase letters (e.g., vector $\mathbf{x}$), and matrices with bold capital letters (e.g., matrix $\mathbf{A}$). The $i$-th row of the matrix $\mathbf{A}$ is denoted by $\mathbf{A}_i$, its $j$-th column is denoted by $\mathbf{A}_{.j}$ and its transpose is denoted by by $\mathbf{A}^\top$. The identity matrix of size $n$ is denoted by $\mathbf{I}_n \in \mathbb{R}^{n \times n}$. The vector of size $n$ with each entry equal to 1 is denoted by $\mathbb{1}_n$. We denote by $\|\mathbf{A}\|_{p,q}$ the $L_{p,q}$ matrix norm where the $p$-norm is over columns and the $q$-norm is over rows. We denote by $\|\mathbf{A}\|$ the operator norm of $\mathbf{A}$ induced by the $\ell_2$ norm and by $\|\mathbf{A}\|_\infty = \max_{ij}|\mathbf{A}_{ij}|$ the operator norm induced by the $\ell_\infty$-norm. Similarly, $\mathbf{x}^\top$ is the transpose of the vector $\mathbf{x}$ and $\|\mathbf{x}\|_p$ is its $\ell_p$-norm. The total variation between two probability distributions $\mathbb{P}, \mathbb{Q}$ is denoted by $d_{\mathrm{TV}}(\mathbb{P}, \mathbb{Q})$. The term "almost surely" is denoted by the notation "a.s." while the term *random variable* is denoted by the notation "r.v.". $\Delta_n := \{\mathbf{p} \in [0,1]^n | \sum_{i=1}^n \mathbf{p}_i = 1\}$ is the probability simplex of $\mathbb{R}^n$.

## B  BACKGROUND ON LARGE LANGUAGE MODELS

We first recall important notions regarding large language models before focusing on the most widely used ones, namely the transformer-based LLMs. We describe the components of the vanilla transformer architecture before describing the whole network at the heart of such a model and formally defining the class of parameters and neural networks considered in our work.

### B.1  LARGE LANGUAGE MODELS

In this section, we recall how the sequences of tokens are processed by the large language model notably regarding the next token generation and the deletion process.

> **Definition B.1** (Generation process). *Given an input $s \in \mathcal{V}_K^*$ of size $p$, an large language model outputs a probability mass function $f_{\boldsymbol{\Theta}}^{T,K}(s)$ over the discrete vocabulary space. A next token $x$ is then sampled from $f_{\boldsymbol{\Theta}}^{T,K}(s)$, to construct a new sequence $(s, x)$ of size $p + 1$.*

Generation can be repeated by considering $(s, x)$ as new input sequence and iterating this process. Since these models are designed to handle only sequences of size at most $K$, a deletion process is required.

> **Definition B.2** (Deletion process). *Given an input $s$ of size $p > K$, an large language model outputs a probability mass function $f_{\boldsymbol{\Theta}}^{T,K}(s_K)$ where $s_K$ is a truncation of $K$ tokens of the sequence $s$. large language models implement **front truncation**, which is done by setting $s_K$ as the last $K$ tokens of $s$.*

As shown in Fig. 7, only the last $K$ tokens of a long input sequence are used. This is why we speak of *deletion*, since we ignore the first tokens.

Note that it is possible to implement other kinds of truncation, but large language models usually do not (Brown et al., 2020; Touvron et al., 2023a), however, in models like BERT (Devlin et al., 2019), which are not autoregressive, back truncation as described in Fig. 8 is also an option.

### B.2  TRANSFORMER ARCHITECTURE

The most popular autoregressive LLMs rely on the transformer architecture (Vaswani et al., 2017) which we describe below following (Brown et al., 2020; Edelman et al., 2022; Zhang et al., 2023b). An autoregressive transformer-based LLM takes as input a sequence of length $n$, with $n \leq K$ and $K$ is the context window, tokens with values in a vocabulary $\mathcal{V}$ of size $T$. The tokens are embedded into a $r$-dimensional space and the input can be written as $\mathbf{S} \in \mathbb{R}^{r \times n}$. We consider a transformer model with $L$ layers and $h$ heads. The output of the $\ell$-th layer writes $\mathbf{S}^{(\ell)}$ and is fed as input of the $(\ell + 1)$-th layer. The input of the whole model is $\mathbf{S}^{(0)} = \mathbf{S}$. Below, we describe the operations performed by the model, including the embeddings of the tokens.

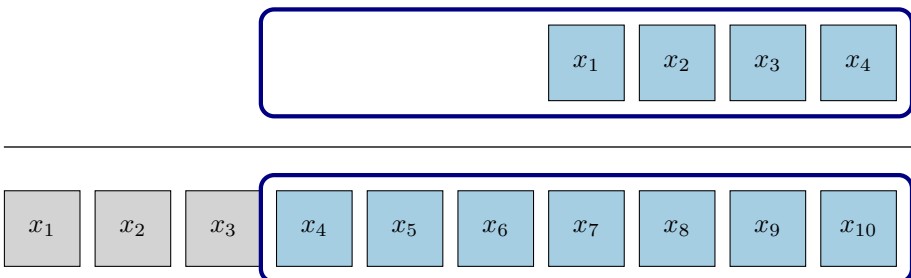

Figure 7: **Deletion process, front truncation**. A large language model with context window $K = 7$ in navy blue, processing sequences of different lengths. **Top.** A sequence of length 4. **Bottom.** Front truncation of a sequence of length 10.

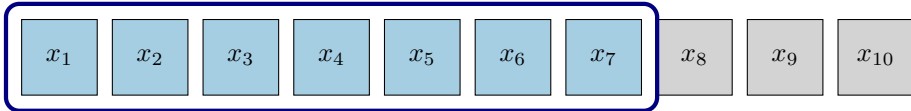

Figure 8: **Back truncation**. A large language model with context window $K = 7$ in navy blue, processing back truncation of a sequence of a sequence of length 10.

- **Token embeddings.** The tokens are embedded in a $r$-dimensional space via an embedding layer $\mathbf{W}_E$ which results in an input of the form $\mathbf{S}^{r \times n}$;

- **Positional embeddings.** (Learnable) positional embeddings are added to each token depending on its position in the input sequence. This breaks the permutation-invariance of the transformer architecture and leads, by abuse of notation, to an output $\mathbf{S} \in \mathbb{R}^{r \times n}$;

- **Multi-head attention (MHA).** Given an input sequence $\mathbf{S} \in \mathbb{R}^{r \times n}$, query, key, and value matrices $\mathbf{W}_Q, \mathbf{W}_K, \mathbf{W}_V \in \mathbb{R}^{r \times r}$ (here the value and output matrices are merged for ease of notations), the self-attention module computes

$$\mathcal{A}(\mathbf{S}; \mathbf{W}_Q, \mathbf{W}_K, \mathbf{W}_V) \coloneqq \mathrm{softmax}\Big(\mathbf{W}_Q \mathbf{S} (\mathbf{W}_K \mathbf{S})^\top / \sqrt{r}\Big)(\mathbf{W}_V \mathbf{S}) \in \mathbb{R}^{r \times n},$$

with $\mathrm{softmax} \colon \mathbf{x} \in \mathbb{R}^n \to \exp(\mathbf{x}) / \sum_i \exp(\mathbf{x})_i \in \Delta_n$. The operation described below corresponds to single-head self-attention. In practice, multi-head attention (MHA) is used with $H$ heads and the query and key matrices are in $\mathbb{R}^{\frac{r}{H} \times \frac{r}{H}}$ and the value matrix is in $\mathbb{R}^{\frac{r}{H} \times r}$ ($r, H$ are taken such that $\frac{r}{H}$ is an integer). The MHA module concatenates on the row dimension the outputs of $\mathcal{A}$ for each head and then projects it back to the embedding dimension $r$ with an output matrix $\mathbf{W}_O \in \mathbb{R}^{r \times r}$. By abuse of notation, we also denote by $\mathcal{A}$ this operation which results in an output of dimension $r \times n$, and we include the output matrix in the argument of the operator. The $\ell$-th layer of the transformer applies attention with layer-specific weight matrices and a residual connection that leads to an output

$$\mathbf{Z}^{(\ell)} = \mathbf{S}^{(\ell-1)} + \mathcal{A}\Big(\mathbf{S}^{(\ell-1)}; \mathbf{W}_Q^{(\ell)}, \mathbf{W}_K^{(\ell)}, \mathbf{W}_V^{(\ell)}, \mathbf{W}_O^{(\ell)}\Big).$$

This is followed by a layer normalization (Zhang & Sennrich, 2019) that projects each token into the $\ell_2$-unit ball, i.e., each column $\mathbf{S}_{\cdot,n}^{(\ell)}$ has an $\ell_2$-norm lower than 1;

- **Feed-forward block (FF).** Finally, a feed-forward block is applied, consisting of a two-layer MLP with hidden dimension $m$, layer-specific weight matrices $\mathbf{W}_1^{(\ell)} \in \mathbb{R}^{m \times r}, \mathbf{W}_2^{(\ell)} \in \mathbb{R}^{r \times m}$ and ReLU activation denoted by $\mathrm{ReLU}(x) = \max\{0, x\}$ and applied entry-wise. The output of this layer reads

$$\mathbf{Y}^{(\ell)} = \mathbf{W}_2^{(\ell)} \mathrm{ReLU}\Big(\mathbf{W}_1^{(\ell)} \mathbf{Z}^{(\ell)}\Big).$$

It is followed by a residual connection to produce the output

$$\mathbf{S}^{(\ell)} = \mathbf{Z}^{(\ell)} + \mathbf{W}_2^{(\ell)} \mathrm{ReLU}\Big(\mathbf{W}_1^{(\ell)} \mathbf{Z}^{(\ell)}\Big) \in \mathbb{R}^{r \times n},$$

on which layer normalization (Zhang & Sennrich, 2019) is applied ensuring that each column $\mathbf{S}^{(\ell)}_{\cdot,n}$ has an $\ell_2$-norm lower than 1.

- **softmax output layer.** In the autoregressive setting, the model outputs a probability distribution on the vocabulary $\mathcal{V}$. To that end, the output $\mathbf{S}^{(L)} \in \mathbb{R}^{r \times n}$ of the final layer is projected back to the vocabulary size by an "unembedding layer" $\mathbf{W}_U \in \mathbb{R}^{T \times r}$ and averaged over the columns to obtain a vector in $\mathbb{R}^T$. A softmax layer is finally applied on top of it to obtain the probability distribution of the next token $\mathbb{P}_{\boldsymbol{\Theta}}(\cdot \mid \mathbf{S})$. Formally, we have

$$\mathbb{P}_{\boldsymbol{\Theta}}(\cdot \mid \mathbf{S}) = \mathrm{softmax}\left(\frac{1}{n\tau}\mathbf{W}_U\mathbf{S}^{(L)}\mathbb{1}_n\right) \in \Delta_T,$$

$n$ is the length (i.e., number of columns) of the input sequence $\mathbf{S}$ (and thus of the last layer output $\mathbf{S}^{(L)}$), $\boldsymbol{\Theta}$ denotes the parameters of the whole network that subsume the parameters of each layer and each block and $\tau$ is the softmax temperature (Hinton, 2015).

### B.3 AUTOREGRESSIVE TRANSFORMER-BASED LLM

The architecture described above is used in most of the transformer-based autoregressive LLM (Anil et al., 2023; Brown et al., 2020; Dubey et al., 2024; Jiang et al., 2023). In the theoretical analysis of Section 4, and unless specified otherwise, we remain faithful to their practical implementation and only make the following mild assumption: we assume that the unembedding layer is bounded. The class of parameters and the class of neural networks it generates respectively writes

$$\mathcal{W} = \{\boldsymbol{\Theta} \mid \|\mathbf{W}_U^\top\|_{2,1} \leq B_U\} \quad \text{and} \quad \mathcal{F} = \{f_{\boldsymbol{\Theta}} \mid \boldsymbol{\Theta} \in \mathcal{W}\}.$$

It should be noted that this assumption is significantly weaker than what is usually done in the literature (Edelman et al., 2022; Zhang et al., 2023b).

## C BACKGROUND ON MARKOV CHAINS

We recall below some important notions related to Markov chains based on (Paulin, 2015; Roberts & Rosenthal, 2004) and that will be used in our proofs.

### C.1 BASIC NOTIONS

Consider two distribution probabilities $\mathbb{P}$ and $\mathbb{Q}$ defined on a measurable space $(\Omega, \mathcal{F})$.

**Definition C.1.** *The total variation between $\mathbb{P}$ and $\mathbb{Q}$ is defined as*

$$d_{\mathrm{TV}}(\mathbb{P}, \mathbb{Q}) := \sup_{A \in \mathcal{F}} |\mathbb{P}(A) - \mathbb{Q}(A)|.$$

In the setting considered in the main paper, we consider Markov chains with finite discrete state space $\Omega$. In this section, we refer to $\Omega$ as a general Polish space, whose elements are referred to as *states*.

Informally, a discrete-time, time-homogeneous Markov chain with state space $\Omega$ is a sequence of random variables $(\mathbf{X}_1, \mathbf{X}_2, \ldots)$ taking values in $\Omega$ such that the next observation is independent of the past given the present. This property is referred to as the Markov property and is defined below.

**Definition C.2.** *A sequence of random variables $(\mathbf{X}_1, \mathbf{X}_2, \ldots)$ is said to satisfy the Markov property if for all $n \geq 1$ and any $(x_1, \ldots, x_{n+1}) \in \Omega^{n+1}$*

$$\mathbb{P}(\mathbf{X}_{n+1} = x_{n+1} \mid \mathbf{X}_n = x_n, \cdots, \mathbf{X}_1 = x_1) = \mathbb{P}(\mathbf{X}_{n+1} = x_{n+1} \mid \mathbf{X}_n = x_n).$$

To a given Markov chain, we associate its *transition kernel* $\mathbf{Q} : \Omega^2 \to [0, 1]$ which collects the transition probabilities from one state to another

$$\forall n \in \mathbb{N}, (x, y) \in \Omega^2, \quad \mathbf{Q}(x, y) = \mathbb{P}(\mathbf{S}_{n+1} = y \mid \mathbf{S}_n = x).$$

In the main text, we refer to $\mathbf{Q}$ as a transition *matrix* as the Markov chains we consider are of finite state space.

**Definition C.3.** *A distribution $\pi$ on $\Omega$ is said to be a stationary distribution if the action of the transition kernel leaves $\pi$ unchanged, that is*

$$(\mathbf{Q}\pi)(A) := \int_{y \in A} \mathbf{Q}(x,y) d\pi(x) = \pi(A)$$

*for all $A \in \mathcal{F}$.*

A natural question is whether such a distribution exists for a generic Markov chain. Before stating an existence theorem, we introduce a classification of states below.

**Class of states.** All definitions bellow are borrowed from (Gallager, 1996)

**Definition C.4** (Accessibility and communication). *A state $x$ is accessible from $y$ (abbreviated as $x \to y$) if there exists $n > 0$ such that $\mathbf{Q}^n(x,y) > 0$. Two distinct states $x$ and $y$ communicate (abbreviated $x \leftrightarrow y$) if $x$ is accessible from $y$ and $y$ is accessible from $x$.*

Accessibility and communication concepts define how states can reach each other within a Markov chain. This leads to an important classification of states into transient and recurrent categories.

**Definition C.5** (Recurrent and transient states). *For finite-state Markov chains, a recurrent state is a state $i$ that is accessible from all states that are accessible from $i$ ($i$ is recurrent if $i \to j$ implies that $j \to i$). A transient state is a state that is not recurrent.*

With the distinction between recurrent and transient states established, we can now group states into classes based on their communication properties.

**Definition C.6** (Class of states). *A class $\mathcal{C}$ of states is a non-empty set of states such that each $i \in \mathcal{C}$ communicates with every other state $j \in \mathcal{C}$ and communicates with no $j \notin \mathcal{C}$*

**Aperiodicity and Ergodicity.** Aperiodicity ensures that the system does not exhibit cyclic behavior, which is a key condition for understanding the asymptotic behavior of states.

**Definition C.7** (Aperiodicity). *The period of a state $i$, denoted $d(i)$, is the greatest common divisor (gcd) of those values of $n$ for which $\mathbf{Q}^n(i,i) > 0$. If the period is 1, the state is aperiodic.*

Under some conditions on the Markov chain (aperiodicity and irreducibility (Roberts & Rosenthal, 2004)), it can be proven that the chain converges to its stationary distribution i.e. for any $x \in \Omega$, $\lim_{n \to \infty} d_{\mathrm{TV}}(Q^n(x,\cdot), \pi) = 0$, where $Q^n(x,\cdot)$ denotes the probability of $\mathbf{S}_n$ conditioned on $\mathbf{S}_1 = x$.

We recall below the notion of mixing time that assesses the time taken by the Markov chain to be $\varepsilon$-close to its stationary distribution (see Definition C.8).

**Definition C.8** (Mixing time for time-homogeneous Markov chains (Paulin, 2015)). *Let $X := (\mathbf{X}_1, \mathbf{X}_2, \ldots)$ be a time-homogeneous Markov chain with a state space $\Omega$, a transition kernel $Q$, and a stationary distribution $\pi$. Its mixing time is defined for any $\varepsilon \in [0,1]$ as*

$$t_{\mathrm{mix}}(\varepsilon) := \min\{t \mid d(t) \leq \varepsilon\} \text{ where } d(t) := \sup_{x \in \Omega} d_{\mathrm{TV}}(Q^t(x,\cdot), \pi).$$

We also introduce the quantity

$$t_{\min} := \inf_{0 \leq \varepsilon < 1} t_{\mathrm{mix}}\left(\frac{\varepsilon}{2}\right) \cdot \left(\frac{2-\varepsilon}{1-\varepsilon}\right)^2$$

which will be useful later on.

**Remark C.1** (Well-posedness of $t_{\min}$). *As we only consider finite state-space Markov chains in our work, we know that a stationary distribution always exists. However, its uniqueness and the convergence to it require additional assumptions (see Appendix C.2). In particular, not all Markov chains admit a finite $t_{\mathrm{mix}}(\varepsilon)$, $t_{\min}$ for some $\varepsilon < \frac{1}{2}$. In such case, $t_{\min}$ can be infinite. In our practical experimentation, this is never the case despite considering various Markov chains.*

## C.2 Ergodic Unichains

We are now ready to state the following theorem, which formalizes the classification of states into recurrent, transient, and aperiodic classes.

> **Theorem C.9** (Recurrent and transient classes). *For finite state Markov chains, either all states in a class are transient or all are recurrent. We refer to these classes as transient and recurrent, respectively.*
>
> *For any Markov chain, all states in a class have the same period. If the period is 1, the class is said to be aperiodic*

Having categorized states into recurrent, transient, and aperiodic classes, we can now define ergodicity.

> **Definition C.10** (Ergodicity). *For a finite-state Markov chain, an ergodic class of states is a class that is both recurrent and aperiodic. A Markov chain consisting entirely of one ergodic class is called an ergodic chain.*

**Unichains.** We now introduce the concept of unichains.

> **Definition C.11** (Unichains and ergodic unichains). *A unichain is a finite-state Markov chain containing a single recurrent class and transient states. An ergodic unichain is a unichain for which the recurrent class is ergodic.*

## C.3 Marton Couplings

While we consider Markov chain inputs in Section 4.2, we consider less structured inputs during the pre-training phase Section 4.1.

More specifically, we model the sequences of tokens used during pre-training as generic dependent random variables. To derive meaningful results, we rely on the notion of Marton couplings introduced by Marton (2004). A Marton coupling can be seen as a weak dependency structure between random variables. The associated notion of the mixing matrix, analogous to the mixing time of a Markov chain, is used to assess the strength of the dependence between those variables.

This minimal modeling choice is made to remain as faithful as possible to the pre-training considered in practical applications of LLMs, for which the pre-training data is not public and may contain arbitrary data points such as video, code snippets, text and images (Achiam et al., 2023; Anil et al., 2023; Brown et al., 2020; Dubey et al., 2024; Jiang et al., 2023; Touvron et al., 2023a).

As shown in Paulin (2015, Remark 2.2.), considering sequences of random variables linked through a Marton coupling is a weaker assumption than what is usually done in the literature on generalization bounds, which typically relies on independent random variables and Markov chains (Hu et al., 2024; Marion, 2023; Wolfer & Kontorovich, 2019; Zhang et al., 2023b).

In particular, the results stated in Section 4.1 encompass the case where the pre-training input sequences of tokens are independent random variables (Kim et al., 2024) or Markov chains (Zhang et al., 2023b). We also note that Markov chains can model bigrams used in natural language (Bietti et al., 2023; Jurafsky & Martin, 2024).

We do not provide an exhaustive review of Marton couplings. We will simply recall its definition and introduce the associated mixing matrix. We refer the interested reader to Marton (2004) and Paulin

(2015). Consider a sequence of dependent random variables $S = (\mathbf{S}_1, \ldots, \mathbf{S}_N)$ taking values in a polish space $\Omega = \Omega_1 \times \ldots \times \Omega_N$. We will denote by $\mathbb{P}(\mathbf{S}_1, \ldots, \mathbf{S}_N)$ the distribution of $S$.

---

**Definition C.12** (Marton coupling). *We define a* Marton coupling *for $S$ as a set of couplings*

$$\left( S^{(s_1, \ldots, s_i, s_i')}, S'^{(s_1, \ldots, s_i, s_i')} \right) \in \Omega \times \Omega,$$

*for every $i \in [N]$, every $s_1 \in \Omega_1, \ldots, s_i \in \Omega_i, s_i' \in \Omega_i$, satisfying the following conditions.*

*(i)* $\quad \mathbf{S}_1^{(s_1, \ldots, s_i, s_i')} = s_1, \qquad \ldots, \qquad \mathbf{S}_i^{(s_1, \ldots, s_i, s_i')} = s_i,$
$\quad \mathbf{S}_1'^{(s_1, \ldots, s_i, s_i')} = s_1, \quad \ldots, \quad \mathbf{S}_{i-1}'^{(s_1, \ldots, s_i, s_i')} = s_{i-1}, \quad \mathbf{S}_i'^{(s_1, \ldots, s_i, s_i')} = s_i'.$

*(ii)* $\quad \left( \mathbf{S}_{i+1}^{(s_1, \ldots, s_i, s_i')}, \ldots, \mathbf{S}_N^{(s_1, \ldots, s_i, s_i')} \right)$
$\qquad \sim \mathbb{P}(\mathbf{S}_{i+1}, \ldots, \mathbf{S}_N \mid \mathbf{S}_1 = s_1, \ldots, \mathbf{S}_i = s_i),$
$\qquad \left( \mathbf{S}_{i+1}'^{(x_1, \ldots, x_i, x_i')}, \ldots, \mathbf{S}_N'^{(x_1, \ldots, x_i, x_i')} \right)$
$\qquad \sim \mathbb{P}(\mathbf{S}_{i+1}, \ldots, \mathbf{S}_N \mid \mathbf{S}_1 = x_1, \ldots, \mathbf{S}_{i-1} = x_{i-1}, \mathbf{S}_i = x_i').$

*(iii) If $x_i = x_i'$, then $S^{(x_1, \ldots, x_i, x_i')} = S'^{(x_1, \ldots, x_i, x_i')}$.*

---

**Definition C.13** (Mixing matrix (Paulin, 2015)). *For a Marton coupling, we define the* mixing matrix $\mathbf{\Gamma} \in \mathbb{R}^{N \times N}$ *as an upper diagonal matrix with*

$$\forall 1 \leq i < j \leq N, \quad \begin{cases} \mathbf{\Gamma}_{i,i} := 1, \\ \mathbf{\Gamma}_{j,i} := 0 \\ \mathbf{\Gamma}_{i,j} := \sup_{s_1, \ldots, s_i, s_i'} \mathbb{P}\left[ \mathbf{S}_j^{(s_1, \ldots, s_i, s_i')} \neq \mathbf{S}_j'^{(s_1, \ldots, s_i, s_i')} \right] \end{cases}.$$

---

For independent random variables, one can define a Marton coupling with a mixing matrix equal to the identity (see Paulin, 2015, Remark 2.2). In particular, it means that for independent variables, we have the operator norm of the mixing matrix equal to 1, i.e., $\|\mathbf{\Gamma}\| = 1$.

## C.4 An (Almost) Distance between Markov Chains

In Theorem D.23, We state elementary properties of $\mathcal{K}$ in the proposition below.

---

**Proposition C.14** (Properties of $\mathcal{K}$). *$\mathcal{K}$ is an almost-distance between transition matrices in the sense that it satisfies the properties below:*

1. ***Non-negativity**. For any $\mathbf{\Theta}_1, \mathbf{\Theta}_2$, $\mathcal{K}(\mathbf{\Theta}_1, \mathbf{\Theta}_2) \geq 0$.*

2. ***Almost sure positivity**. $\mathcal{K}(\mathbf{\Theta}_1, \mathbf{\Theta}_2) = 0 \iff \forall n \in [N], \mathbb{P}_{\mathbf{\Theta}_1}(\cdot \mid \mathbf{S}_n) = \mathbb{P}_{\mathbf{\Theta}_2}(\cdot \mid \mathbf{S}_n)$ a.s..*

3. ***Symmetry**. For any $\mathbf{\Theta}_1, \mathbf{\Theta}_2$, $\mathcal{K}(\mathbf{\Theta}_1, \mathbf{\Theta}_2) = \mathcal{K}(\mathbf{\Theta}_1, \mathbf{\Theta}_2)$.*

4. ***Triangular inequality.**. For any $\mathbf{\Theta}_1, \mathbf{\Theta}_2, \mathbf{\Theta}_3$, $\mathcal{K}(\mathbf{\Theta}_1, \mathbf{\Theta}_3) \leq \mathcal{K}(\mathbf{\Theta}_1, \mathbf{\Theta}_2) + \mathcal{K}(\mathbf{\Theta}_2, \mathbf{\Theta}_3)$.*

---

*Proof of Proposition C.14.* We first recall the following technical lemma.

---

**Lemma C.15** (Proposition 2.16 in Folland (1999)). *Let $Y$ be a non-negative random variable defined on a probability space $\Omega$ with probability function $\mathbb{P}$. If $\mathbb{E}[Y] = 0$, then $Y = 0$ almost*

---

*surely, i.e.,*

$$\mathbb{P}(\{\omega \in \Omega \mid Y(\omega) = 0\}) = 1$$

The non-negativity and symmetry of $\mathcal{K}$ directly come from the symmetry and non-negativity of the total variation distance. The triangular inequality follows from the fact that the total variation is a distance and that the expectation respects inequalities. For the almost positivity, consider $\boldsymbol{\Theta}_1, \boldsymbol{\Theta}_2$ such that $\mathcal{K}(\boldsymbol{\Theta}_1, \boldsymbol{\Theta}_2) = 0$. By non-negativity of all the terms in the sum, it means that for all $n \in [N]$, we have

$$\mathbb{E}_{\mathbf{S}_n}[d_{\mathrm{TV}}(\mathbb{P}_{\boldsymbol{\Theta}_1}(\cdot \mid \mathbf{S}_n), \mathbb{P}_{\boldsymbol{\Theta}_2}(\cdot \mid \mathbf{S}_n))] = 0.$$

As the total variation is a distance, we know that the random variable under the expectation is non-negative. Applying Lemma C.15 leads to

$$d_{\mathrm{TV}}(\mathbb{P}_{\boldsymbol{\Theta}_1}(\cdot \mid \mathbf{S}_n), \mathbb{P}_{\boldsymbol{\Theta}_2}(\cdot \mid \mathbf{S}_n)) = 0 \quad \text{almost surely.}$$

On the probability space, deprived of the set where the distance is non-zero (which is of null measure), the total variation is equal to zero and as a distance between probability distributions, it means that on this subset of the probability space, the probabilities are equal. Again, as the set on which they are not equal is of null measure, we have

$$\mathbb{P}_{\boldsymbol{\Theta}_1}(\cdot \mid \mathbf{S}_n) = \mathbb{P}_{\boldsymbol{\Theta}_2}(\cdot \mid \mathbf{S}_n) \quad \text{almost surely.}$$

Putting everything together, we have

$$\forall n \in [N], \mathbb{P}_{\boldsymbol{\Theta}_1}(\cdot \mid \mathbf{S}_n) = \mathbb{P}_{\boldsymbol{\Theta}_2}(\cdot \mid \mathbf{S}_n) \quad \text{a.s.,} \tag{5}$$

which concludes the direct sense. The converse sense is proved by assuming that Eq. (5) holds and using the distance properties of the total variation. This concludes the proof. $\qquad\square$

# D PROOFS

## D.1 PROOF OF PROPOSITION 3.1

We detail below the proof of Proposition 3.1.

*Proof of Proposition 3.1.* **Step 1: large language models as Markov chains.** Given an input $v_i \in \mathcal{V}_K^*$ of $p$ tokens, an large language model outputs a probability mass function $f_{\boldsymbol{\Theta}}^{T,K}(v_i)$ over the discrete vocabulary space. As the temperature is positive, i.e., $\tau > 0$, and as the exponential is positive, we know that all the tokens in the vocabulary will be given a positive mass.

A next sequence $v_j \in \mathcal{V}_K^*$ is then sampled according to $f_{\boldsymbol{\Theta}}^{T,K}(v_i)$. But the $v_j$ sequences that fit necessarily contain the $v_i$ sequence (except possibly the first element of $v_i$, thanks to Definition B.2), i.e. $\forall l, (v_j)_l = (v_i)_{l+1}$. Note also the size of $v_j$ is $p + 1$ when $p < k$ and $k$ when $p = k$. All other sequences $v_j$ that do not satisfy this condition are not suitable.

In that sense, $f_{\boldsymbol{\Theta}}^{T,K}$ can be represented by a Markov chain $\mathrm{MC}(\mathcal{V}_K^*, \mathbf{Q}_f)$ with transition kernel $\mathbf{Q}_f \in \mathbb{R}^{|\mathcal{V}_K^*| \times |\mathcal{V}_K^*|}$, as defined in Proposition 3.1.

**Step 2: Proportion of non-zero elements.** We denote by $\mathscr{R}$ the set of states of length $K$. The set of states of length strictly less than equal $K$ is denoted by $\mathscr{T}$. We can construct a transition matrix $P_{\mathscr{R}} \in \mathbb{R}^{T^K \times T^K}$ with the states of this class, containing the probabilities of moving from one state of $\mathscr{R}$ to another. $P_{\mathscr{R}}$ corresponds to the blue block in Fig. 1 while green rectangle blocks correspond to part of $P_{\mathscr{T}}$ and $P_{\mathscr{T}\mathscr{R}}$ in the following description of large language models as Markov chains,

$$\mathbf{Q}_f = \left( \begin{array}{c|c} P_{\mathscr{T}} & P_{\mathscr{T}\mathscr{R}} \\ \hline 0 & P_{\mathscr{R}} \end{array} \right). \tag{6}$$

Now, let us count the number of non-zero elements in each of these $4$ large blocks.

$\underline{P_{\mathscr{T}} \text{ block}}$ : The size of this block is $\left[ \frac{T}{T-1}(T^{K-1} - 1) \right] \times \left[ \frac{T}{T-1}(T^{K-1} - 1) \right]$. There are $K - 2$ green blocks contained in $P_{\mathscr{T}}$. The block number $i \in [K-2]$ is of size $T^i \times T^{i+1}$. Since each

sentence of size $i$ can be completed with non-zero probability, by any other token, there are a total of $\sum_{p=1}^{T^i} T = T^{i+1}$ non-zero elements. There are therefore $\sum_{i=1}^{K-2} T^{i+1}$ non-zero elements in the entire $P_{\mathcal{T}}$ block.

$\underline{P_{\mathcal{T}\mathcal{R}}$ block :} The size of this block is $\left[\frac{T}{T-1}(T^{K-1} - 1)\right] \times T^K$. The green block contained in $P_{\mathcal{T}\mathcal{R}}$ that contains non-zero elements is of size $T^{K-1} \times T^K$. Since each sentence of size $K - 1$ can be completed with non-zero probability, by any other token, there are a total of $\sum_{p=1}^{T^{K-1}} T = T^K$ non-zero elements.

$\underline{P_{\mathcal{R}}$ block :} The size of this block is $T^K \times T^K$. Each sentence $v = (v_1, \ldots, v_K)$ of size $K$ is mapped to another sentence $v' = (v'_1, \ldots, v'_K)$ of size $K$ with non-zero probability, if and only if $v'_1 = v_2, v'_2 = v_3, \ldots, v'_{k-1} = v_K$. The final token $v'_K$ can by any other token in the vocabulary. It means that there are a total of $\sum_{p=1}^{T^K} T = T^{K+1}$ non-zero elements.

$\underline{0's$ block :} Obviously, there are no non-zero elements in this block.

Finally, there are

$$\sum_{i=1}^{K-2} T^{i+1} + T^K + T^{K+1} = \sum_{i=1}^{K} T^{i+1} = T^2 \left(\frac{T^K - 1}{T - 1}\right)$$

non-zero elements. This means that the proportion of non-zero elements in the matrix is exactly

$$\frac{T^2 \left(\frac{T^K - 1}{T-1}\right)}{\left(T\left(\frac{T^K - 1}{T-1}\right)\right)^2} = \frac{T - 1}{T^K - 1}.$$

Note that for large $T$ and $K$ we have that

$$\frac{T - 1}{T^K - 1} \sim \frac{1}{T^{K-1}}.$$

$\square$

### D.2 PROOF OF PROPOSITION 3.2

We begin with a preliminary lemma.

**Lemma D.1** (Powers of $\mathbf{Q}_f$ greater than $K$). *For any initial state $i$, the following hold:*

- $\forall k \geq K, \forall j \in \mathcal{T}, (\mathbf{Q}_f^k)_{i,j} = 0,$

- $\forall k \geq K, \forall j \in \mathcal{R}, (\mathbf{Q}_f^k)_{i,j} > 0.$

*Proof.* By considering $\mathbf{Q}_f$ as defined in (6), we can compute its powers. For any $k \geq 1$,

$$\mathbf{Q}_f^k = \left(\begin{array}{c|c} P_{\mathcal{T}}^k & B_k \\ \hline 0 & P_{\mathcal{R}}^k \end{array}\right),$$

where $B_k = \sum_{m=0}^{k-1} P_{\mathcal{T}}^m P_{\mathcal{T}\mathcal{R}} P_{\mathcal{R}}^{k-1-m}$.

To prove the first item, we focus on the blocks on the left of $\mathbf{Q}_f$. Since the lower left block is zero, we have that $\forall k \geq 1, \forall i \in \mathcal{R}, \forall j \in \mathcal{T}, (\mathbf{Q}_f^k)_{i,j} = 0$. In the upper left block, the element $(P_{\mathcal{T}}^k)_{i,j}$ designates the probability of moving from one transient state $i \in \mathcal{T}$ to another transient state $j \in \mathcal{T}$ after $k$ iterations. According to Definition B.1, if state $i$ is a sequence of $p \geq 1$ tokens, state $j$ is necessarily a sequence of $\min\{K, p + K\} = K$ elements. Thus, $P_{\mathcal{T}}$ is a nilpotent matrix and $\forall k \geq K, \forall i, j, (P_{\mathcal{T}}^k)_{i,j} = 0$. This proves that $\forall k \geq K, \forall j \in \mathcal{T}, (\mathbf{Q}_f^k)_{i,j} = 0$.

We now move on to the second item. From the above, $\forall k \geq K, B_k = \sum_{m=0}^{K-1} P_{\mathcal{T}}^m P_{\mathcal{T}\mathcal{R}} P_{\mathcal{R}}^{k-1-m}$. Note that this sum is finite, but there is still a dependence on $k$, in the powers of the matrix $P_{\mathcal{R}}$. In the lower right block, the element $(P_{\mathcal{R}}^k)_{i,j}$ designates the probability of moving from one recurrent state $i \in \mathcal{R}$ to another recurrent state $j \in \mathcal{R}$ after $k$ iterations. According to the definition of $\mathbf{Q}_f$ in Proposition 3.1 and Definition B.2, $\forall k \geq K, \forall i,j \in \mathcal{R}^2, (P_{\mathcal{R}}^k)_{i,j}$ is nonzero. Exploiting this also in $B_k$, we obtain the result, i.e. $\forall k \geq K, \forall j \in \mathcal{R}, (\mathbf{Q}_f^k)_{i,j} > 0$. □

We are now ready to prove Proposition 3.2, which is inspired by Gallager (1996).

*Proof of Proposition 3.2.* The states of length strictly less than equal $K$ (elements of $\mathcal{T}$) are transient, because of Definition B.1. To discuss the nature of states of length $K$ (elements of $\mathcal{R}$), let us introduce a result regarding the powers of the $\mathbf{Q}_f$ matrix as defined in (6). Thanks to Lemma D.1, the set of states $\mathcal{R}$ (i.e. the states of length $K$) form a class. Lemma D.1 gives us also that $\mathcal{R}$ is ergodic. In fact, every state in $\mathcal{R}$ only communicates with all the other states in $\mathcal{R}$, which proves the recurrence. Since $\forall i,j \in \mathcal{R}^2, (\mathbf{Q}_f^K)_{i,j} > 0$, we can move between any two states in exactly $K$ steps, regardless of the initial position. This ensures that $\mathcal{R}$ is aperiodic because the transition probabilities do not depend on a specific cycle, and states can be revisited at various time steps, not just multiples of a particular number. More simply, by considering a token $x$, the state defined as $i = \underbrace{xx \ldots x}_{K \text{ times}}$ has period 1, i.e. $(\mathbf{Q}_f)_{i,i} > 0$. This is a consequence of Definition B.2 and Proposition 3.1. Thanks to Theorem C.9, it means that the whole class $\mathcal{R}$ is aperiodic. Finally, this means that $\mathrm{MC}(\mathcal{V}_K^*, \mathbf{Q}_f)$ are ergodic unichains, in the sense of Definition C.11. □

### D.3 PROOF OF PROPOSITION 3.3

We start by introducing three technical lemmas that will be useful in the proof of Proposition 3.3. We start with the Chapman–Kolmogorov equation.

**Lemma D.2** (Chapman-Kolmogorov equation). *Let $P$ be a matrix of size $d$. Then, $P$ satisfies*

$$\forall i,j \in [d]^2, \forall n,n' \in \mathbb{N}^2, (P^{n+n'})_{i,j} = \sum_{k=1}^{d} (P^n)_{i,k}(P^{n'})_{k,j}.$$

*Proof.* The result follows from the fact that $\forall n,n' \in \mathbb{N}^2, P^{n+n'} = P^n P^{n'}$. □

Then, we introduce a simple but useful result of monotonicity.

**Lemma D.3** (Lemma 3.3.1. in Gallager (1996)). *Let the transition matrix $P$ of a finite state Markov chain. Then, for all states $j$ and $n \geq 1$, we have*

$$\max_i (P^{n+1})_{i,j} \leq \max_i (P^n)_{i,j}, \quad and \quad \min_i (P^{n+1})_{i,j} \geq \min_i (P^n)_{i,j}.$$

We now refer to a result on Markov chains with positive transition matrices.

**Lemma D.4** (Lemma 3.3.2. in Gallager (1996)). *Let the transition matrix $P$ of a finite state Markov chain satisfy $\forall i,j, P_{i,j} > 0$, and let $\alpha = \min_{i,j} P_{i,j} > 0$. Then, for all states $j$ and $n \geq 1$,*

$$\max_i (P^n)_{i,j} - \min_i (P^n)_{i,j} \leq (1-2\alpha)\Big(\max_i (P^n)_{i,j} - \min_i (P^n)_{i,j}\Big),$$

$$\max_i (P^n)_{i,j} - \min_i (P^n)_{i,j} \leq (1-2\alpha)^n,$$

$$\lim_{n\to\infty} \max_i (P^n)_{i,j} = \lim_{n\to\infty} \min_i (P^n)_{i,j} > 0.$$

We are now ready to prove Proposition 3.3 using a similar argument as in Gallager (1996).

*Proof of Proposition 3.3.* Let $\mathscr{T}$ and $\mathscr{R}$ denote respectively the sets of transient and recurrent states. For any state $j$, we define $\pi_j := \lim_{n\to\infty} \max_i (\mathbf{Q}_f^n)_{i,j} = \lim_{n\to\infty} \min_i (\mathbf{Q}_f^n)_{i,j}$. Then $\boldsymbol{\pi} = (\pi_j)_{j\in\Omega}$ is the stationary distribution for $\mathbf{Q}_f$.

**Step 1: Stationary distribution for transient states.** Lemma D.1 gives us that $\forall i, \forall k \geq K, \forall j \in \mathscr{T}, (\mathbf{Q}_f^k)_{i,j} = 0$. This means that $\forall j \in \mathscr{T}, \pi_j = 0$ and hence the limit is reached at most after $K$ iteration.

**Step 2: Stationary distribution for recurrent states.** Lemma D.1 gives us $\forall i, j \in \mathscr{R}^2, (\mathbf{Q}_f^K)_{i,j} > 0$. By defining $\varepsilon := \min_{i,j\in\mathscr{R}^2} (\mathbf{Q}_f^K)_{i,j}$, Lemma D.4 shows that for any integer $\ell \geq 1$,

$$\max_{i\in\mathscr{R}} (\mathbf{Q}_f^{\ell K})_{i,j} - \min_{i\in\mathscr{R}} (\mathbf{Q}_f^{\ell K})_{i,j} \leq (1-2\varepsilon)\left( \max_{i\in\mathscr{R}} (\mathbf{Q}_f^{\ell K})_{i,j} - \min_{i\in\mathscr{R}} (\mathbf{Q}_f^{\ell K})_{i,j} \right), \tag{7}$$

$$\max_{i\in\mathscr{R}} (\mathbf{Q}_f^{\ell K})_{i,j} - \min_{i\in\mathscr{R}} (\mathbf{Q}_f^{\ell K})_{i,j} \leq (1-2\varepsilon)^{\ell}, \tag{8}$$

$$\lim_{\ell\to\infty} \max_{i\in\mathscr{R}} (\mathbf{Q}_f^{\ell K})_{i,j} = \lim_{\ell\to\infty} \min_{i\in\mathscr{R}} (\mathbf{Q}_f^{\ell K})_{i,j} > 0. \tag{9}$$

Thanks to Lemma D.3, $\max_i (\mathbf{Q}_f^{n+1})_{i,j}$ is non-decreasing in $n$, so the limit on the left in Eq. (9) can be replaced with a limit in $n$. The same argument for the limit on the right gives that, $\forall j \in \mathscr{R}$,

$$\max_{i\in\mathscr{R}} (\mathbf{Q}_f^n)_{i,j} - \min_{i\in\mathscr{R}} (\mathbf{Q}_f^n)_{i,j} \leq (1-2\varepsilon)^{\lfloor n/K\rfloor},$$

$$\lim_{n\to\infty} \max_{i\in\mathscr{R}} (\mathbf{Q}_f^n)_{i,j} = \lim_{n\to\infty} \min_{i\in\mathscr{R}} (\mathbf{Q}_f^n)_{i,j} > 0,$$

where we have taken the floor function to also convert the result of (8). Since $\pi_j$ lies between the minimum and maximum $(\mathbf{Q}_f^n)_{i,j}$ for each $n$, we have that $\forall i, j \in \mathscr{R}^2$,

$$|(\mathbf{Q}_f^n)_{i,j} - \pi_j| \leq (1-2\varepsilon)^{\lfloor \frac{n}{K}\rfloor}.$$

It means that $\forall i, j \in \mathscr{R}^2, \pi_j = \lim_{n\to\infty}(\mathbf{Q}_f^n)_{i,j}$. This also gives us the convergence rate when the initial state $i$ is recurrent. In the next step, we consider the general convergence rate, regardless of the nature of the initial state $i$.

**Step 3: Convergence bound.** We proceed to the remaining case, i.e. the case where the initial state $i \in \mathscr{T}$ and the final state $j \in \mathscr{R}$. Lemma D.2 says that $\forall n \geq K$,

$$(\mathbf{Q}_f^n)_{i,j} = \sum_{k\in\mathscr{T}} (\mathbf{Q}_f^K)_{i,k}(\mathbf{Q}_f^{n-K})_{k,j} + \sum_{k\in\mathscr{R}} (\mathbf{Q}_f^K)_{i,k}(\mathbf{Q}_f^{n-K})_{k,j}.$$

We then have that $\forall i \in \mathscr{T}, \forall n \in \mathbb{N}$,

$$|(\mathbf{Q}_f^n)_{i,j} - \pi_j| \leq \Big| \sum_{k\in\mathscr{T}} (\mathbf{Q}_f^K)_{i,k}\big[(\mathbf{Q}_f^{n-K})_{k,j} - \pi_j\big] + \sum_{k\in\mathscr{R}} (\mathbf{Q}_f^K)_{i,k}\big[(\mathbf{Q}_f^{n-K})_{k,j} - \pi_j\big] \Big|$$

$$\leq \sum_{k\in\mathscr{T}} (\mathbf{Q}_f^K)_{i,k}\big|(\mathbf{Q}_f^{n-K})_{k,j} - \pi_j\big| + \sum_{k\in\mathscr{R}} (\mathbf{Q}_f^K)_{i,k}\big|(\mathbf{Q}_f^{n-K})_{k,j} - \pi_j\big|$$

$$\leq \sum_{k\in\mathscr{T}} (\mathbf{Q}_f^K)_{i,k} + \sum_{k\in\mathscr{R}} (\mathbf{Q}_f^K)_{i,k}\big|(\mathbf{Q}_f^{n-K})_{k,j} - \pi_j\big|$$

$$\leq (1-2\varepsilon)^{\lfloor \frac{n-K}{K}\rfloor},$$

where the first sum vanishes and $\sum_{k\in\mathscr{R}}(\mathbf{Q}_f^K)_{i,k} \leq 1$. Finally, $\forall i \in \mathscr{T}, \forall n \geq K$,

$$|(\mathbf{Q}_f^n)_{i,j} - \pi_j| \leq (1-2\varepsilon)^{\lfloor \frac{n}{K}\rfloor - 1}.$$

Combining this with the result of Step 2 concludes the proof. $\qquad\square$

### D.4 Proof of Theorem 4.1

In this section, we detail the proof of Theorem 4.1. We provide below an overview of the proof before detailing it.

**Overview of the proof.** We are going to use McDiarmid's inequality for dependent random variables of Paulin (2015, Theorem 2.9). To adapt the arguments of Paulin (2015, Theorem 2.9) to our setting, we bound the total variation between the true probability of the next token and the one estimated by the LLM. The rest of this section is organized as follows. First in Appendix D.4.1, we adapt the concentration inequality of Paulin (2015, Theorem 2.9). Then in Appendix D.4.2, we show how to bound the total variation between the true and the estimated probability of the next token. , in Appendix D.4.3, we restate Theorem 4.1 and conclude the proof.

### D.4.1 CONCENTRATION INEQUALITIES FOR DEPENDENT RANDOM VARIABLES

We first state a concentration inequality for dependent random variables that will be used to obtain our final bound.

**Proposition D.5** (McDiarmid's inequality for dependent random variables). *Let $S :=$ $(\mathbf{S}_1, \ldots, \mathbf{S}_N)$ be a sequence of random variables that take values in $\Omega = \Omega_1 \times \ldots \times \Omega_N$. Assume there exists a Marton coupling for $S$ with mixing matrix $\boldsymbol{\Gamma}$. Let $\|\boldsymbol{\Gamma}\|$ be the operator norm of $\boldsymbol{\Gamma}$. If $f \colon \Omega \to \mathbb{R}$ is such that there exists $\mathbf{c} \in \mathbb{R}^N$ satisfying*

$$\forall \mathbf{x}, \mathbf{y} \in \Omega, \quad f(\mathbf{x}) - f(\mathbf{y}) \leq \sum_{i=1}^{N} \mathbf{c}_i \mathbb{1}_{\{\mathbf{x}_i \neq \mathbf{y}_i\}},$$

*then we have for any $u \geq 0$,*

$$\mathbb{P}(|f(S) - \mathbb{E}_S[f(S)]| \geq u) \leq 2 \exp\left(\frac{-2u^2}{\|\boldsymbol{\Gamma}\|^2 \|\mathbf{c}\|_2^2}\right).$$

*Proof.* Consider a function $f$ verifying the properties of Proposition D.5. Paulin (2015, Theorem 2.9) ensures that for a partition $\hat{S}$ of $S$ (see Paulin, 2015, Definition 2.3) the following inequality holds

$$\forall u \geq 0, \quad \mathbb{P}\left(\left|f(\hat{S}) - \mathbb{E}\left[f(\hat{S})\right]\right| \geq u\right) \leq 2 \exp\left(\frac{-2u^2}{\|\boldsymbol{\Gamma} \cdot C(\mathbf{c})\|_2^2}\right), \tag{10}$$

where $C(\mathbf{c})$ is a vector of $\mathbb{R}^N$ whose $i$-th element is the sum of the $\mathbf{c}_j$ such that $j$ is an index of the elements of $\hat{\mathbf{S}}_i$. Taking the trivial partition $\hat{S} = S$ implies that the index of the elements in $\hat{\mathbf{S}}_i$ are reduced to $\{i\}$. Hence the $i$-th entry of $C(\mathbf{c})$ is equal to $\mathbf{c}_i$ and $C(\mathbf{c}) = \mathbf{c}$. By definition of the operator norm (naturally induced by the $\ell_2$-norm), we have

$$\|\boldsymbol{\Gamma} \cdot \mathbf{c}\|_2 = \frac{\|\boldsymbol{\Gamma}\mathbf{c}\|_2}{\|\mathbf{c}\|_2} \cdot \|\mathbf{c}\|_2 \leq \underbrace{\sup_{\mathbf{x} \neq 0} \frac{\|\boldsymbol{\Gamma}\mathbf{x}\|_2}{\|\mathbf{x}\|_2}}_{=\|\boldsymbol{\Gamma}\|} \cdot \|\mathbf{c}\|_2 \leq \|\boldsymbol{\Gamma}\| \cdot \|\mathbf{c}\|_2,$$

where the first inequality comes from the fact that $\mathbf{c}$ is non-zero (otherwise the only possible $f$ is the zero function which is not of great interest). Using the fact that the function $x \to \exp\left(-\frac{2u^2}{x}\right)$ is increasing, we obtain

$$\exp\left(\frac{-2u^2}{\|\boldsymbol{\Gamma} \cdot \mathbf{c}\|_2^2}\right) \leq \exp\left(\frac{-2u^2}{\|\boldsymbol{\Gamma}\|^2 \cdot \|\mathbf{c}\|_2^2}\right),$$

which concludes the proof. $\square$

By looking at the definition of the risk $\widehat{\mathcal{R}}_{\mathrm{pre}}(\boldsymbol{\Theta})$, we can see that applying Proposition D.5 to the function

$$f \colon (\mathbf{S}_1, \ldots, \mathbf{S}_{N_{\mathrm{train}}}) = \frac{1}{N_{\mathrm{train}}} \sum_{n=1}^{N_{\mathrm{train}}} d_{\mathrm{TV}}(\mathbb{P}_{\mathcal{L}}(\cdot \mid \mathbf{S}_n), \mathbb{P}_{\boldsymbol{\Theta}}(\cdot \mid \mathbf{S}_n)),$$

would lead to the desired bound as we already know $S$ admits a Marton coupling with mixing matrix $\boldsymbol{\Gamma}$. We investigate in the next section how to find the bounding vector $\mathbf{c}$ to apply Proposition D.5.

### D.4.2 FINDING THE BOUNDING VECTOR

**Technical lemmas.** We first recall the following important notions from (Tsybakov, 2008). Let $(\Omega, \mathcal{F})$ be a measure space and consider two probability distributions $\mathbb{P}, \mathbb{Q}$ defined on $(\Omega, \mathcal{F})$. For any $\sigma$-finite measure $\nu$ on $(\Omega, \mathcal{F})$ such that $\mathbb{P}, \mathbb{Q}$ are absolutely continuous with respect to $\nu$, we can define $p = \frac{d\mathbb{P}}{d\nu}, q = \frac{d\mathbb{Q}}{d\nu}$ which can also be written as $\mathbb{P}(d\omega) = q(\omega)\nu(d\omega)$ and $\mathbb{Q}(d\omega) = p(\omega)\nu(d\omega)$. We will adopt both notations interchangeably. It should be noted that there always exists at least one such measure $\nu$ as one can take $\nu = \mathbb{P} + \mathbb{Q}$. With these notations, the squared Hellinger distance between $\mathbb{P}$ and $\mathbb{Q}$ is defined as

$$H(\mathbb{P}, \mathbb{Q})^2 := \int_{\omega \in \Omega} \left( \sqrt{p(\omega)} - \sqrt{q(\omega)} \right)^2 \nu(d\omega) = \int_{\omega \in \Omega} \left( \sqrt{\mathbb{P}(d\omega)} - \sqrt{\mathbb{Q}(d\omega)} \right)^2.$$

The lemma below shows that the total variation between two probability distributions is controlled from above by the absolute value of the logarithm of their ratio.

> **Lemma D.6.** *Consider two probability distributions $\mathbb{P}, \mathbb{Q}$ defined on a measure space $(\Omega, \mathcal{F})$ and a $\sigma$-finite measure $\nu$ on $(\Omega, \mathcal{F})$. Let $p, q$ be the corresponding probabilities densities, i.e., we have $\mathbb{P}(d\omega) = q(\omega)\nu(d\omega)$ and $\mathbb{Q}(d\omega) = p(\omega)\nu(d\omega)$, the total variation between $\mathbb{P}$ and $\mathbb{Q}$ satisfies*
>
> $$d_{\mathrm{TV}}(\mathbb{P}, \mathbb{Q}) \leq \left( 2 \int_{\omega \in \Omega} \left| \log \sqrt{\frac{\mathbb{P}(d\omega)}{\mathbb{Q}(d\omega)}} \right| q(\omega) d\nu(d\omega) \right)^{1/2}.$$
>
> *If there exists a non-negative constant $B$ such that for any $z \in \Omega$, $\left| \log \sqrt{\frac{\mathbb{P}(z)}{\mathbb{Q}(z)}} \right| \leq B$, then we have*
>
> $$d_{\mathrm{TV}}(\mathbb{P}, \mathbb{Q}) \leq \sqrt{2B}.$$

*Proof.* We have the following relation between the total variation and the Hellinger distance (cf. Tsybakov, 2008, Lem. 2.3, Chapt. 2, p. 86):

$$d_{\mathrm{TV}}(\mathbb{P}, \mathbb{Q})^2 \leq H(\mathbb{P}, \mathbb{Q})^2 \cdot \left( 1 - \underbrace{H(\mathbb{P}, \mathbb{Q})^2/4}_{\geq 0} \right) \leq H(\mathbb{P}, \mathbb{Q})^2, \tag{11}$$

where the last inequality uses the positivity of the Hellinger distance. Inspired by the decomposition of the Hellinger distance in (Agarwal et al., 2020, Lem. 25), we have

$$H(\mathbb{P}, \mathbb{Q})^2 = \int_{\omega \in \Omega} \left( \sqrt{\mathbb{P}(d\omega)} - \sqrt{\mathbb{Q}(d\omega)} \right)^2 = \int_{\omega \in \Omega} \left( \mathbb{P}(d\omega) + \mathbb{Q}(d\omega) - 2\sqrt{\mathbb{P}(d\omega)}\sqrt{\mathbb{Q}(d\omega)} \right)$$

$$= 2 \cdot \left( 1 - \int_{\omega \in \Omega} \sqrt{\mathbb{P}(d\omega)}\sqrt{\mathbb{Q}(d\omega)} \right) = 2 \cdot \left( 1 - \int_{\omega \in \Omega} \sqrt{\frac{\mathbb{P}(d\omega)}{\mathbb{Q}(d\omega)}} \mathbb{Q}(d\omega) \right)$$

$$= 2 \cdot \left( 1 - \int_{\omega \in \Omega} \sqrt{\frac{\mathbb{P}(d\omega)}{\mathbb{Q}(d\omega)}} q(\omega) d\nu(d\omega) \right) \qquad \text{(by definition of } \mathbb{Q}(d\omega))$$

$$\leq -2 \log \left( \int_{\omega \in \Omega} \sqrt{\frac{\mathbb{P}(d\omega)}{\mathbb{Q}(d\omega)}} q(\omega) d\nu(d\omega) \right) \qquad \text{(using } 1 - x \leq -\log(x))$$

It follows using Eq. (11)

$$d_{\mathrm{TV}}(\mathbb{P}, \mathbb{Q})^2 \leq H(\mathbb{P}, \mathbb{Q})^2$$

$$\leq 2 \int_{\omega \in \Omega} -\log \left( \sqrt{\frac{\mathbb{P}(d\omega)}{\mathbb{Q}(d\omega)}} \right) q(\omega) d\nu(d\omega) \qquad \text{(by Jensen as } -\log \text{ is convex)}$$

$$\leq 2 \left| \int_{\omega \in \Omega} -\log \left( \sqrt{\frac{\mathbb{P}(d\omega)}{\mathbb{Q}(d\omega)}} \right) q(\omega) d\nu(d\omega) \right|$$

$$\leq 2 \int_{\omega \in \Omega} \left| -\log \left( \sqrt{\frac{\mathbb{P}(d\omega)}{\mathbb{Q}(d\omega)}} \right) \right| q(\omega) d\nu(d\omega) \qquad \text{(by Jensen as } |\cdot| \text{ is convex)}$$

$$\leq 2 \int_{\omega \in \Omega} \underbrace{\left| \log \left( \sqrt{\frac{\mathbb{P}(d\omega)}{\mathbb{Q}(d\omega)}} \right) \right|}_{\leq B} q(\omega) d\nu(d\omega) \qquad \text{(first part of Lemma D.6)}$$

$$\leq 2B \underbrace{\int_{\omega \in \Omega} q(\omega) d\nu(d\omega)}_{=1} \leq 2B. \qquad \text{(second part of Lemma D.6)}$$

This concludes both parts of the proof. $\qquad \square$

The next lemma provides a lower bound on the softmax output if its input is upper-bounded (in $\ell_1$-norm).

**Lemma D.7.** *Let $\mathbf{x} \in \mathbb{R}^m$ be such that $\|\mathbf{x}\|_1 \leq c_1$ for some $c_1 > 0$. Then, we have*

$$\text{softmax}(\mathbf{u}) \geq \frac{1}{m \exp{(2c_1)}},$$

*where the inequality holds for each component of* $\text{softmax}(\mathbf{u})$.

*Proof.* Using the fact that

$$\|\mathbf{x}\|_1 = \sum_{i=1}^{m} |\mathbf{x}_i| \leq c_1,$$

we know that for any $i \in [m]$, we have

$$-c_1 \leq \mathbf{x}_i \leq c_1.$$

Hence, using the fact that the exponential is increasing, we have for any $i \in [m]$

$$\exp{(-c_1)} \leq \exp{(\mathbf{x}_i)} \leq \exp{(c_1)}. \qquad (12)$$

Summing and taking the inverse leads to

$$\sum_{i=1}^{m} \exp{(-c_1)} \leq \sum_{i=1}^{m} \exp{(\mathbf{x}_j)} \leq \sum_{i=1}^{m} \exp{(c_1)}$$

$$\iff \frac{1}{\sum_{j=1}^{m} \exp{(c_1)}} \leq \frac{1}{\sum_{j=1}^{m} \exp{(\mathbf{x}_j)}} \leq \frac{1}{\sum_{j=1}^{m} \exp{(-c_1)}}. \qquad (13)$$

Combining Eq. (12) and Eq. (13) yields

$$\frac{\exp{(-c_1)}}{\sum_{j=1}^{m} \exp{(c_1)}} \leq \frac{\exp{(\mathbf{x}_i)}}{\sum_{j=1}^{m} \exp{(\mathbf{x}_j)}} \leq \frac{\exp{(c_1)}}{\sum_{j=1}^{m} \exp{(-c_1)}}.$$

As we desire a lower bound, we only focus on the left-hand side of the previous inequality. Multiplying the numerator and denominator by $\exp{(c_1)}$ leads to

$$\forall i \in [m], \quad \text{softmax}(\mathbf{x})_i = \frac{\exp{(\mathbf{x}_i)}}{\sum_{j=1}^{m} \exp{(\mathbf{x}_j)}} \geq \frac{1}{m \exp{(2c_1)}},$$

which concludes the proof. $\qquad \square$

**Upper-bounding the total variation.** We now proceed with finding an upper bound on the total variation between the true probability of the next token and the one estimated by the LLM $f_\Theta$. It will enable us to find the bounding vector $\mathbf{c}$. The next lemma shows that the input of the softmax layer of the model is bounded.

**Lemma D.8.** *Consider an LLM $f_\Theta \in \mathcal{F}$. For any input sequence $\mathbf{S} \in \mathbb{R}^{r \times n}$, the following inequality holds*

$$\|\frac{1}{n\tau}\mathbf{W}_U\mathbf{S}^{(L)}\mathbb{1}_n\|_1 \leq \frac{1}{\tau}\|\mathbf{W}_U^\top\|_{2,1},$$

*where $\tau$ is the temperature, $\mathbf{W}_U$ is the unembedding matrix (which is bounded as stated in the definition of the parameters space $\mathcal{W}$), and $\mathbf{S}^{(L)}$ is the output of the last transformer layer.*

*Proof.* We recall that the layer normalization ensures that at each layer, the tokens are in the unit $\ell_2$-ball. This is, in particular, the case for the output of the last layer $\mathbf{S}^{(L)}$. It means that the columns of $\mathbf{S}^{(L)}$ verifies

$$\forall k \in [n], \quad \|\mathbf{S}^{(L)}_{\cdot,k}\|_2 \leq 1, \tag{14}$$

which implies

$$\max_{1 \leq k \leq n} \|\mathbf{S}^{(L)}_{\cdot,i}\|_2 \leq 1. \tag{15}$$

Recalling that the $L_{p,q}$-norm of a matrix $\mathbf{A} \in \mathbb{R}^{n \times m}$ can be rewritten as

$$\|\mathbf{A}\|_{p,q} := \left(\sum_{j=1}^m \left(\sum_{i=1}^n |\mathbf{A}_{ij}|^p\right)^{\frac{q}{p}}\right)^{\frac{1}{q}} = \|(\|\mathbf{A}_{\cdot,j}\|_p)_{j=1}^m\|_q, \tag{16}$$

the $\ell_1$-norm of the last layer before the softmax layer satisfies

$$\|\frac{1}{n\tau}\mathbf{W}_U\mathbf{S}^{(L)}\mathbb{1}_n\|_1 = \frac{1}{n\tau}\sum_{i=1}^T \left|\sum_{j=1}^r \mathbf{W}_{ij}\sum_{k=1}^n \mathbf{S}_{jk}\right| = \frac{1}{n\tau}\sum_{i=1}^T \left|\sum_{j=1}^r \sum_{k=1}^n \mathbf{W}_{ij}\mathbf{S}_{jk}\right|$$

$$\leq \frac{1}{n\tau}\sum_{i=1}^T \sum_{j=1}^r \sum_{k=1}^n |\mathbf{W}_{ij}\mathbf{S}_{jk}| \qquad \text{(triangular inequality)}$$

$$\leq \frac{1}{n\tau}\sum_{i=1}^T \sum_{k=1}^n |\mathbf{W}_i^\top \mathbf{S}_{\cdot,k}|$$

$$\leq \frac{1}{n\tau}\sum_{i=1}^T \sum_{k=1}^n \|\mathbf{W}_i\|_2 \|\mathbf{S}_{\cdot,k}\|_2 \qquad \text{(Cauchy-Schwartz inequality)}$$

$$\leq \frac{1}{n\tau}\sum_{i=1}^T \sum_{k=1}^n \|\mathbf{W}_i\|_2 \max_{1 \leq k \leq n} \|\mathbf{S}_{\cdot,k}\|_2 \leq \frac{1}{n\tau}n \max_{1 \leq k \leq n} \|\mathbf{S}_{\cdot,k}\|_2 \sum_{i=1}^T \|\mathbf{W}_i\|_2$$

$$\leq \frac{1}{\tau}\sum_{i=1}^T \|\mathbf{W}_i\|_2 \leq \frac{1}{\tau}\|\mathbf{W}_U^\top\|_{2,1} \quad \text{(by Eq. (15) and the def. of $L_{2,1}$ in Eq. (16))}$$

where we dropped the subscript and superscript on $\mathbf{W}_U$ and $\mathbf{S}^{(L)}$ to ease the notations. This concludes the proof. $\qquad\square$

The previous lemma can be used to show that the logarithm of the ratio between the true probability of the next token and the one estimated by the LLM $f_\Theta$ is upper bounded as a function of the vocabulary size $T$, the temperature, the upper-bound on $\mathbf{W}_U$ and some constant related to the ambiguity of language (see Eq. (1)).

**Proposition D.9** (Upper-bound on the logarithm). *Consider an LLM $f_\Theta \in \mathcal{F}$ with vocabulary size $T$. We recall that $B_U$ is the upper bound on the norm of $\mathbf{W}_U$ in the definition of parameter space $\mathcal{W}$, $\tau$ is the softmax temperature and $c_0$ is the constant related to the ambiguity of language*

*(see Eq. (1)). We have*

$$\forall n \in [N], \quad \left| \log \left( \frac{\mathbb{P}_{\mathcal{L}}(\mathbf{X}_{n+1} \mid \mathbf{S}_n)}{\mathbb{P}_{\boldsymbol{\Theta}}(\mathbf{X}_{n+1} \mid \mathbf{S}_n)} \right) \right| \leq \bar{B} = \max\{\log(T) + \frac{2B_U}{\tau}, \log\left(\frac{1}{c_0}\right)\}.$$

*Proof.* The main idea of the proof is to bound the probability ratio and use the fact that $\log$ is non-decreasing. Let $n \in [N]$. The model $f_{\boldsymbol{\Theta}}$ receives as input sequences of tokens $\mathbf{S}_n$ of size $n \leq K$. We first lower-bound each term of the probability ratio. From Eq. (1), we have

$$\mathbb{P}_{\mathcal{L}}(\mathbf{X}_{n+1} \mid \mathbf{S}_n) \geq c_0. \tag{17}$$

We want to obtain a similar inequality for $\mathbb{P}_{\boldsymbol{\Theta}}(\mathbf{X}_{n+1} \mid \mathbf{S}_n)$. As the parameters $\boldsymbol{\Theta}$ of the LLM are in $\mathcal{W}$, we know that $\|\mathbf{W}_U^\top\|_{2,1} \leq B_U$. Lemma D.8 ensures that

$$\|\frac{1}{n\tau}\mathbf{W}_U\mathbf{S}^{(L)}\mathbb{1}_n\|_1 \leq \frac{1}{\tau}\|\mathbf{W}_U^\top\|_{2,1} \leq \frac{B_U}{\tau}.$$

We can then apply Lemma D.7 with $c_1 = \frac{B_U}{\tau}$ and given that $\frac{1}{n\tau}\mathbf{W}_U\mathbf{S}^{(L)}\mathbb{1}_n \in \mathbb{R}^T$, it leads to

$$\mathbb{P}_{\boldsymbol{\Theta}}(\cdot \mid \mathbf{S}_n) = \mathrm{softmax}\left(\frac{1}{n\tau}\mathbf{W}_U\mathbf{S}^{(L)}\mathbb{1}_n\right) \geq \frac{1}{T\exp(2B_U/\tau)},$$

where the inequality holds for each component of $\mathbb{P}_{\boldsymbol{\Theta}}(\cdot \mid \mathbf{S}_n)$. This is in particular the case for $\mathbb{P}_{\boldsymbol{\Theta}}(\mathbf{X}_{n+1} \mid \mathbf{S}_n)$ which is the entry we are interested in, i.e., we have

$$\mathbb{P}_{\boldsymbol{\Theta}}(\mathbf{X}_{n+1} \mid \mathbf{S}_n) \geq \frac{1}{T\exp(2B_U/\tau)}. \tag{18}$$

Going back to the ratio of probability, consider the situation where we have

$$\frac{\mathbb{P}_{\mathcal{L}}(\mathbf{X}_{n+1} \mid \mathbf{S}_n)}{\mathbb{P}_{\boldsymbol{\Theta}}(\mathbf{X}_{n+1} \mid \mathbf{S}_n)} \geq 1.$$

Then, using Eq. (18), we have

$$1 \leq \frac{\mathbb{P}_{\mathcal{L}}(\mathbf{X}_{n+1} \mid \mathbf{S}_n)}{\mathbb{P}_{\boldsymbol{\Theta}}(\mathbf{X}_{n+1} \mid \mathbf{S}_n)} \leq \frac{1}{\mathbb{P}_{\boldsymbol{\Theta}}(\mathbf{X}_{n+1} \mid \mathbf{S}_n)} \leq T\exp(2B_U/\tau),$$

which implies, as the $\log$ is non-decreasing monotonically,

$$0 \leq \log\left(\frac{\mathbb{P}_{\mathcal{L}}(\mathbf{X}_{n+1} \mid \mathbf{S}_n)}{\mathbb{P}_{\boldsymbol{\Theta}}(\mathbf{X}_{n+1} \mid \mathbf{S}_n)}\right) \leq \log(T\exp(2B_U/\tau)) = \log(T) + \frac{2B_U}{\tau}. \tag{19}$$

Similarly, consider the case where we have

$$\frac{\mathbb{P}_{\mathcal{L}}(\mathbf{X}_{n+1} \mid \mathbf{S}_n)}{\mathbb{P}_{\boldsymbol{\Theta}}(\mathbf{X}_{n+1} \mid \mathbf{S}_n)} \leq 1.$$

Then, we have

$$\frac{\mathbb{P}_{\boldsymbol{\Theta}}(\mathbf{X}_{n+1} \mid \mathbf{S}_n)}{\mathbb{P}_{\mathcal{L}}(\mathbf{X}_{n+1} \mid \mathbf{S}_n)} \geq 1,$$

and similarly to above, we can use Eq. (17) to obtain

$$1 \leq \frac{\mathbb{P}_{\boldsymbol{\Theta}}(\mathbf{X}_{n+1} \mid \mathbf{S}_n)}{\mathbb{P}_{\mathcal{L}}(\mathbf{X}_{n+1} \mid \mathbf{S}_n)} \leq \frac{1}{\mathbb{P}_{\mathcal{L}}(\mathbf{X}_{n+1} \mid \mathbf{S}_n)} \leq \frac{1}{c_0}.$$

This implies

$$0 \leq \log\left(\frac{\mathbb{P}_{\boldsymbol{\Theta}}(\mathbf{X}_{n+1} \mid \mathbf{S}_n)}{\mathbb{P}_{\mathcal{L}}(\mathbf{X}_{n+1} \mid \mathbf{S}_n)}\right) \leq \log\left(\frac{1}{c_0}\right),$$

which also rewrites

$$0 \leq -\log\left(\frac{\mathbb{P}_{\mathcal{L}}(\mathbf{X}_{n+1} \mid \mathbf{S}_n)}{\mathbb{P}_{\boldsymbol{\Theta}}(\mathbf{X}_{n+1} \mid \mathbf{S}_n)}\right) \leq \log\left(\frac{1}{c_0}\right). \tag{20}$$

By definition of the absolute value, combining Eq. (19) and Eq. (20) leads to

$$\left| \log \left( \frac{\mathbb{P}_{\mathcal{L}}(\mathbf{X}_{n+1} \mid \mathbf{S}_n)}{\mathbb{P}_{\boldsymbol{\Theta}}(\mathbf{X}_{n+1} \mid \mathbf{S}_n)} \right) \right| \leq \max\{\log(T) + \frac{2B_U}{\tau}, \log\left(\frac{1}{c_0}\right)\}.$$

This concludes the proof. $\qquad \square$

We are now ready to upper-bound the total variation.

**Corollary D.10** (Upper-bound on the total variation). *Consider an LLM $f_{\Theta} \in \mathcal{F}$ with vocabulary size $T$. We recall that $B_U$ is the upper bound on the norm of $\mathbf{W}_U$ in the definition of parameter space $\mathcal{W}$, $\tau$ is the softmax temperature and $c_0$ is the constant related to the ambiguity of language (see Eq. (1)). For $n \in [N]$, we have*

$$d_{\mathrm{TV}}(\mathbb{P}_{\mathcal{L}}(\cdot \mid \mathbf{S}_n), \mathbb{P}_{\Theta}(\cdot \mid \mathbf{S}_n)) \leq \sqrt{2\max\{\log(T) + \frac{2B_U}{\tau}, \log\left(\frac{1}{c_0}\right)\}} := c_2. \quad (21)$$

*Proof.* Using Proposition D.9, we can directly apply Lemma D.6 with $B = \max\{\log(T) + \frac{2B_U}{\tau}, \log\left(\frac{1}{c_0}\right)\}$ for any $n \in [N]$. This leads to

$$\forall n \in [N], \quad d_{\mathrm{TV}}(\mathbb{P}_{\mathcal{L}}(\cdot \mid \mathbf{S}_n), \mathbb{P}_{\Theta}(\cdot \mid \mathbf{S}_n)) \leq \sqrt{2\max\{\log(T) + \frac{2B_U}{\tau}, \log\left(\frac{1}{c_0}\right)\}}.$$

This concludes the proof. $\qquad\square$

### D.4.3 CONCLUDING THE PROOF

We are now ready to state our main result.

**Theorem D.11** (Restatement of Theorem 4.1). *Consider an LLM $f_{\Theta} \in \mathcal{F}$ with vocabulary size $T$. We denote by $\mathbf{\Gamma}$ the mixing matrix of the pretraining sequences of tokens $(\mathbf{S}_1, \ldots, \mathbf{S}_{N_{\mathrm{train}}})$. Let $\delta > 0$. Then, with probability at least $1 - \delta$,*

$$\mathcal{R}_{\mathrm{pre}}(\Theta) \leq \widehat{\mathcal{R}}_{\mathrm{pre}}(\Theta) + \frac{\bar{B}}{\sqrt{N_{\mathrm{train}}}}\sqrt{\log\left(\frac{2}{\delta}\right)},$$

*where $\bar{B}$ is a constant depending on the parameters of the problem. More precisely,*

$$\bar{B} = 2\|\mathbf{\Gamma}\|\sqrt{\max\{\log(T) + \frac{2B_U}{\tau}, \log\left(\frac{1}{c_0}\right)\}}.$$

*Proof of Theorem 4.1.* By definition of the risk, we have

$$\widehat{\mathcal{R}}_{\mathrm{pre}}(\Theta) = \frac{1}{N_{\mathrm{train}}}\sum_{n=1}^{N_{\mathrm{train}}} \underbrace{d_{\mathrm{TV}}(\mathbb{P}_{\mathcal{L}}(\cdot \mid \mathbf{S}_n), \mathbb{P}_{\Theta}(\cdot \mid \mathbf{S}_n))}_{=g_n(\mathbf{S}_n)} = \frac{1}{N_{\mathrm{train}}}\sum_{n=1}^{N_{\mathrm{train}}} g_n(\mathbf{S}_n)$$

$$= f(\mathbf{S}_1, \ldots, \mathbf{S}_{N_{\mathrm{train}}}) = f(S).$$

Using Corollary D.10, we know that

$$|g_n(\mathbf{S}_n)| \leq \sqrt{2\max\{\log(T) + \frac{2B_U}{\tau}, \log\left(\frac{1}{c_0}\right)\}} := c_2.$$

By definition, each sequence of tokens $\mathbf{S}_n$ takes its values in $\mathcal{V}^n$ (again by abuse of notation, $n = \min\{n, K\}$) and $S$ takes its values in $\mathcal{V}^1 \times \ldots \times \mathcal{V}^{N_{\mathrm{train}}}$. For any two sequences $\zeta, \Sigma$ with values in $\mathcal{V}^1 \times \ldots \times \mathcal{V}^{N_{\mathrm{train}}}$, we have

$$f(\zeta) - f(\Sigma) = \frac{1}{N_{\mathrm{train}}}\sum_{n=1}^{N_{\mathrm{train}}}\left(\underbrace{d_{\mathrm{TV}}(\mathbb{P}_{\mathcal{L}}(\cdot \mid \zeta_n), \mathbb{P}_{\Theta}(\cdot \mid \zeta_n))}_{=g_n(\zeta_n)} - \underbrace{d_{\mathrm{TV}}(\mathbb{P}_{\mathcal{L}}(\cdot \mid \Sigma_n), \mathbb{P}_{\Theta}(\cdot \mid \Sigma_n))}_{=g_n(\Sigma_n)}\right)$$

$$= \frac{1}{N_{\mathrm{train}}}\sum_{n=1}^{N_{\mathrm{train}}}(g_n(\zeta_n) - g_n(\Sigma_n))$$

$$= \frac{1}{N_{\text{train}}} \sum_{n=1}^{N_{\text{train}}} (g_n(\boldsymbol{\zeta}_n) - g_n(\boldsymbol{\Sigma}_n)) \mathbb{1}_{\{\boldsymbol{\zeta}_n \neq \boldsymbol{\Sigma}_n\}} \qquad \text{(removing the zero terms)}$$

$$\leq \left| \frac{1}{N_{\text{train}}} \sum_{n=1}^{N_{\text{train}}} (g_n(\boldsymbol{\zeta}_n) - g_n(\boldsymbol{\Sigma}_n)) \mathbb{1}_{\{\boldsymbol{\zeta}_n \neq \boldsymbol{\Sigma}_n\}} \right|$$

$$\leq \frac{1}{N_{\text{train}}} \sum_{n=1}^{N_{\text{train}}} \left| (g_n(\boldsymbol{\zeta}_n) - g_n(\boldsymbol{\Sigma}_n)) \mathbb{1}_{\{\boldsymbol{\zeta}_n \neq \boldsymbol{\Sigma}_n\}} \right|$$

$$\leq \frac{1}{N_{\text{train}}} \sum_{n=1}^{N_{\text{train}}} |g_n(\boldsymbol{\zeta}_n) - g_n(\boldsymbol{\Sigma}_n)| \, \mathbb{1}_{\{\boldsymbol{\zeta}_n \neq \boldsymbol{\Sigma}_n\}}$$

$$\leq \frac{1}{N_{\text{train}}} \sum_{n=1}^{N_{\text{train}}} \left( \underbrace{|g_n(\boldsymbol{\zeta}_n)|}_{\leq c_2} + \underbrace{|g_n(\boldsymbol{\Sigma}_n)|}_{\leq c_2} \right) \mathbb{1}_{\{\boldsymbol{\zeta}_n \neq \boldsymbol{\Sigma}_n\}} \qquad \text{(Corollary D.10)}$$

$$\leq \frac{1}{N_{\text{train}}} \sum_{n=1}^{N_{\text{train}}} 2c_2 \mathbb{1}_{\{\boldsymbol{\zeta}_n \neq \boldsymbol{\Sigma}_n\}} = \sum_{n=1}^{N_{\text{train}}} \left( \frac{2c_2}{N_{\text{train}}} \right) \mathbb{1}_{\{\boldsymbol{\zeta}_n \neq \boldsymbol{\Sigma}_n\}},$$

where $c_2 = \sqrt{2 \max\{\log(T) + \frac{2B_U}{\tau}, \log\left(\frac{1}{c_0}\right)\}}$. As $\zeta$ and $\Sigma$ were taken arbitrary, choosing $\mathbf{c} \in \mathbb{R}^{N_{\text{train}}}$ with all entries equal to $\frac{2c_2}{N_{\text{train}}}$ ensures that $f$ verifies the condition in Proposition D.5, i.e.,

$$\forall \zeta, \Sigma, \quad f(\zeta) - f(\Sigma) \leq \sum_{n=1}^{N_{\text{train}}} \mathbf{c}_n \mathbb{1}_{\{\boldsymbol{\zeta}_n \neq \boldsymbol{\Sigma}_n\}}.$$

We assumed in Section 4.1 that the sequences $\mathbf{S}_n$ were related via a Marton coupling with mixing matrix $\boldsymbol{\Gamma}$. Putting everything together, we can apply Proposition D.5 which leads to

$$\forall u \geq 0, \quad \mathbb{P}(|f(S) - \mathbb{E}_S[f(S)]| \geq u) \leq 2 \exp\left( \frac{-2u^2}{\|\boldsymbol{\Gamma}\|^2 \|\mathbf{c}\|_2^2} \right). \tag{22}$$

Let $u \geq 0$. We have the following events ordering

$$(\mathbb{E}_S[f(S)] - f(S) \geq u) \subseteq (\mathbb{E}_S[f(S)] - f(S) \geq u) \cup (f(S) - \mathbb{E}_S[f(S)] \geq u)$$
$$= (|f(S) - \mathbb{E}_S[f(S)]| \geq u).$$

Hence, as $u$ was taken arbitrary and using Eq. (22), we have

$$\forall u \geq 0, \quad \mathbb{P}(\mathbb{E}_S[f(S)] - f(S) \geq u) \leq 2 \exp\left( \frac{-2u^2}{\|\boldsymbol{\Gamma}\|^2 \|\mathbf{c}\|_2^2} \right).$$

We recall that by definition

$$f(S) = \widehat{\mathcal{R}}_{\text{pre}}(\boldsymbol{\Theta}) \text{ and } \mathcal{R}_{\text{pre}}(\boldsymbol{\Theta}) = \mathbb{E}_S\left[ \widehat{\mathcal{R}}_{\text{pre}}(\boldsymbol{\Theta}) \right].$$

Since the previous inequality holds for any $u \geq 0$, we can hence choose $u$ such that

$$\delta = 2 \exp\left( \frac{-2u^2}{\|\boldsymbol{\Gamma}\|^2 \|\mathbf{c}\|_2^2} \right) \iff \frac{-2u^2}{\|\boldsymbol{\Gamma}\|^2 \|\mathbf{c}\|_2^2} = \log\left( \frac{\delta}{2} \right) \iff u^2 = \frac{1}{2} \|\boldsymbol{\Gamma}\|^2 \|\mathbf{c}\|_2^2 \log\left( \frac{2}{\delta} \right)$$

$$\iff u = \frac{1}{\sqrt{2}} \|\boldsymbol{\Gamma}\| \|\mathbf{c}\|_2 \sqrt{\log\left( \frac{2}{\delta} \right)}.$$

Using the fact that

$$\|\mathbf{c}\|_2 = \sqrt{\sum_{n=1}^{N_{\text{train}}} \mathbf{c}_n^2} = \sqrt{\sum_{n=1}^{N_{\text{train}}} \left( \frac{2c_2}{N_{\text{train}}} \right)^2} = \sqrt{\sum_{n=1}^{N_{\text{train}}} \frac{4c_2^2}{N_{\text{train}}^2}} = \sqrt{\frac{4c_2^2}{N_{\text{train}}}} = \frac{2c_2}{\sqrt{N_{\text{train}}}}$$

and using the fact that $c_2 = \sqrt{2\max\{\log{(T)} + \frac{2B_U}{\tau}, \log{\left(\frac{1}{c_0}\right)}\}}$ from Corollary D.10, we obtain

$$u = \frac{1}{\sqrt{2}}\frac{2c_2}{\sqrt{N_{\text{train}}}}\|\mathbf{\Gamma}\|\sqrt{\log{\left(\frac{2}{\delta}\right)}} = \frac{\sqrt{2}c_2}{\sqrt{N_{\text{train}}}}\|\mathbf{\Gamma}\|\sqrt{\log{\left(\frac{2}{\delta}\right)}}$$

$$= \frac{2\|\mathbf{\Gamma}\|\sqrt{\max\{\log{(T)} + \frac{2B_U}{\tau}, \log{\left(\frac{1}{c_0}\right)}\}}}{\sqrt{N_{\text{train}}}}\sqrt{\log{\left(\frac{2}{\delta}\right)}} = \frac{\bar{B}}{\sqrt{N_{\text{train}}}}\sqrt{\log{\left(\frac{2}{\delta}\right)}},$$

where we define

$$\bar{B} = 2\|\mathbf{\Gamma}\|\sqrt{\max\{\log{(T)} + \frac{2B_U}{\tau}, \log{\left(\frac{1}{c_0}\right)}\}}.$$

Putting everything together, we have

$$\mathbb{P}\left(\mathcal{R}_{\text{pre}}(\mathbf{\Theta}) - \widehat{\mathcal{R}}_{\text{pre}}(\mathbf{\Theta}) \geq \frac{\bar{B}}{\sqrt{N_{\text{train}}}}\sqrt{\log{\left(\frac{2}{\delta}\right)}}\right) \leq \delta.$$

Taking the opposite event leads to the following inequality with probability at least $1 - \delta$

$$\mathcal{R}_{\text{pre}}(\mathbf{\Theta}) \leq \widehat{\mathcal{R}}_{\text{pre}}(\mathbf{\Theta}) + \frac{\bar{B}}{\sqrt{N_{\text{train}}}}\sqrt{\log{\left(\frac{2}{\delta}\right)}},$$

which concludes the proof. □

### D.5 PROOF OF COROLLARY 4.2

As the layer norm is not applied anymore, each token is no longer in the $\ell_2$-unit ball, and Lemma D.8 does not hold anymore. We want to provide an analogous lemma for our setting. We first prove the following technical lemmas.

**Lemma D.12.** *The* ReLU *is a norm-decreasing operator, i.e., we have*

$$\forall \mathbf{A} \in \mathbb{R}^{n \times m}, \quad \|\text{ReLU}(\mathbf{A})\|_{1,1} \leq \|\mathbf{A}\|_{1,1},$$

*where the* ReLU *is applied entry-wise.*

*Proof.* Recalling that $\text{ReLU}(x) = \max\{0, x\}$ is applied entry-wise, using the fact that $|\max\{0, x\}| \leq |x|$ and considering $\mathbf{A}$ and $\tilde{\mathbf{A}} = \text{ReLU}(\mathbf{A})$, we have

$$\|\tilde{\mathbf{A}}\|_{1,1} = \sum_{i,j}|\tilde{\mathbf{A}}_{i,j}| = \sum_{i,j}|\max\{0, \tilde{\mathbf{A}}_{i,j}\}| \leq \sum_{i,j}|\mathbf{A}_{i,j}| \leq \|\mathbf{A}\|_{1,1},$$

which concludes the proof. □

**Lemma D.13.** *The $L_{1,1}$-norm verifies the following property:*

$$\forall \mathbf{A} \in \mathbb{R}^{n \times m}, \mathbf{B} \in \mathbb{R}^{m \times p}, \quad \|\mathbf{A}\mathbf{B}\|_{1,1} \leq n\|\mathbf{A}\|_{\infty}\|\mathbf{B}\|_{1,1}.$$

*Proof.* We have

$$\|\mathbf{A}\mathbf{B}\|_{1,1} = \sum_{j=1}^{p}\sum_{i=1}^{n}|(\mathbf{A}\mathbf{B})_{ij}| = \sum_{j=1}^{p}\sum_{i=1}^{n}|\sum_{k=1}^{m}\mathbf{A}_{ik}\mathbf{B}_{kj}| \leq \sum_{j=1}^{p}\sum_{i=1}^{n}\sum_{k=1}^{m}|\mathbf{A}_{ik}\mathbf{B}_{kj}|$$

$$\leq \sum_{j=1}^{p}\sum_{i=1}^{n}\sum_{k=1}^{m}|\mathbf{A}_{ik}||\mathbf{B}_{kj}| \leq \max_{ik}|\mathbf{A}_{ik}|\sum_{j=1}^{p}\sum_{i=1}^{n}\sum_{k=1}^{m}|\mathbf{B}_{kj}|$$

$$\leq n\|\mathbf{A}\|_{\infty}\sum_{j=1}^{p}\sum_{k=1}^{m}|\mathbf{B}_{kj}| \leq n\|\mathbf{A}\|_{\infty}\|\mathbf{B}\|_{1,1},$$

which concludes the proof. $\qquad\square$

**Lemma D.14.** *The $L_{2,1}$ and $L_{\infty,1}$-norms verify the following relation*

$$\forall \mathbf{A} \in \mathbb{R}^{n\times m}, \quad \|\mathbf{A}\|_{\infty,1} \leq \|\mathbf{A}\|_{2,1}.$$

*Proof.* By definition of the $L_{p,q}$-norm, we have

$$\|\mathbf{A}\|_{\infty,1} = \sum_{j=1}^{M} \max_{1\leq i\leq n}|\mathbf{A}_{ij}| = \sum_{j=1}^{M} \sqrt{\max_{1\leq i\leq n}|\mathbf{A}_{ij}^2|} \qquad \text{(as } x \to x^2 \text{ is increasing)}$$

$$\leq \sum_{j=1}^{M} \sqrt{\sum_{i=1}^{n}|\mathbf{A}_{ij}^2|} \leq \sum_{j=1}^{M}\|\mathbf{A}_{\cdot,j}\|_2 \leq \|\mathbf{A}\|_{2,1},$$

where the first inequality comes from adding non-negative terms. $\qquad\square$

We are now ready to state the lemma analogous to Lemma D.8.

**Lemma D.15.** *Consider an LLM $f_{\boldsymbol{\Theta}} \in \tilde{F}$ with $L$ layers. For any input sequence $\mathbf{S} \in \mathbb{R}^{r\times n}$, the following inequality holds*

$$\|\frac{1}{n\tau}\mathbf{W}_U\mathbf{S}^{(L)}\mathbb{1}_n\|_1 \leq \frac{c_3}{\tau}\|\mathbf{W}_U^{\top}\|_{2,1},$$

*where $\tau$ is the temperature and $c_3$ is a constant depending on the parameters upper-bound. More precisely,*

$$c_3 = \left[(1 + rmB_1B_2) \cdot \left(1 + \frac{r^3}{H}B_OB_V\right)\right]^L \cdot B_{\text{tok}}.$$

*$\mathbf{W}_U$ is the unembedding matrix (which is bounded as stated in the definition of the parameters space $\mathcal{W}$), and $\mathbf{S}^{(L)}$ is the output of the last transformer layer.*

*Proof of Lemma D.15.* Our model $f_{\boldsymbol{\Theta}} \in \tilde{\mathcal{F}}$ is given as input a sequence $\mathbf{S} \in \mathbb{R}^{r\times n}$. With similar computations than in Lemma D.8, we have

$$\frac{1}{n\tau}\|\mathbf{W}_U\mathbf{S}^{(L)}\mathbb{1}_n\|_1 = \frac{1}{n\tau}\sum_{i=1}^{T}\left|\sum_{j=1}^{r}\mathbf{W}_{ij}\sum_{k=1}^{n}\mathbf{S}_{jk}\right| = \frac{1}{n\tau}\sum_{i=1}^{T}\left|\sum_{j=1}^{r}\sum_{k=1}^{n}\mathbf{W}_{ij}\mathbf{S}_{jk}\right|$$

$$\leq \frac{1}{n\tau}\sum_{i=1}^{T}\sum_{j=1}^{r}\sum_{k=1}^{n}|\mathbf{W}_{ij}\mathbf{S}_{jk}| \qquad \text{(triangular inequality)}$$

$$\leq \frac{1}{n\tau}\sum_{i=1}^{T}\sum_{k=1}^{n}|\mathbf{W}_i^{\top}\mathbf{S}_{\cdot,k}| \leq \frac{1}{n\tau}\sum_{i=1}^{T}\sum_{k=1}^{n}\|\mathbf{W}_i\|_{\infty}\|\mathbf{S}_{\cdot,k}\|_1 \quad \text{(Hölder inequality)}$$

$$\leq \frac{1}{n\tau}\left(\sum_{i=1}^{T}\|\mathbf{W}_i\|_{\infty}\right) \cdot \left(\sum_{k=1}^{n}\|\mathbf{S}_{\cdot,k}\|_1\right) \leq \frac{1}{n\tau}\|\mathbf{W}_U^{\top}\|_{\infty,1}\|\mathbf{S}^{(L)}\|_{1,1}$$

$$\leq \frac{1}{n\tau}\|\mathbf{W}_U^{\top}\|_{2,1}\|\mathbf{S}^{(L)}\|_{1,1}, \qquad \text{(Lemma D.14)}$$

where, again, we dropped the subscript and superscript on $\mathbf{W}_U$ and $\mathbf{S}^{(L)}$ to ease the notations. We obtain

$$\|\frac{1}{n\tau}\mathbf{W}_U\mathbf{S}^{(L)}\mathbb{1}_n\|_1 \leq \frac{1}{n\tau}\|\mathbf{W}_U^{\top}\|_{2,1}\|\mathbf{S}^{(L)}\|_{1,1}. \qquad (23)$$

As we do not use layer normalization, we want to find another way to bound $\mathbf{S}^{(L)}$. To that end, we will first express $\mathbf{S}^{(\ell)}$, the output of the $(\ell)$-th layer of the transformer, as a function of $\mathbf{S}^{(\ell-1)}$, the output of the $(\ell-1)$-th layer. Using the definition of the transformer model (see Appendix B), we have

$$
\begin{cases}
\mathbf{Z}^{(\ell)} = \mathbf{S}^{(\ell-1)} + \mathcal{A}\Big(\mathbf{S}^{(\ell-1)}; \mathbf{W}_Q^{(\ell)}, \mathbf{W}_K^{(\ell)}, \mathbf{W}_V^{(\ell)}, \mathbf{W}_O^{(\ell)}\Big), \\
\mathbf{Y}^{(\ell)} = \mathbf{W}_2^{(\ell)} \mathrm{ReLU}\Big(\mathbf{W}_1^{(\ell)} \mathbf{Z}^{(\ell)}\Big), \\
\mathbf{S}^{(\ell)} = \mathbf{Z}^{(\ell)} + \mathbf{Y}^{(\ell)}.
\end{cases}
$$

We will compute each layer's $L_{1,1}$-norm.

**Step 1: MHA.** By definition, denoting the number of heads by $H$, we know that $\mathcal{A}\Big(\mathbf{S}^{(\ell-1)}; \mathbf{W}_Q^{(\ell)}, \mathbf{W}_K^{(\ell)}, \mathbf{W}_V^{(\ell)}, \mathbf{W}_O^{(\ell)}\Big) \in \mathbb{R}^{r \times n}$ multiplies $\mathbf{W}^{(\ell)} \in \mathbb{R}^{r \times r}$ with the concatenation on the rows of the $H$ softmax layers that each writes

$$
\mathrm{softmax}\Big(\mathbf{W}_Q^{(\ell)} \mathbf{S}^{(\ell)} \Big(\mathbf{W}_K^{(\ell)} \mathbf{S}^{(\ell-1)}\Big)^\top / \sqrt{r}\Big)\Big(\mathbf{W}_V^{(\ell)} \mathbf{S}^{(\ell-1)}\Big) \in \mathbb{R}^{\frac{r}{H} \times n},
$$

We keep the notations $\ell$ without explicating the index of the head to ease notations. Denoting the concatenation on the rows by $\mathbf{C}^{(\ell)} \in \mathbb{R}^{r \times n}$, we have

$$
\|\mathcal{A}\Big(\mathbf{S}^{(\ell-1)}; \mathbf{W}_Q^{(\ell)}, \mathbf{W}_K^{(\ell)}, \mathbf{W}_V^{(\ell)}, \mathbf{W}_O^{(\ell)}\Big)\|_{1,1} = \|\mathbf{W}_O^{(\ell)} \mathbf{C}^{(\ell)}\|_{1,1} \le r \cdot \|\mathbf{W}_O^{(\ell)}\|_\infty \|\mathbf{C}^{(\ell)}\|_{1,1}
$$

$$
\le r B_O \|\mathbf{C}^{(\ell)}\|_{1,1}. \qquad \text{(definition of } \tilde{\mathcal{W}}\text{)}
$$

Moreover, by definition of $\mathbf{C}^{(\ell)}$, we have

$$
\|\mathbf{C}^{(\ell)}\|_{1,1} = \sum_{j=1}^{r} \sum_{i=1}^{r} |\mathbf{C}_{ij}^{(\ell)}| = \sum_{j=1}^{r} \sum_{i=1}^{r/H} \sum_{h=1}^{H} |\mathbf{C}_{ij}^{(\ell,h)}| = \sum_{h=1}^{H} \|\mathbf{C}^{(\ell,h)}\|_{1,1}, \tag{24}
$$

where $\mathbf{C}^{(\ell,h)} \in \mathbb{R}^{\frac{r}{H} \times n}$ is the softmax matrix of the $h$-th layer. We recall that the softmax matrix is a row-stochastic matrix of $\mathbb{R}^{\frac{r}{H} \times r}$ so it has all values lower than 1. In the next computations, we drop the $h$ index on the query, key, and value matrices to ease the notations. Using Lemma D.13 on the softmax matrix and on the value matrix $\mathbf{W}_V^{(\ell)} \in \mathbb{R}^{\frac{r}{H} \times r}$, we have

$$
\|\mathbf{C}^{(\ell,h)}\|_{1,1} = \|\mathrm{softmax}\Big(\mathbf{W}_Q^{(\ell)} \mathbf{S}^{(\ell)} \Big(\mathbf{W}_K^{(\ell)} \mathbf{S}^{(\ell-1)}\Big)^\top / \sqrt{r}\Big)\Big(\mathbf{W}_V^{(\ell)} \mathbf{S}^{(\ell-1)}\Big)\|_{1,1}
$$

$$
\le \frac{r}{H} \cdot \|\mathrm{softmax}\Big(\mathbf{W}_Q^{(\ell)} \mathbf{S}^{(\ell)} \Big(\mathbf{W}_K^{(\ell)} \mathbf{S}^{(\ell-1)}\Big)^\top / \sqrt{r}\Big)\|_\infty \cdot \|\Big(\mathbf{W}_V^{(\ell)} \mathbf{S}^{(\ell-1)}\Big)\|_{1,1}
$$

$$
\le \frac{r}{H} \cdot \|\Big(\mathbf{W}_V^{(\ell)} \mathbf{S}^{(\ell-1)}\Big)\|_{1,1} \qquad \text{(the softmax matrix is row-stochastic)}
$$

$$
\le \frac{r}{H} \cdot \frac{r}{H} \|(\mathbf{W}_V^{(\ell)}\|_\infty \|\mathbf{S}^{(\ell-1)}\|_{1,1} \le \Big(\frac{r}{H}\Big)^2 B_V \|\mathbf{S}^{(\ell-1)}\|_{1,1}. \qquad \text{(definition of } \tilde{\mathcal{W}}\text{)}
$$

Combining the previous inequality with Eq. (24) leads to

$$
\|\mathbf{C}^{(\ell)}\|_{1,1} \le \frac{r^2}{H} B_V \|\mathbf{S}^{(\ell-1)}\|_{1,1}.
$$

In the end, the multi-head attention norm verifies

$$
\|\mathcal{A}\Big(\mathbf{S}^{(\ell-1)}; \mathbf{W}_Q^{(\ell)}, \mathbf{W}_K^{(\ell)}, \mathbf{W}_V^{(\ell)}, \mathbf{W}_O^{(\ell)}\Big)\|_{1,1} \le \frac{r^3}{H} B_O B_V \|\mathbf{S}^{(\ell-1)}\|_{1,1}.
$$

Using the triangular inequality, we obtain

$$
\|\mathbf{Z}^{(\ell)}\|_{1,1} \le \Big(1 + \frac{r^3}{H} B_O B_V\Big) \cdot \|\mathbf{S}^{(\ell-1)}\|_{1,1}. \tag{25}
$$

**Step 2: FF.** We recall that $\mathbf{W}_1 \in \mathbb{R}^{m \times r}$ and $\mathbf{W}_2 \in \mathbb{R}^{r \times m}$. Using similar arguments to the above, we have

$$
\|\mathbf{Y}^{(\ell)}\|_{1,1} = \|\mathbf{W}_2^{(\ell)} \mathrm{ReLU}\Big(\mathbf{W}_1^{(\ell)} \mathbf{Z}^{(\ell)}\Big)\|_{1,1}
$$

$$\leq r \cdot \|\mathbf{W}_2^{(\ell)}\|_\infty \|\mathrm{ReLU}\left(\mathbf{W}_1^{(\ell)}\mathbf{Z}^{(\ell)}\right)\|_{1,1} \qquad \text{(Lemma D.13)}$$

$$\leq r \cdot \|\mathbf{W}_2^{(\ell)}\|_\infty \|\mathbf{W}_1^{(\ell)}\mathbf{Z}^{(\ell)}\|_{1,1} \qquad \text{(Lemma D.12)}$$

$$\leq r \cdot m \cdot \|\mathbf{W}_2^{(\ell)}\|_\infty \|\mathbf{W}_1^{(\ell)}\|_\infty \|\mathbf{Z}^{(\ell)}\|_{1,1} \qquad \text{(Lemma D.13)}$$

$$\leq rmB_1B_2\|\mathbf{Z}^{(\ell)}\|_{1,1}. \qquad \text{(definition of } \tilde{\mathcal{W}})$$

**Step 3: output layer.** Again, applying the triangular inequality and using the previous inequality and Eq. (25), we have

$$\|\mathbf{S}^{(\ell)}\|_{1,1} \leq \|\mathbf{Z}^{(\ell)}\|_{1,1} + \|\mathbf{Y}^{(\ell)}\|_{1,1} \leq (1 + rmB_1B_2)\|\mathbf{Z}^{(\ell)}\|_{1,1}$$

$$\leq (1 + rmB_1B_2)\left(1 + \frac{r^3}{H}B_OB_V\right)\|\mathbf{S}^{(\ell-1)}\|_{1,1}.$$

Iterating through the layers, recalling that $\mathbf{S}^{(0)} = \mathbf{S}$, we finally obtain

$$\|\mathbf{S}^{(L)}\|_{1,1} \leq \left[(1 + rmB_1B_2)\left(1 + \frac{r^3}{H}B_OB_V\right)\right]^L \|\mathbf{S}\|_{1,1},$$

where $\mathbf{S}$ is the input sequence. Combining this inequality with Eq. (23) leads to

$$\|\frac{1}{n\tau}\mathbf{W}_U\mathbf{S}^{(L)}\mathbb{1}_n\|_1 \leq \left[(1 + rmB_1B_2)\left(1 + \frac{r^3}{H}B_OB_V\right)\right]^L \frac{\|\mathbf{S}\|_{1,1}}{n}\left(\frac{1}{\tau}\|\mathbf{W}_U^\top\|_{2,1}\right).$$

Using the fact that each token has a $\ell_1$-norm bounded by $B_{\text{tok}}$. Hence, each column of $\mathbf{S}$ is too and we have

$$\frac{1}{n}\|\mathbf{S}\|_{1,1} = \frac{1}{n}\sum_{j=1}^n\sum_{i=1}^r|\mathbf{S}_{ij}| = \frac{1}{n}\sum_{j=1}^n \underbrace{\|\mathbf{S}_{\cdot,j}\|_1}_{\leq B_{\text{tok}}} \leq B_{\text{tok}}.$$

Combining the last two inequalities concludes the proof. $\qquad\square$

We can now restate Corollary 4.2.

**Corollary D.16** (Restatement of Corollary 4.2)**.** *Consider an LLM $f_\Theta \in \tilde{\mathcal{F}}$ with vocabulary size $T$ composed of $L$ transformer blocks and $H$ attention heads. We denote by $\Gamma$ the mixing matrix of the pretraining sequences of tokens $(\mathbf{S}_1, \ldots, \mathbf{S}_{N_{\text{train}}})$. Let $\delta > 0$. Then, with probability at least $1 - \delta$,*

$$\mathcal{R}_{\text{pre}}(\Theta) \leq \widehat{\mathcal{R}}_{\text{pre}}(\Theta) + \frac{\bar{B}}{\sqrt{N_{\text{train}}}}\sqrt{\log\left(\frac{2}{\delta}\right)},$$

*where $\bar{B}$ is a constant depending on the parameters of the problem. More precisely,*

$$\bar{B} = 2\|\Gamma\|\sqrt{\max\{\log(T) + \frac{2(B_\Theta)^L}{\tau}, \log\left(\frac{1}{c_0}\right)\}},$$

*with $B_\Theta = \left[(1 + rmB_1B_2)\left(1 + \frac{r^3}{H}B_OB_V\right)\right](B_{\text{tok}}B_U)^{1/L}$.*

*Proof of Corollary 4.2.* We first note that the only change from Lemma D.8 to Lemma D.15 is the multiplicative constant $c_3 = \left[(1 + rmB_1B_2)\left(1 + \frac{r^3}{H}B_OB_V\right)\right]^L B_{\text{tok}}$ in front of $\frac{1}{\tau}\|\mathbf{W}_U^\top\|_{2,1}$. In particular, as we know that $\tilde{\mathcal{W}} \subset \mathcal{W}$, we also have $\|\mathbf{W}_U^\top\|_{2,1} \leq B_U$. Hence, we can apply the proof of Theorem 4.1 in a straightforward manner by changing $\frac{B_U}{\tau}$ by $c_3 \cdot \frac{B_U}{\tau}$. This concludes the proof. $\qquad\square$

### D.6 PROOF OF COROLLARY 4.3

We detail the proof of Corollary 4.3 below.

*Proof.* We first note that by definition of the total variation distance (Wolfer & Kontorovich, 2019), we have

$$\mathbb{E}_{\mathbf{S}\sim\mathbb{P}_{\mathcal{L}}}\|\mathbf{Q}^*(\mathbf{S},\cdot) - \mathbf{Q}_f(\mathbf{S},\cdot)\|_1 = \mathbb{E}_{\mathbf{S}\sim\mathbb{P}_{\mathcal{L}}}[2\cdot d_{\mathrm{TV}}(\mathbf{Q}^*(\mathbf{S},\cdot), \mathbf{Q}_f(\mathbf{S},\cdot))]$$
$$= 2\cdot\mathbb{E}_{\mathbf{S}\sim\mathbb{P}_{\mathcal{L}}}[d_{\mathrm{TV}}(\mathbf{Q}^*(\mathbf{S},\cdot), \mathbf{Q}_f(\mathbf{S},\cdot))]$$
$$= 2\cdot\mathcal{R}_{\mathrm{pre}}(\mathbf{\Theta}). \qquad \text{(by definition of the risk Eq. (2))}$$

Applying Theorem 4.1 (or similarly Corollary 4.2), we know that

$$\mathcal{R}_{\mathrm{pre}}(\mathbf{\Theta}) \leq \widehat{\mathcal{R}}_{\mathrm{pre}}(\mathbf{\Theta}) + \frac{\bar{B}}{\sqrt{N_{\mathrm{train}}}}\sqrt{\log\left(\frac{2}{\delta}\right)},$$

where $\bar{B}$ is formally defined in Theorem 4.1 (respectively Corollary 4.2). Assuming a perfect pre-training error amounts to consider $\widehat{\mathcal{R}}_{\mathrm{pre}}(\mathbf{\Theta}) = 0$. We denote by $N^*$ the integer such that the error is equal to $\frac{\epsilon}{2}$, i.e.,

$$\frac{\bar{B}}{\sqrt{N^*}}\sqrt{\log\left(\frac{2}{\delta}\right)} = \frac{\epsilon}{2} \iff \frac{\bar{B}^2}{N^*}\log\left(\frac{2}{\delta}\right) = \frac{\epsilon^2}{4} \iff N^* = \left(\frac{2\bar{B}}{\epsilon}\right)^2\log\left(\frac{2}{\delta}\right).$$

Taking the ceiling function ensures that $N^*$ is an integer. Hence, taking $N_{\mathrm{train}} \geq N^* = \lceil\left(\frac{2\bar{B}}{\epsilon}\right)^2\log\left(\frac{2}{\delta}\right)\rceil$ ensures that

$$\frac{\bar{B}}{\sqrt{N_{\mathrm{train}}}}\sqrt{\log\left(\frac{2}{\delta}\right)} \leq \frac{\bar{B}}{\sqrt{N^*}}\sqrt{\log\left(\frac{2}{\delta}\right)} = \frac{\epsilon}{2}.$$

Putting everything together, taking $N_{\mathrm{train}} \geq N^*$ leads to

$$\mathbb{E}_{\mathbf{S}\sim\mathbb{P}_{\mathcal{L}}}\|\mathbf{Q}^*(\mathbf{S},\cdot) - \mathbf{Q}_f(\mathbf{S},\cdot)\|_1 \leq 2\cdot\mathcal{R}_{\mathrm{pre}}(\mathbf{\Theta}) \leq 2\cdot\frac{\epsilon}{2} = \epsilon,$$

which concludes the proof. $\qquad\square$

### D.7 PROOF OF THEOREM 4.4

In this section, we detail the proof of Theorem 4.4. We first recall the problem setup.

**Markov chains inputs.** In this section, we give as input of the model a single Markov chain $X = (\mathbf{X}_1, \ldots, \mathbf{X}_{N_{\mathrm{icl}}})$ with finite, discrete state space $\Omega$ of size $d$ with transition probability $\mathbb{P}$. We assume the $\mathbf{X}_n$ are already tokenized and thus we have $\Omega \subset \mathcal{V}$. We denote the sequence of tokens the LLM receives by $\mathbf{S}_n = (\mathbf{X}_1, \ldots, \mathbf{X}_n)$ if $n \leq K$ and $\mathbf{S}_n = (\mathbf{X}_{n-K+1}, \ldots, \mathbf{X}_n)$ otherwise due to the deletion process (see Definition B.2). In particular, the $\mathbf{S}_n$ are elements of $\mathcal{V}_K^*$. We note that $S = (\mathbf{S}_1, \ldots, \mathbf{S}_{N_{\mathrm{icl}}})$ is also a Markov chain (see Appendix D.7.1). By definition of $\mathbb{P}$, we know that for any $n \in [N_{\mathrm{icl}}]$, the next token $\mathbf{X}_{n+1}$ follows the distribution $\mathbb{P}(\cdot \mid \mathbf{S}_n)$. We assume that there exists a positive constant $p_{\min}$ that lower bounds all the transition probability between states, i.e., $\forall n \in [N_{\mathrm{icl}}], \forall x, y \in \Omega, \quad \mathbb{P}(\mathbf{X}_{n+1} = y \mid \mathbf{X}_n = x) \geq p_{\min} > 0$. This is akin to the ambiguity of language constant $c_0$ considered in the previous section and in Hu et al. (2024); Wies et al. (2024); Xie et al. (2022); Zhang et al. (2023b).

**Next token probability distribution.** An important difference with the setting considered in Theorem 4.1 is that here, we predict a probability distribution on the state space $\Omega$ of the Markov chain and not on the vocabulary of the LLM $\mathcal{V}$. To that end, we restrict the predicted probability given the past tokens $\mathbf{S}_n$ to the state space $\Omega$. Formally, denoting the output of the last layer of $f_{\mathbf{\Theta}}$ by $\mathbf{S}^{(L)}$, the last layer before the softmax outputs a vector $\mathbf{u} = \frac{1}{n_\tau}\mathbf{W}_U\mathbf{S}^{(L)}\mathbb{1}_n \in \mathbb{R}^T$. We first extract the entries of $\mathbf{u}$ whose index $i$ are such that the $i$-th element of the vocabulary space $\mathcal{V}$ is in $\Omega$. This can be formalized as follows. We denote by $\mathbb{I}_d = (i_1 \leq i_2 \leq \ldots \leq i_d) \in [T]^d$ the subset of $d$ distinct

elements of $[T]$ and consider the matrix $\mathbf{M}_j = \mathbf{e}_{i_j}^\top$, where $\mathbf{e}_{i_j} \in \mathbb{R}^T$ has value 1 at entry $i_j \in \mathbb{I}$ and 0 elsewhere. Extracting only the $d$ entries of $\mathbf{u}$ that corresponds to the state space yields a vector in $\mathbb{R}^d$ that writes $\mathbf{v} = \frac{1}{n\tau}\mathbf{M}\mathbf{W}_U\mathbf{S}^{(L)}\mathbb{1}_n \in \mathbb{R}^T$. Similarly to Appendix B, the probability distribution of next token $\mathbf{X}_{n+1}$ provided by the LLM $f_\Theta$ now writes

$$\mathbb{P}_\Theta(\cdot \mid \mathbf{S}_n) = \mathrm{softmax}\left(\frac{1}{n\tau}\mathbf{M}\mathbf{W}_U\mathbf{S}^{(L)}\mathbb{1}_n\right) \in \Delta_d.$$

We aim to obtain a similar generalization bound than in Theorem 4.1 where the reference probability distribution is the Markov chain transition probability $\mathbb{P}$ instead of the probability distribution of language $\mathbb{P}_{\mathcal{L}}$. In particular, $\mathbb{P}$ will replace $\mathbb{P}_{\mathcal{L}}$ in the definition of the risks in Eq. (2). We provide below an overview of the proof before detailing it.

**Overview of the proof.** We are going to use McDiarmid's inequality for Markov chains of Paulin (2015, Corollary 2.11). To adapt their arguments to our setting, we bound the total variation between the true probability of the next token and the one estimated by the LLM. The rest of this section is organized as follows. First, in Appendix D.7.1, we show that $S = (\mathbf{S}_1, \ldots, \mathbf{S}_{N_{\mathrm{icl}}})$ is a Markov chain. Then in Appendix D.7.2, we adapt the concentration inequality of Paulin (2015, Corollary 2.11). Afterwards in Appendix D.7.3, we show how to bound the total variation between the true and the estimated probability of the next token. Finally Appendix D.7.4 concludes the proof.

### D.7.1 CONNECTION BETWEEN TOKENS AND SEQUENCES OF TOKENS MARKOV CHAINS

We first show that $S = (\mathbf{S}_1, \ldots, \mathbf{S}_{N_{\mathrm{icl}}})$ is also a Markov chain.

**Lemma D.17.** *Consider a sequence (not necessarily a Markov chain) $X = (\mathbf{X}_1, \ldots, \mathbf{X}_N)$ with values in $\Omega$ and let $\mathbf{S}_n = (\mathbf{X}_1, \ldots, \mathbf{X}_n)$ if $n < K$ and $\mathbf{S}_n = (\mathbf{X}_{n-K+1}, \ldots, \mathbf{X}_n)$ otherwise. Then, the sequence $S = (\mathbf{S}_1, \ldots, \mathbf{S}_N)$ is a Markov chain with state space $\Omega_K^*$ that contains the sequence of elements in $\Omega$ of length smaller than $K$.*

*Proof.* By definition of the $\mathbf{S}_n$, we know that they take values in $\Omega_K^*$. Let $x_1, \ldots, x_{n+1} \in \Omega$. We first assume that $n > K$ and denote $s_i = (x_{n-K+1}, \ldots, x_i)$. We have

$$\mathbb{P}(\mathbf{S}_{n+1} = s_{n+1} \mid \mathbf{S}_n = s_n, \ldots, \mathbf{S}_{n-K+1} = s_{n-K+1})$$
$$= \mathbb{P}(\mathbf{S}_{n+1} = s_{n+1} \mid \mathbf{X}_n = x_n, \ldots, \mathbf{X}_{n-K+1} = x_{n-K+1})$$
$$= \mathbb{P}(\mathbf{S}_{n+1} = s_{n+1} \mid \mathbf{S}_n = s_n). \qquad \text{(by definition of } \mathbf{S}_n)$$

Similarly, we assume $n < K$ and denote $s_i = (x_1, \ldots, x_i)$. We have

$$\mathbb{P}(\mathbf{S}_{n+1} = s_{n+1} \mid \mathbf{S}_n = s_n, \ldots, \mathbf{S}_1 = s_1)$$
$$= \mathbb{P}(\mathbf{S}_{n+1} = s_{n+1} \mid \mathbf{X}_n = x_n, \ldots, \mathbf{X}_1 = x_1)$$
$$= \mathbb{P}(\mathbf{S}_{n+1} = s_{n+1} \mid \mathbf{S}_n = s_n). \qquad \text{(by definition of } \mathbf{S}_n)$$

Finally, for $n = K$, we denote $s_i = (x_1, \ldots, x_i)$ for $i \leq K$ and $s_{K+1} = (x_2, \ldots, x_{K+1})$. We have

$$\mathbb{P}(\mathbf{S}_{K+1} = s_{K+1} \mid \mathbf{S}_n = s_n, \ldots, \mathbf{S}_2 = s_2)$$
$$= \mathbb{P}(\mathbf{S}_{K+1} = s_{K+1} \mid \mathbf{X}_K = x_K, \ldots, \mathbf{X}_1 = x_1)$$
$$= \mathbb{P}(\mathbf{S}_{K+1} = s_{K+1} \mid \mathbf{S}_K = s_K).$$
$$\text{(by definition of } \mathbf{S}_K)$$

This establishes the Markov property for $\mathbf{S}$. $\qquad \square$

### D.7.2 CONCENTRATION INEQUALITIES FOR MARKOV CHAINS

We first state a concentration inequality for time-homogeneous Markov chains that will be used to obtain our final bound.

**Proposition D.18** (McDiarmid's inequality for time-homogeneous Markov chains). *Let $S :=$ $(\mathbf{S}_1, \ldots, \mathbf{S}_N)$ be a Markov chain with value in a discrete, finite state space $\Omega$ and mixing time $t_{\mathrm{mix}}(\varepsilon)$. Let $t_{\min} := \inf_{0 \leq \varepsilon < 1} t_{\mathrm{mix}}\left(\frac{\varepsilon}{2}\right) \cdot \left(\frac{2-\varepsilon}{1-\varepsilon}\right)^2$. If $f : \Omega \to \mathbb{R}$ is such that there exists $\mathbf{c} \in \mathbb{R}^N$ satisfying*

$$\forall \mathbf{x}, \mathbf{y} \in \Omega, \quad f(\mathbf{x}) - f(\mathbf{y}) \leq \sum_{i=1}^{N} \mathbf{c}_i \mathbb{1}_{\{\mathbf{x}_i \neq \mathbf{y}_i\}},$$

*then we have for any $u \geq 0$,*

$$\mathbb{P}(|f(S) - \mathbb{E}_S[f(S)]| \geq u) \leq 2 \exp\left(\frac{-2u^2}{\|\mathbf{c}\|_2^2 \cdot t_{\min}}\right).$$

*Proof.* We recall that Corollary 2.11 of Paulin (2015) ensures that for such a function $f$, we have

$$\mathbb{P}(|f(S) - \mathbb{E}[f(S)]| \geq u) \leq 2 \exp\left(\frac{-2u^2}{\|\mathbf{c}\|_2^2 \cdot \tau_{\min}}\right), \tag{26}$$

where $\tau_{\min}$ is defined as

$$\tau_{\min} := \inf_{0 \leq \varepsilon < 1} \tau(\varepsilon) \left(\frac{2-\varepsilon}{1-\varepsilon}\right)^2,$$

with $\tau(\varepsilon)$ being the mixing time of a Markov chain *without assuming time homogeneity* (see Paulin (2015, Definition 1.4)). As in our case, we assume the time homogeneity, this inequality in Eq. (26) has to be adapted. Following Remark 1.5 of Paulin (2015), we notice that

$$\forall \varepsilon \in [0, 1], \quad \tau(2\varepsilon) \leq t_{\mathrm{mix}}(\varepsilon) \leq \tau(\varepsilon).$$

Let $0 \leq \varepsilon < 1$. Using the fact that $\left(\frac{2-\varepsilon}{1-\varepsilon}\right)^2 > 0$, the previous inequality ensures

$$\tau(\varepsilon) \leq t_{\mathrm{mix}}\left(\frac{\varepsilon}{2}\right) \iff \tau(\varepsilon)\left(\frac{2-\varepsilon}{1-\varepsilon}\right)^2 \leq t_{\mathrm{mix}}\left(\frac{\varepsilon}{2}\right)\left(\frac{2-\varepsilon}{1-\varepsilon}\right)^2.$$

Taking the infimum on the left-hand side leads to

$$\tau_{\min} = \inf_{0 \leq \varepsilon < 1} \tau(\varepsilon)\left(\frac{2-\varepsilon}{1-\varepsilon}\right)^2 \leq t_{\mathrm{mix}}\left(\frac{\varepsilon}{2}\right)\left(\frac{2-\varepsilon}{1-\varepsilon}\right)^2.$$

As we took $\varepsilon$ arbitrary in $[0, 1)$, we can take the infimum on the right-hand side, which leads to

$$\tau_{\min} \leq t_{\min}.$$

As the function $x \to \exp\left(\frac{-2u^2}{\|\mathbf{c}\|_2^2 x}\right)$ is decreasing, we finally obtain

$$\exp\left(\frac{-2u^2}{\|\mathbf{c}\|_2^2 \tau_{\min}}\right) \leq \exp\left(\frac{-2u^2}{\|\mathbf{c}\|_2^2 t_{\min}}\right). \tag{27}$$

Combining Eqs. (26) and (27) concludes the proof. $\square$

Similarly to Theorem 4.1, we want to apply Proposition D.18 to a function $f$ that consists of sums of total variation. We investigate in the next section how to find the bounding vector $\mathbf{c}$ to apply Proposition D.18.

### D.7.3 FINDING THE BOUNDING VECTOR

We want to apply the same arguments as in the proof of Theorem 4.1 to find the bounding vector $\mathbf{c}$. The only difference in terms of setting is the definition of the probability of the next token. Indeed, in our case, we apply an extraction matrix $\mathbf{M} \in \mathbb{R}^{d \times T}$ to recover the $d$ states of the input Markov chain. We first prove the following technical lemma.

**Lemma D.19.** *Let $d \leq T$ and consider a subset of $d$ distinct elements of $[T]$ that writes $\mathbb{I}_d = (i_1 \leq i_2 \leq \ldots \leq i_d) \in [T]^d$. We denote by $\mathbf{M} \in \mathbb{R}^{d \times T}$ the matrix with rows $\mathbf{M}_j = \mathbf{e}_{i_j}^\top$, where $\mathbf{e}_{i_j} \in \mathbb{R}^T$ has value 1 at entry $i_j \in \mathbb{I}$ and 0 elsewhere. For any vector $\mathbf{u} \in \mathbb{R}^T$, we have*

$$\|\mathbf{M}\mathbf{u}\|_1 \leq \|\mathbf{u}\|_1.$$

*Proof.* By definition of the $\ell_1$-norm, we have

$$\|\mathbf{M}\mathbf{u}\|_1 = \sum_{k=1}^{d}|\sum_{l=1}^{T}\mathbf{M}_{kl}\mathbf{u}_l| \leq \sum_{k=1}^{d}\sum_{l=1}^{T}|\mathbf{M}_{kl}\mathbf{u}_l| \leq \sum_{l=1}^{T}|\mathbf{u}_l|\sum_{k=1}^{d}|\mathbf{M}_{kl}|.$$

Moreover, each column of $\mathbf{M}$ contains at most one non-zero entry (with value 1). Otherwise, it means that two $\mathbf{e}_{i_j}$ are identical (as they only have one non-zero entry with value 1, having it at the same position ensures their equality) which contradicts the fact that the $i_j$ where taken distinct. Hence, for all $l$, we have $\sum_{k=1}^{d}|\mathbf{M}_{kl}| \leq 1$, which concludes the proof. $\qquad\square$

We now prove a lemma analogous to Lemma D.8.

**Lemma D.20.** *Let $\mathbf{S} \in \mathbb{R}^{r \times n}$ denote the entry of the LLM $f_\Theta$ and $\mathbf{S}^{(L)}$ denote the output of the last layer before the softmax. Let $d \leq T$ and consider a subset of $d$ distinct elements of $[T]$ that writes $\mathbb{I}_d = (i_1 \leq i_2 \leq \ldots \leq i_d) \in [T]^d$. We denote by $\mathbf{M} \in \mathbb{R}^{d \times T}$ the matrix with rows $\mathbf{M}_j = \mathbf{e}_{i_j}^\top$, where $\mathbf{e}_{i_j} \in \mathbb{R}^T$ has value 1 at entry $i_j \in \mathbb{I}$ and 0 elsewhere. Then, the following inequality holds*

$$\frac{1}{n\tau}\|\mathbf{M}\mathbf{W}_U\mathbf{S}^{(L)}\mathbb{1}_n\|_1 \leq \frac{1}{\tau}\|\mathbf{W}_U^\top\|_{2,1}.$$

*Proof.* Applying Lemma D.19 with the matrix $\mathbf{M} \in \mathbb{R}^d$ and the vector $\frac{1}{n\tau}\mathbf{W}_U\mathbf{X}^{(L)}\mathbb{1}_n \in \mathbb{R}^T$ leads to

$$\frac{1}{n\tau}\|\mathbf{M}\mathbf{W}_U\mathbf{S}^{(L)}\mathbb{1}_n\|_1 \leq \frac{1}{n\tau}\|\mathbf{W}_U\mathbf{X}^{(L)}\mathbb{1}_n\|_1.$$

Applying Lemma D.8 concludes the proof. $\qquad\square$

The previous lemma can be used to show that the logarithm of the ratio between the true probability of the next token and the one estimated by the LLM $f_\Theta$ is upper bounded as a function of the number of states of the Markov chain $d$, the temperature $\tau$, the upper-bound on $\mathbf{W}_U$ and some constant related to the ambiguity of language (see Eq. (1)).

**Proposition D.21** (Upper-bound on the logarithm). *Consider an LLM $f_\Theta \in \mathcal{F}$ and an input Markov chain $X = (\mathbf{X}_1, \ldots, \mathbf{X}_{N_{\text{icl}}})$ with $d$ states. We recall that $B_U$ is the upper bound on the norm of $\mathbf{W}_U$ in the definition of parameter space $\mathcal{W}$, $\tau$ is the softmax temperature, and $p_{\min}$ is the constant related to the minimal transition probability between states. We have*

$$\forall n \in [N], \quad \left|\log\left(\frac{\mathbb{P}(\mathbf{X}_{n+1} \mid \mathbf{S}_n)}{\mathbb{P}_\Theta(\mathbf{X}_{n+1} \mid \mathbf{S}_n)}\right)\right| \leq \bar{B} = \max\{\log(d) + \frac{2B_U}{\tau}, \log\left(\frac{1}{p_{\min}}\right)\}.$$

*Proof.* The main idea of the proof is to bound the probability ratio and use the non-decreasing monotonicity of the log. Let $n \in [N]$. The model $f_\Theta$ receives as input sequences of tokens $\mathbf{S}_n$ of size $n \leq K$. We first lower-bound each term of the probability ratio. By definition of $p_{\min}$, we have

$$\mathbb{P}(\mathbf{X}_{n+1} \mid \mathbf{S}_n) = \mathbb{P}(\mathbf{X}_{n+1} \mid \mathbf{X}_n) \geq p_{\min} > 0, \tag{28}$$

where we used the Markov property for the first equality. We want to obtain a similar inequality for $\mathbb{P}_\Theta(\mathbf{X}_{n+1} \mid \mathbf{S}_n)$. As the parameters $\Theta$ of the LLM are in $\mathcal{W}$, we know that $\|\mathbf{W}_U^\top\|_{2,1} \leq B_U$. Lemma D.20 ensures that

$$\|\frac{1}{n\tau}\mathbf{M}\mathbf{W}_U\mathbf{S}^{(L)}\mathbb{1}_T\|_1 \leq \frac{1}{\tau}\|\mathbf{W}_U^\top\|_{2,1} \leq \frac{B_U}{\tau}.$$

We can then apply Lemma D.7 with $c_1 = \frac{B_U}{\tau}$ and given that $\frac{1}{T\tau}\mathbf{M}\mathbf{W}_U\mathbf{S}^{(L)}\mathbb{1}_T \in \mathbb{R}^d$, it leads to

$$\mathbb{P}_\Theta(\cdot \mid \mathbf{S}_n) = \text{softmax}\left(\frac{1}{n\tau}\mathbf{M}\mathbf{W}_U\mathbf{S}^{(L)}\mathbb{1}_n\right) \geq \frac{1}{d\exp(2B_U/\tau)},$$

where the inequality holds for each component of $\mathbb{P}_\Theta(\cdot \mid \mathbf{S}_n)$. This is in particular the case $\mathbb{P}_\Theta(\mathbf{X}_{n+1} \mid \mathbf{S}_n)$ which is the entry we are interested in, i.e., we have

$$\mathbb{P}_\Theta(\mathbf{X}_{n+1} \mid \mathbf{S}_n) \geq \frac{1}{d\exp(2B_U/\tau)}. \tag{29}$$

Going back to the ratio of probability, consider the situation where we have

$$\frac{\mathbb{P}(\mathbf{X}_{n+1} \mid \mathbf{S}_n)}{\mathbb{P}_\Theta(\mathbf{X}_{n+1} \mid \mathbf{S}_n)} \geq 1.$$

Then, using Eq. (29), we have

$$1 \leq \frac{\mathbb{P}(\mathbf{X}_{n+1} \mid \mathbf{S}_n)}{\mathbb{P}_\Theta(\mathbf{X}_{n+1} \mid \mathbf{S}_n)} \leq \frac{1}{\mathbb{P}_\Theta(\mathbf{X}_{n+1} \mid \mathbf{S}_n)} \leq d\exp(2B_U/\tau),$$

which implies, as the $\log$ is non-decreasing monotonically,

$$0 \leq \log\left(\frac{\mathbb{P}(\mathbf{X}_{n+1} \mid \mathbf{S}_n)}{\mathbb{P}_\Theta(\mathbf{X}_{n+1} \mid \mathbf{S}_n)}\right) \leq \log(d\exp(2B_U/\tau)) = \log(d) + \frac{2B_U}{\tau}. \tag{30}$$

Similarly, consider the case where we have

$$\frac{\mathbb{P}(\mathbf{X}_{n+1} \mid \mathbf{S}_n)}{\mathbb{P}_\Theta(\mathbf{X}_{n+1} \mid \mathbf{S}_n)} \leq 1.$$

Then, we have

$$\frac{\mathbb{P}_\Theta(\mathbf{X}_{n+1} \mid \mathbf{S}_n)}{\mathbb{P}(\mathbf{X}_{n+1} \mid \mathbf{S}_n)} \geq 1,$$

and similarly to above, we can use Eq. (28) to obtain

$$1 \leq \frac{\mathbb{P}_\Theta(\mathbf{X}_{n+1} \mid \mathbf{S}_n)}{\mathbb{P}(\mathbf{X}_{n+1} \mid \mathbf{S}_n)} \leq \frac{1}{\mathbb{P}(\mathbf{X}_{n+1} \mid \mathbf{S}_n)} \leq \frac{1}{p_{\min}}.$$

This implies

$$0 \leq \log\left(\frac{\mathbb{P}_\Theta(\mathbf{X}_{n+1} \mid \mathbf{S}_n)}{\mathbb{P}(\mathbf{X}_{n+1} \mid \mathbf{S}_n)}\right) \leq \log\left(\frac{1}{p_{\min}}\right),$$

which also rewrites

$$0 \leq -\log\left(\frac{\mathbb{P}(\mathbf{X}_{n+1} \mid \mathbf{S}_n)}{\mathbb{P}_\Theta(\mathbf{X}_{n+1} \mid \mathbf{S}_n)}\right) \leq \log\left(\frac{1}{p_{\min}}\right). \tag{31}$$

By definition of the absolute value, combining Eq. (30) and Eq. (31) leads to

$$\left|\log\left(\frac{\mathbb{P}(\mathbf{X}_{n+1} \mid \mathbf{S}_n)}{\mathbb{P}_\Theta(\mathbf{X}_{n+1} \mid \mathbf{S}_n)}\right)\right| \leq \max\{\log(d) + \frac{2B_U}{\tau}, \log\left(\frac{1}{p_{\min}}\right)\}.$$

This concludes the proof. $\qquad\square$

We are now ready to upper-bound the total variation.

**Corollary D.22** (Upper-bound on the total variation). *Consider an LLM $f_\Theta \in \mathcal{F}$ and an input Markov chain $X = (\mathbf{X}_1, \ldots, \mathbf{X}_{N_{\text{icl}}})$ with $d$ states. We recall that $B_U$ is the upper bound on the norm of $\mathbf{W}_U$ in the definition of parameter space $\mathcal{W}$, $\tau$ is the softmax temperature, and $p_{\min}$ is the constant related to the minimal transition probability between states. We have*

$$\forall n \in [N], \quad d_{\text{TV}}(\mathbb{P}(\cdot \mid \mathbf{S}_n), \mathbb{P}_\Theta(\cdot \mid \mathbf{S}_n)) \leq \sqrt{2\max\{\log(d) + \frac{2B_U}{\tau}, \log\left(\frac{1}{p_{\min}}\right)\}} := c_4. \tag{32}$$

*Proof.* Using Proposition D.21, we can directly apply Lemma D.6 with $B = \max\{\log(d) + \frac{2B_U}{\tau}, \log\left(\frac{1}{p_{\min}}\right)\}$ for any $n \in [N]$. It leads to

$$\forall n \in [N], \quad d_{\mathrm{TV}}(\mathbb{P}(\cdot \mid \mathbf{S}_n), \mathbb{P}_{\boldsymbol{\Theta}}(\cdot \mid \mathbf{S}_n)) \leq \sqrt{2\max\{\log(d) + \frac{2B_U}{\tau}, \log\left(\frac{1}{p_{\min}}\right)\}}.$$

This concludes the proof. $\qquad\square$

### D.7.4 Concluding the Proof

We are now ready to state our main result.

> **Theorem D.23** (Restatement of Theorem 4.4). *Consider an LLM $f_{\boldsymbol{\Theta}} \in \mathcal{F}$. We provide as input of $f_{\boldsymbol{\Theta}}$ a $d-$state Markov chain $X = (\mathbf{X}_1, \ldots, \mathbf{X}_{N_{\mathrm{icl}}})$. The sequence of subsequences of the first $n$ terms is denoted by $S = (\mathbf{S}_1, \ldots, \mathbf{S}_{N_{\mathrm{icl}}})$. $S$ is also a Markov chain, and we denote by $t_{\mathrm{mix}}(\varepsilon)$ its mixing time. Let $t_{\min} := \inf_{0 \leq \varepsilon < 1} t_{\mathrm{mix}}\left(\frac{\varepsilon}{2}\right) \cdot \left(\frac{2-\varepsilon}{1-\varepsilon}\right)^2$. Let $\delta > 0$. Then, with probability at least $1 - \delta$,*
>
> $$\mathcal{R}_{\mathrm{icl}}(\boldsymbol{\Theta}) \leq \inf_{\boldsymbol{\vartheta} \in \mathcal{W}_{\mathrm{mc}}} \{\widehat{\mathcal{R}}_{\mathrm{icl}}(\boldsymbol{\vartheta}) + K(\boldsymbol{\vartheta}, \boldsymbol{\Theta})\} + \bar{B}\sqrt{\frac{t_{\min}}{N_{\mathrm{icl}}}}\sqrt{\log\left(\frac{2}{\delta}\right)},$$
>
> *where $\bar{B}$ is a constant depending on the parameters of the problem. More precisely,*
>
> $$\bar{B} = 2\sqrt{\max\{\log(d) + \frac{2B_U}{\tau}, \log\left(\frac{1}{p_{\min}}\right)\}}.$$

*Proof.* Let $\boldsymbol{\vartheta} \in \mathcal{W}_{\mathrm{mc}}$. We first benefit from the metric properties of the total variation to decompose the risk.

$$\mathcal{R}_{\mathrm{icl}}(\boldsymbol{\Theta}) = \frac{1}{N_{\mathrm{icl}}} \sum_{n=1}^{N_{\mathrm{icl}}} \mathbb{E}_{\mathbf{S}_n}[d_{\mathrm{TV}}(\mathbb{P}(\cdot \mid \mathbf{S}_n), \mathbb{P}_{\boldsymbol{\Theta}}(\cdot \mid \mathbf{S}_n))]$$

$$\leq \frac{1}{N_{\mathrm{icl}}} \sum_{n=1}^{N_{\mathrm{icl}}} \mathbb{E}_{\mathbf{S}_n}[d_{\mathrm{TV}}(\mathbb{P}(\cdot \mid \mathbf{S}_n), \mathbb{P}_{\boldsymbol{\vartheta}}(\cdot \mid \mathbf{S}_n)) + d_{\mathrm{TV}}(\mathbb{P}_{\boldsymbol{\vartheta}}(\cdot \mid \mathbf{S}_n), \mathbb{P}_{\boldsymbol{\Theta}}(\cdot \mid \mathbf{S}_n))]$$

$$\leq \frac{1}{N_{\mathrm{icl}}} \sum_{n=1}^{N_{\mathrm{icl}}} \mathbb{E}_{\mathbf{S}_n}[d_{\mathrm{TV}}(\mathbb{P}(\cdot \mid \mathbf{S}_n), \mathbb{P}_{\boldsymbol{\vartheta}}(\cdot \mid \mathbf{S}_n))]$$

$$+ \frac{1}{N_{\mathrm{icl}}} \sum_{n=1}^{N_{\mathrm{icl}}} \mathbb{E}_{\mathbf{S}_n}[d_{\mathrm{TV}}(\mathbb{P}_{\boldsymbol{\vartheta}}(\cdot \mid \mathbf{S}_n), \mathbb{P}_{\boldsymbol{\Theta}}(\cdot \mid \mathbf{S}_n))]$$

$$\leq \mathcal{R}_{\mathrm{icl}}(\boldsymbol{\vartheta}) + K(\boldsymbol{\vartheta}, \boldsymbol{\Theta}). \tag{33}$$

By definition of the risk, we have

$$\widehat{\mathcal{R}}_{\mathrm{icl}}(\boldsymbol{\vartheta}) = \frac{1}{N_{\mathrm{icl}}} \sum_{n=1}^{N_{\mathrm{icl}}} \underbrace{d_{\mathrm{TV}}(\mathbb{P}(\cdot \mid \mathbf{S}_n), \mathbb{P}_{\boldsymbol{\vartheta}}(\cdot \mid \mathbf{S}_n))}_{=g_n(\mathbf{S}_n)} = \frac{1}{N_{\mathrm{icl}}} \sum_{n=1}^{N_{\mathrm{train}}} g_n(\mathbf{S}_n) = f(\mathbf{S}_1, \ldots, \mathbf{S}_{N_{\mathrm{icl}}}) = f(S).$$

Using Corollary D.22, we know that

$$|g_n(\mathbf{S}_n)| \leq \sqrt{2\max\{\log(d) + \frac{2B_U}{\tau}, \log\left(\frac{1}{p_{\min}}\right)\}} := c_4.$$

Similarly to Theorem 4.1, and using the fact that $S = (\mathbf{S}_1, \ldots, \mathbf{S}_{N_{\text{icl}}})$ is a Markov chain, we can show that choosing $\mathbf{c} \in \mathbb{R}^{N_{\text{icl}}}$ with all entries equal to $\frac{2c_4}{N_{\text{icl}}}$ ensures that $f$ verifies the condition in Proposition D.5, i.e.,

$$\forall S, \Sigma, \quad f(S) - f(\Sigma) \leq \sum_{n=1}^{N_{\text{icl}}} \mathbf{c}_n \mathbb{1}_{\{\mathbf{S}_n \neq \mathbf{\Sigma}_n\}}.$$

Putting everything together, we can apply Proposition D.18 which leads to

$$\forall u \geq 0, \quad \mathbb{P}(|f(S) - \mathbb{E}_S[f(S)]| \geq u) \leq 2 \exp\left(\frac{-2u^2}{t_{\min}\|\mathbf{c}\|_2^2}\right). \tag{34}$$

Let $u \geq 0$. We have the following events ordering

$$(\mathbb{E}_S[f(S)] - f(S) \geq u) \subseteq (\mathbb{E}_S[f(S)] - f(S) \geq u) \cup (f(S) - \mathbb{E}_S[f(S)] \geq u)$$
$$= (|f(S) - \mathbb{E}_S[f(S)]| \geq u).$$

Hence, as $u$ was taken arbitrary and using Eq. (34), we have

$$\forall u \geq 0, \quad \mathbb{P}(\mathbb{E}_S[f(S)] - f(S) \geq u) \leq 2 \exp\left(\frac{-2u^2}{t_{\min}\|\mathbf{c}\|_2^2}\right).$$

We recall that by definition

$$f(S) = \widehat{\mathcal{R}}_{\text{icl}}(\boldsymbol{\vartheta}) \text{ and } \mathcal{R}_{\text{icl}}(\boldsymbol{\vartheta}) = \mathbb{E}_S\left[\widehat{\mathcal{R}}_{\text{icl}}(\boldsymbol{\vartheta})\right].$$

Moreover, the inequality on the probability holds for any $u \geq 0$, we can choose $u$ such that

$$\delta = 2 \exp\left(\frac{-2u^2}{t_{\min}\mathbf{c}_2^2}\right) \iff \frac{-2u^2}{t_{\min}\|\mathbf{c}\|_2^2} = \log\left(\frac{\delta}{2}\right) \iff u^2 = \frac{1}{2} t_{\min}\|\mathbf{c}\|_2^2 \log\left(\frac{2}{\delta}\right)$$

$$\iff u = \frac{1}{\sqrt{2}}\sqrt{t_{\min}}\|\mathbf{c}\|_2\sqrt{\log\left(\frac{2}{\delta}\right)}.$$

Using the fact that

$$\|\mathbf{c}\|_2 = \sqrt{\sum_{n=1}^{N_{\text{icl}}} \mathbf{c}_n^2} = \sqrt{\sum_{n=1}^{N_{\text{icl}}} \left(\frac{2c_4}{N_{\text{icl}}}\right)^2} = \sqrt{\sum_{n=1}^{N_{\text{icl}}} \frac{4c_4^2}{N_{\text{icl}}^2}} = \sqrt{\frac{4c_4^2}{N_{\text{icl}}}} = \frac{2c_4}{\sqrt{N_{\text{icl}}}}.$$

Using the fact that $c_4 = \sqrt{2\max\{\log(d) + \frac{2B_U}{\tau}, \log\left(\frac{1}{p_{\min}}\right)\}}$ (Corollary D.22), we obtain

$$u = \frac{1}{\sqrt{2}} \frac{2c_4}{\sqrt{N_{\text{icl}}}} \sqrt{t_{\min}}\sqrt{\log\left(\frac{2}{\delta}\right)} = \frac{\sqrt{2}c_4}{\sqrt{N_{\text{icl}}}} \sqrt{t_{\min}}\sqrt{\log\left(\frac{2}{\delta}\right)}$$

$$= \frac{2\sqrt{t_{\min}}\sqrt{\max\{\log(d) + \frac{2B_U}{\tau}, \log\left(\frac{1}{p_{\min}}\right)\}}}{\sqrt{N_{\text{train}}}} \sqrt{\log\left(\frac{2}{\delta}\right)}$$

$$= \bar{B}\sqrt{\frac{t_{\min}}{N_{\text{icl}}}} \sqrt{\log\left(\frac{2}{\delta}\right)},$$

where we define

$$\bar{B} = 2\sqrt{\max\{\log(d) + \frac{2B_U}{\tau}, \log\left(\frac{1}{p_{\min}}\right)\}}.$$

Putting everything together, we have

$$\mathbb{P}\left(\mathcal{R}_{\text{icl}}(\boldsymbol{\vartheta}) - \widehat{\mathcal{R}}_{\text{icl}}(\boldsymbol{\vartheta}) \geq \bar{B}\sqrt{\frac{t_{\min}}{N_{\text{icl}}}} \sqrt{\log\left(\frac{2}{\delta}\right)}\right) \leq \delta.$$

Taking the opposite event leads to the following inequality with probability at least $1 - \delta$

$$\mathcal{R}_{\mathrm{icl}}(\boldsymbol{\vartheta}) \leq \widehat{\mathcal{R}}_{\mathrm{icl}}(\boldsymbol{\vartheta}) + \bar{B}\frac{\sqrt{t_{\min}}}{\sqrt{N_{\mathrm{icl}}}}\sqrt{\log\left(\frac{2}{\delta}\right)}.$$

Going back to the decomposition of the risk in Eq. (33) and rearranging the terms, we obtain

$$\mathcal{R}_{\mathrm{icl}}(\boldsymbol{\Theta}) \leq \widehat{\mathcal{R}}_{\mathrm{icl}}(\boldsymbol{\vartheta}) + K(\boldsymbol{\Theta}, \boldsymbol{\vartheta}) + \bar{B}\frac{\sqrt{t_{\min}}}{\sqrt{N_{\mathrm{icl}}}}\sqrt{\log\left(\frac{2}{\delta}\right)}.$$

As the left-hand side and the bound function of $\bar{B}$ do not depend on $\boldsymbol{\vartheta}$, we can put them both on the left side of the inequality and then take the infimum on $\boldsymbol{\vartheta}$. Rearranging the terms to keep only $\widehat{\mathcal{R}}_{\mathrm{icl}}(\boldsymbol{\Theta})$ on the left side of the inequality leads to

$$\mathcal{R}_{\mathrm{icl}}(\boldsymbol{\Theta}) \leq \inf_{\boldsymbol{\vartheta} \in \mathcal{W}_{\mathrm{mc}}} \{\widehat{\mathcal{R}}_{\mathrm{icl}}(\boldsymbol{\vartheta}) + K(\boldsymbol{\vartheta}, \boldsymbol{\Theta})\} + \bar{B}\sqrt{\frac{t_{\min}}{N_{\mathrm{icl}}}}\sqrt{\log\left(\frac{2}{\delta}\right)},$$

which concludes the proof. □

# E  ADDITIONAL EXPERIMENTS

## E.1  EXPERIMENTAL SETUP AND TOKENIZATION

**Experimental setup.**  To ensure a fair validation of our theoretical results, we conduct our experiments on some of the most recent and widely used LLMs: `Gemma 2B` (Team et al., 2024), `Llama2 7B & 13B` (Touvron et al., 2023b), `Llama3 8B`, `Llama3.2 1B & 3B` (Dubey et al., 2024), and `Mistral 7Bv0.1` (Jiang et al., 2023).

**Tokenization.**  As the models we consider have different tokenizations, we need to do this step with extra care as it is a crucial part of the experimental procedure. Indeed, LLMs' ability to handle numerical values has been proved to be dependent on the tokenization algorithm (Ali et al., 2024; Gruver et al., 2023; Singh & Strouse, 2024). The most widely used tokenization algorithm to-date, *BPE* (Sennrich et al., 2016), tends to assign tokens to arbitrary 3-digits numbers based on their occurrences in large-scale corpora, and the tokenizer's vocabulary size. As highlighted by (Gruver et al., 2023), this artifact severely hinders LLMs' ability to predict numerical values in-context. This is the case for popular LLMs such as `GPT-3` (Brown et al., 2020). Newer models (`LLama3`, `GPT-3.5`, `GPT-4`) however, tend to have hard-coded rules on top of *BPE*, making them able to encode all 3-digits numbers with their own token. Although this feature would accelerate the ICL procedure by eliminating the need for the *Hierarchy-PDF* algorithm in (Liu et al., 2024), the under-representability of larger numbers in the training data could be an issue. Other tokenization techniques that are numerical values-focused has been presented in the literature (Golkar et al., 2023; Wu et al., 2024), paving the way for another research direction that may benefit our method.

**Rodmap.**  In the rest of this section, we extend our experiments to study the following setups:

- In Appendix E.2: impact of the number of states $d$;
- In Appendix E.3: extension to Markov chains with $p_{\min} = 0$;
- In Appendix E.4: impact of the tokenization;
- In Appendix E.5: extension to dynamical systems.

## E.2  IMPACT OF THE NUMBER OF STATES $d$

We further analyze the effect of the number of states $d$ on the risk and consider randomly generated $d$-state transition matrices in Fig. 9. After a first stage of stagnation, the risk tends to take the correct scaling law coefficient. As in (Liu et al., 2024), we notice that considering randomly generated transition matrices seems to be difficult for an LLM to learn when there are more than 9 states. We interpret this behavior as the distribution shift term in Theorem 4.4. Indeed, the lack of structure in

these transition matrices can hinder the correct decay of this term. Note also that the increase in $d$ tends to implicitly increase $t_{\min}$, which could have an impact on the upper bound on $\mathcal{R}_{\mathrm{icl}}$ (both in the generalization term and in the distribution shift term). We will now consider more structured Markov chains, and look at their impact on decay.

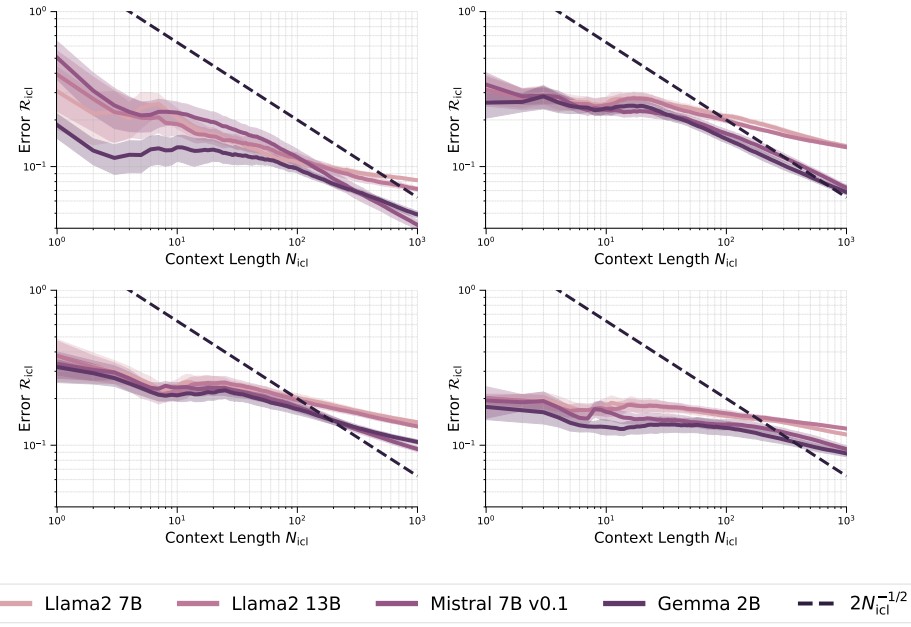

Figure 9: **Impact of the number of states** $d$**.** We plot the risk $\mathcal{R}_{\mathrm{icl}}$ as functions of $N_{\mathrm{icl}}$, with $95\%$ confidence intervals. **Upper Left.** $2-$states Markov transition matrices. **Upper Right.** $4-$states Markov transition matrices. **Lower Left.** $6-$states Markov transition matrices. **Lower Right.** $8-$states Markov transition matrices.

### E.3 MORE STRUCTURED MARKOV CHAINS

In this section, we empirically verify our theoretical results on more general Markov chains that do not verify $p_{\min} > 0$.

### E.3.1 RANDOM WALKS

Random walks are a simple example of more structured Markov chains. Although we still have the possibility of discretizing the kernel of Markov chains with infinite state spaces as it is done in (Liu et al., 2024), we consider two types of random walks on finite state spaces.

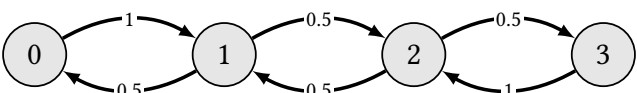

Figure 10: Constrained random walk with $d = 3$.

**Constrained random walk.** We define the transition matrix $P$ of a constrained random walk of $d$ states as in Eq. (35). We draw the probabilistic graph in Fig. 10 for the case $d = 3$.

$$P_{ij} = \begin{cases} 1, & \text{if } i = 0 \text{ and } j = 1, \\ 1, & \text{if } i = \mathrm{d} - 1 \text{ and } j = \mathrm{d} - 2, \\ 0.5, & \text{if } 1 \leq i \leq \mathrm{d} - 2 \text{ and } j = i - 1, \\ 0.5, & \text{if } 1 \leq i \leq \mathrm{d} - 2 \text{ and } j = i + 1, \\ 0, & \text{otherwise.} \end{cases} \quad (35)$$

Fig. 11 highlights the scaling laws of Theorem 4.4, as well as the $\log(d)$ dependency. As before, the best-performing models generalize almost perfectly.

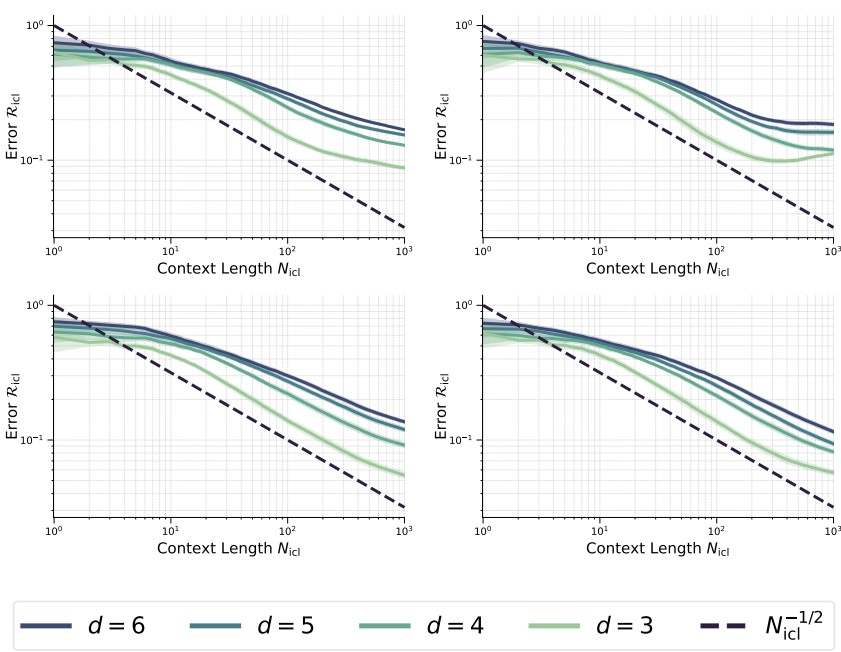

Figure 11: **Constrained random walks.** We plot the risk $\mathcal{R}_{\text{icl}}$ as functions of $N_{\text{icl}}$, with $99\%$ confidence intervals. We consider different size $d$. **Upper Left.** `Llama2 7B` **Upper Right.** `Llama2 13B` **Lower Left.** `Mistral 7Bv0.1` **Lower Right.** `Gemma 2B`

**Polygonal random walk.** We define the transition matrix $\mathbb{P}$ of a polygonal random walk of $d$ states as in Eq. (36). We draw the probabilistic graph in Fig. 12 for the case $d = 4$.

$$P_{ij} = \begin{cases} 0.5, & \text{if } j = (i+1) \mod \mathrm{d} \text{ (clockwise transition)}, \\ 0.5, & \text{if } j = (i-1) \mod \mathrm{d} \text{ (counterclockwise transition)}, \\ 0, & \text{otherwise.} \end{cases} \tag{36}$$

We draw the same conclusions as above for this second type of random walk, in Fig. 13.

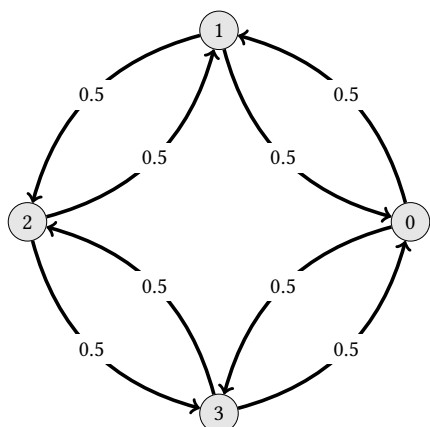

Figure 12: Polygonal random walk with $d = 4$.

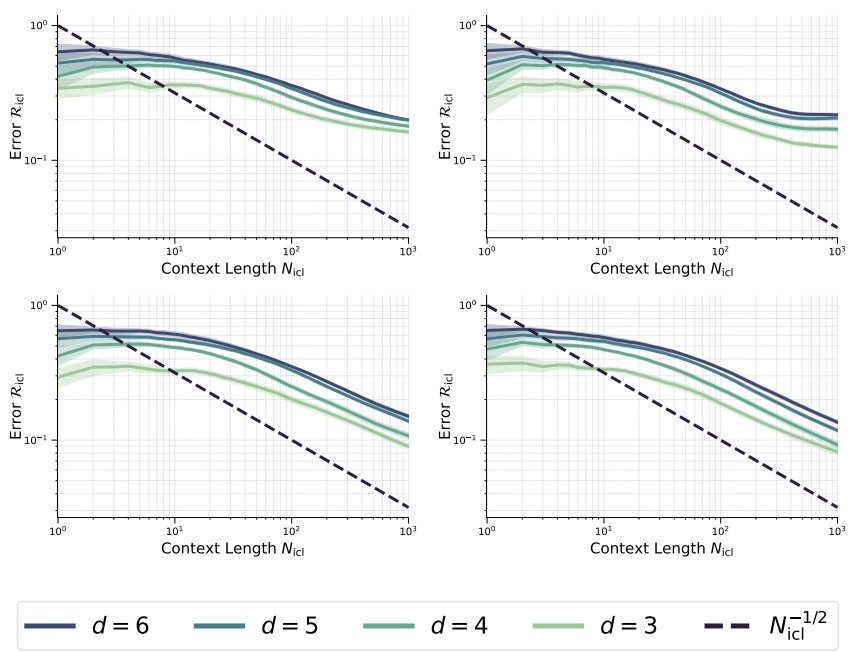

Figure 13: **Polygonal random walks.** We plot the risk $\mathcal{R}_{\text{icl}}$ as functions of $N_{\text{icl}}$, with $99\%$ confidence intervals. We consider different size $d$. **Upper Left.** `Llama2 7B` **Upper Right.** `Llama2 13B` **Lower Left.** `Mistral 7Bv0.1` **Lower Right.** `Gemma 2B`

### E.3.2 INNER CLIQUES AND OUTER RIMS

**Inner Cliques and Outer Rims.** We also want to test our method on the class of Markov chain put forward in (Wolfer & Kontorovich, 2019) to derive their lower bound. Let $\eta > 0$ and $d = 3k$ for some $k \in \mathbb{N}$, and define the collection of Markov matrices $\mathcal{H}_\eta = \{M_{\eta,\boldsymbol{\tau}} : \boldsymbol{\tau} \in \{0,1\}^{d/3}\}$. Every element of this set consists of an *inner clique* and an *outer rim*. $M_{\eta,\boldsymbol{\tau}}$ is the block matrix defined as follows,

$$M_{\eta,\boldsymbol{\tau}} = \begin{pmatrix} C_\eta & R_{\boldsymbol{\tau}} \\ R_{\boldsymbol{\tau}}^{\mathsf{T}} & L_{\boldsymbol{\tau}} \end{pmatrix},$$

where $C_\eta \in \mathbb{R}^{d/3 \times d/3}$, $L_{\boldsymbol{\tau}} \in \mathbb{R}^{2d/3 \times 2d/3}$, and $R_{\boldsymbol{\tau}} \in \mathbb{R}^{d/3 \times 2d/3}$ are given by

$$L_{\boldsymbol{\tau}} = \frac{1}{8} \operatorname{diag}\left(7 - 4\tau_1\varepsilon, 7 + 4\tau_1\varepsilon, \ldots, 7 - 4\tau_{d/3}\varepsilon, 7 + 4\tau_{d/3}\varepsilon\right),$$

$$C_\eta = \begin{pmatrix} \frac{3}{4} - \eta & \frac{\eta}{d/3-1} & \cdots & \frac{\eta}{d/3-1} \\ \frac{\eta}{d/3-1} & \frac{3}{4} - \eta & \ddots & \vdots \\ \vdots & \ddots & \ddots & \frac{\eta}{d/3-1} \\ \frac{\eta}{d/3-1} & \cdots & \frac{\eta}{d/3-1} & \frac{3}{4} - \eta \end{pmatrix},$$

$$R_{\boldsymbol{\tau}} = \frac{1}{8} \begin{pmatrix} 1 + 4\tau_1\varepsilon & 1 - 4\tau_1\varepsilon & 0 & \ldots & \ldots & \ldots & 0 \\ 0 & 0 & 1 + 4\tau_2\varepsilon & 1 - 4\tau_2\varepsilon & 0 & \ldots & 0 \\ \vdots & \vdots & \vdots & \vdots & \vdots & \vdots & \vdots \\ 0 & \ldots & \ldots & \ldots & 0 & 1 + 4\tau_{d/3}\varepsilon & 1 - 4\tau_{d/3}\varepsilon \end{pmatrix}.$$

We provide in Fig. 14 a probabilistic graph of the case $M_{\eta,\mathbf{0}}$ and $d = 9$.

Fig. 15 compares different LLMs with the frequentist method, on the case depicted in Fig. 15 with $\eta = 0.02$. Although the frequentist method achieves a lower loss, the power laws seem to be the same with LLMs.

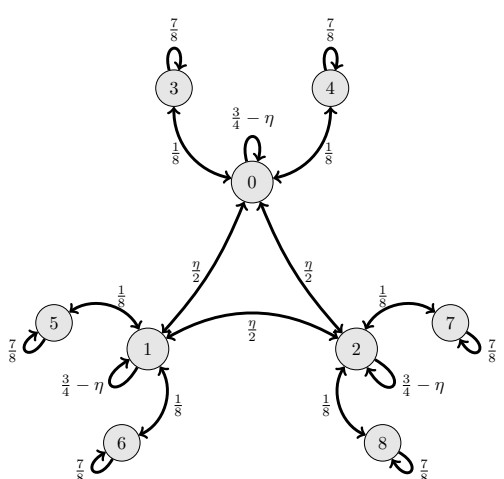

Figure 14: Probabilistic graph of $M_{\eta,\mathbf{0}}$ when $d = 9$.

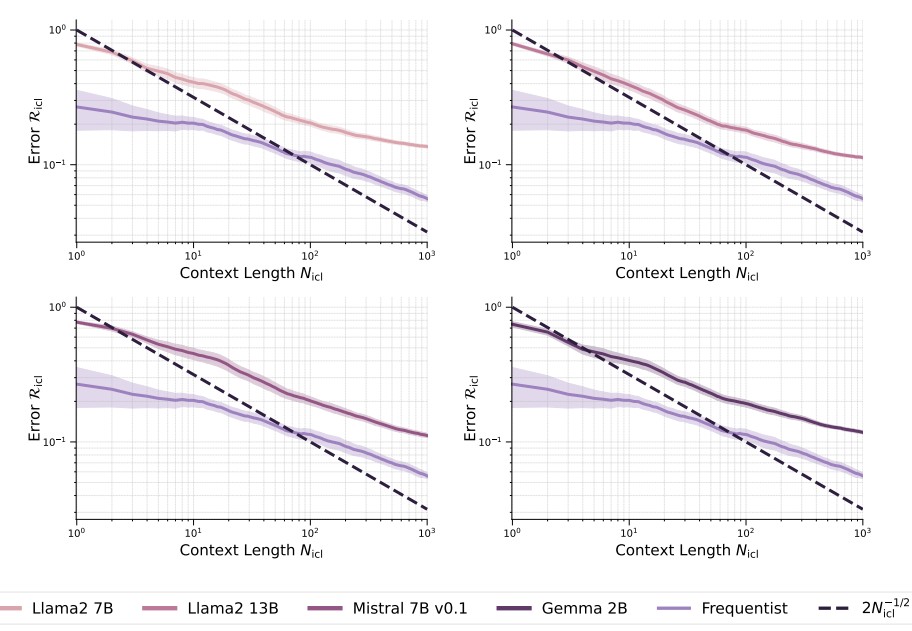

Figure 15: We plot the risk $\mathcal{R}_{\text{icl}}$ as functions of $N_{\text{icl}}$, with $95\%$ confidence intervals. **Upper Left.** `Llama2 7B` **Upper Right.** `Llama2 13B` **Lower Left.** `Mistral 7Bv0.1` **Lower Right.** `Gemma 2B`

### E.4 RECENT MODELS: IMPACT OF THE TOKENIZATION

As explained in Appendix E.1, models like Llama 3 tokenize 3-digit numbers with a single token. This saves a lot of inference compute time, but not necessarily in terms of performance when considering Markov chains with a few number of states $d$, since we have to separate the states by a comma to force tokenization into a single digit (e.g. the transitions $1 \rightarrow 0 \rightarrow 1$ will be prompted as `1,0,1` (5 tokens) instead of `101` (1 token). In Fig. 16, we reproduce the same experiment as in Fig. 5(left), but with Llama 3 models. The scaling laws are quite good, but much less so than those obtained with `Gemma 2B` and `Mistral 7Bv0.1` on the same inputs. On the other hand, with these models, it can be extremely interesting to consider Markov chains with many states, as we did in Fig. 6(right). In the next section, we will use `LLama3` to learn other dynamic systems presented in Liu et al. (2024).

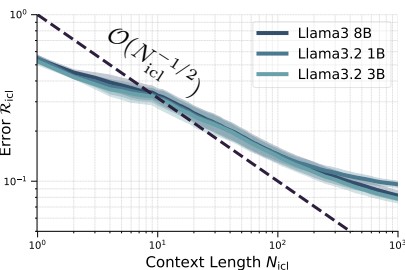

Figure 16: **In-context scaling laws for `LLama3` herd of models.** We plot the risk $\mathcal{R}_{\mathrm{icl}}$ as functions of $N_{\mathrm{icl}}$, with 95% confidence intervals.

### E.5 DYNAMICAL SYSTEMS

We consider four of the dynamic systems highlighted in (Liu et al., 2024) : a geometric Brownian motion, a correlated Gaussian, an uncorrelated Gaussian and an uncorrelated uniform processes. We display in Fig. 17 the risks of `LLama3 8B` and the frequentist method, which once again highlight the emerging capacity of in-context learning.

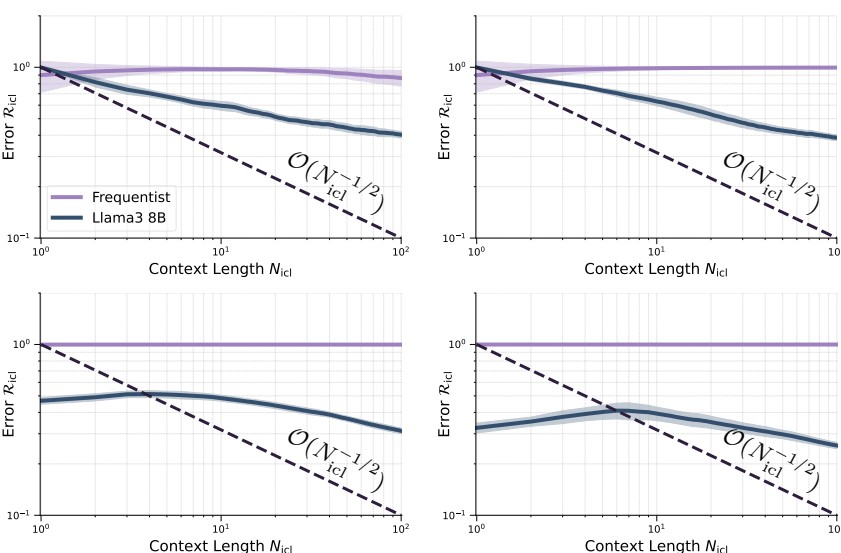

Figure 17: **`LLama3 8B` on dynamical systems.** We plot the risks $\mathcal{R}_{\mathrm{icl}}$ as functions of $N_{\mathrm{icl}}$ for `LLama3 8B` and the frequentist approach (Wolfer & Kontorovich, 2019) with 95% confidence intervals. **Upper Left.** Geometric Brownian motion. **Upper Right.** Correlated Gaussian. **Lower Left.** Uncorrelated Gaussian. **Lower Right.** Uncorrelated Uniform.

## F   EXTENDED RESULTS WITH THE KL DIVERGENCE

As explained in Remark 4.1, the total variation is the natural choice to define the risks in Eq. (2). Another possibility in the Markov chain literature is to use the KL divergence to compare probability distributions (Hao et al., 2018). This is an interesting candidate as the KL divergence is naturally connected to the cross-entropy loss commonly used to train neural networks (the cross-entropy corresponds to the KL divergence between the true distribution and the predicted softmax distribution (Blondel et al., 2019). In this section, we discuss the extension of the theoretical results of Section 4 by replacing the TV distance with the KL divergence in the risks' definition, i.e.,

$$\mathcal{R}(\boldsymbol{\Theta}) := \mathbb{E}_{\mathbf{S}\sim\mathbb{P}_{\mathcal{L}}}[d_{\mathrm{KL}}(\mathbf{Q}^*(\mathbf{S},\cdot)||\mathbf{Q}_f(\mathbf{S},\cdot))], \quad \widehat{\mathcal{R}}(\boldsymbol{\Theta}) := \frac{1}{N}\sum_{n=1}^{N} d_{\mathrm{KL}}(\mathbb{P}_{\mathcal{L}}(\cdot\mid\mathbf{S}_n)||\mathbb{P}_{\boldsymbol{\Theta}}(\cdot\mid\mathbf{S}_n)). \quad (37)$$

### F.1   PRE-TRAINING GENERALIZATION BOUNDS

Theorem 4.1, Corollary 4.2 and Corollary 4.3 related to the pre-training phase in Section 4.1 can be obtained similarly if the risks are defined with the KL divergence following Eq. (37). Indeed, the key step to derive the proofs is to obtain a similar result to Lemma D.6 but with the KL divergence. The next lemma provides this result.

**Lemma F.1.** *Consider two probability distributions* $\mathbb{P}, \mathbb{Q}$ *defined on a measure space* $(\Omega, \mathcal{F})$ *and a $\sigma$-finite measure $\nu$ on* $(\Omega, \mathcal{F})$. *Let $p, q$ be the corresponding probabilities densities, i.e., we have* $\mathbb{P}(d\omega) = q(\omega)\nu(d\omega)$ *and* $\mathbb{Q}(d\omega) = p(\omega)\nu(d\omega)$. *If there exists a non-negative constant $B$ such that for any $z \in \Omega$,* $\left|\log\sqrt{\frac{\mathbb{P}(z)}{\mathbb{Q}(z)}}\right| \leq B$, *then we have*

$$d_{\mathrm{KL}}(\mathbb{P}||\mathbb{Q}) \leq B.$$

*Proof.* We have

$$\begin{aligned}
0 \leq d_{\mathrm{KL}}(\mathbb{P}||\mathbb{Q}) &= |d_{\mathrm{KL}}(\mathbb{P}||\mathbb{Q})| \\
&= \left|\int \mathbb{P}(z)\log(\frac{\mathbb{P}(z)}{\mathbb{Q}(z)})dz\right| \\
&\leq \int |\mathbb{P}(z)||\log(\frac{\mathbb{P}(z)}{\mathbb{Q}(z)})|dz \\
&\leq B\int |\mathbb{P}(z)|dz \\
&= B\int \mathbb{P}(z)dz \\
&= B,
\end{aligned}$$

which concludes the proof. □

We can now state the results similar to Theorem 4.1, Corollary 4.2 and Corollary 4.3 from the pre-training phase when the risk is defined according to Eq. (37).

**Theorem F.2** (Pre-training generalization bound). *Consider an LLM* $f_{\boldsymbol{\Theta}} \in \mathcal{F}$. *We denote by $\boldsymbol{\Gamma}$ the mixing matrix of the pre-training sequences of tokens* $(\mathbf{S}_1, \ldots, \mathbf{S}_{N_{\mathrm{train}}})$. *Let* $0 < \delta < 1$, *then with probability at least* $1 - \delta$,

$$\mathcal{R}_{\mathrm{pre}}(\boldsymbol{\Theta}) \leq \widehat{\mathcal{R}}_{\mathrm{pre}}(\boldsymbol{\Theta}) + \frac{\bar{B}}{\sqrt{N_{\mathrm{train}}}}\sqrt{\log\left(\frac{2}{\delta}\right)},$$

*where $\bar{B} = \sqrt{2}\|\mathbf{\Gamma}\|\max\{\log{(T)} + 2B_U/\tau, \log{(1/c_0)}\}$ is a constant depending on the parameters of the problem.*

*Proof.* The proof simply follows from the proof of Theorem 4.1 by replacing the upper bound $\sqrt{2B}$ by $B$ (with the appropriate upper-bound $B$) when Lemma D.6 is used in the proof. $\square$

**Corollary F.3** (Depth-dependent bound). *Consider an LLM $f_\mathbf{\Theta} \in \tilde{\mathcal{F}} := \{f_\mathbf{\Theta} \mid \mathbf{\Theta} \in \tilde{\mathcal{W}}\}$. With the same assumptions as in Theorem 4.1, we have*

$$\mathcal{R}_{\mathrm{pre}}(\mathbf{\Theta}) \leq \widehat{\mathcal{R}}_{\mathrm{pre}}(\mathbf{\Theta}) + \frac{\bar{B}}{\sqrt{N_{\mathrm{train}}}}\sqrt{\log\left(\frac{2}{\delta}\right)},$$

*where $\bar{B} = \sqrt{2}\|\mathbf{\Gamma}\|\max\{\log{(T)} + 2(B_\mathbf{\Theta})^L/\tau, \log{(1/c_0)}\}$ is a constant depending on the parameters of the problem, and $B_\mathbf{\Theta} = [(1 + rmB_1B_2)(1 + \frac{r^3}{H}B_OB_V)](B_{\mathrm{tok}}B_U)^{1/L}$.*

*Proof.* The proof simply follows from the proof of Theorem 4.1 by replacing the upper bound $\sqrt{2B}$ by $B$ (with the appropriate upper-bound $B$) when Lemma D.6 is used in the proof. $\square$

**Corollary F.4** (Sample complexity). *Let $\bar{B}$ be the parameter-dependent constant of Theorem F.2 or Corollary F.3. Let $\delta \in [0, 1]$ and let $\epsilon > 0$. If $N_{\mathrm{train}} \geq N^* := \lceil\frac{4\bar{B}^2}{\epsilon^2}\log\left(\frac{2}{\delta}\right)\rceil$ and if we assume a perfect pre-training error for $f_\mathbf{\Theta}$, then we have with probability at least $1 - \delta$,*

$$\mathbb{E}_{\mathbf{S}\sim\mathbb{P}_\mathcal{L}}\|\mathbf{Q}^*(\mathbf{S}, \cdot) - \mathbf{Q}_f(\mathbf{S}, \cdot)\|_1 \leq \epsilon.$$

*Proof.* The proof simply follows from the proof of Theorem 4.1 by replacing the upper bound $\sqrt{2B}$ by $B$ (with the appropriate upper-bound $B$) when Lemma D.6 is used in the proof. $\square$

## F.2 LIMITATIONS

We recall from Remark 4.1 that the TV distance is a natural choice to compare transition matrices in the Markov chain literature. In addition, while the KL divergence can be used to compare probability distributions, it does not define a metric space. Hence, we cannot straightforwardly extend Theorem 4.4 with the KL divergence because the proof relies on the use of the triangular inequality. As Theorem 4.4 is one of our main results and enables us to show that the theory and the practice align (Section 5), this also contributed to our preference for the TV distance instead of the KL divergence.

