# OpenReview forum: "Large Language Models as Markov Chains"
_ICLR.cc/2025/Conference — Submitted to ICLR 2025_

### Official Review · Reviewer_Gi2N · 2024-11-03

**Soundness:** 3
**Presentation:** 3
**Contribution:** 3
**Rating:** 6
**Confidence:** 2

**Summary:**

This paper theoretically characterizes the inference mechanism of large language models (LLMs) as a finite-state Markov chain, providing insights into LLM properties based on the transition kernel of the associated Markov chain. Specifically, by modeling an LLM as a Markov chain, the authors show that the output distribution of the LLM converges to a specific stationary distribution and that the convergence rate depends on certain parameters in the LLM. They also derive generalization bounds for the pre-training and in-context learning of LLMs and experimentally verify the behavior of some recent LLMs using these theoretical results.

**Strengths:**

* This paper introduces a unique approach by modeling LLMs as finite-state Markov chains. This approach provides a clear explanation of the inference process and convergence properties of LLMs to a stationary distribution and also provides a new perspective on understanding LLM behavior theoretically. This approach is expected to expand our understanding of the overall model structure of existing LLM schemes, such as autoregressive models with self-attention mechanisms.
* A major contribution is the derivation of generalization bounds for pre-training and in-context learning, applying the concentration inequality from Markov chain theory to gain a quantitative understanding of LLM's learning characteristics. This increases confidence in the accuracy and inference capabilities of LLMs and is also helpful for comparing different LLMs.

**Weaknesses:**

* As noted above, while modeling LLMs as a Markov chain provides a new perspective, there is no new development in Markov chain theory itself. The author's theoretical framework is limited to an extension of existing Markov chain theory, and it remains unclear to what extent this framework fully captures the complex structure and dynamic properties unique to LLMs. Due to the multilayered structure and self-attention mechanisms within LLMs, it is unclear to what extent these can be accurately presented using Markov chains.
* The effect of temperature parameters on convergence rate is analyzed, but the influence of other hyperparameters and model structure, such as depth, number of heads, and layer-specific settings, on convergence and inference performance, is not examined in detail. Consequently, the paper provides insufficient guidance for optimal parameter design in LLMs.

**Questions:**

* I believe that Fig.3(d) on line 243 might be a typo for Fig.3(c). Setting that aside, what is the meaning of Fig.3(c)? If the parameter $K$ is only assigned to Llama and GPT, there seems to be no additional information added to Proposition 3.3. Could the authors explain how to evaluate the specific value of $\epsilon$ in LLMs?

* Fig.4(d) presents the temperature dependence of $\epsilon$, but how was this obtained? I may have missed something, but I would appreciate clarification from the authors. It is obvious that convergence is faster at high temperatures, so I am curious about the functional form of $\epsilon(T)$.

* The probability distribution formula on line 327 seems redundant.

* In evaluating the generalization bound, it is assumed that the pre-training data is generated from the Marton coupling. Is this condition essential for obtaining the bound?

* In Fig.5(right), the context length $N_\text{icl}$ dependence of the risk is presented for several values of $t_\text{min}$. While the results for small $N_\text{icl}$ deviate from the scaling law, the theoretical prediction is $\sqrt{t_\text{min}/N_\text{icl}}$. Did the authors attempt to plot the results as a function of $t_\text{min}/N_\text{icl}$ or $N_\text{icl}/t_\text{min}$ to check for a universal curve, indicating a crossover?

---

> ### Author Response · Authors · 2024-11-15
> **Answer to Reviewer Gi2N (Part 1)**
>
> We thank the reviewer Gi2N for their thoughtful feedback. We are happy that the reviewer acknowledged our theoretical results and found our approach unique and the paper well-written.
>
> We address the reviewer's comments point by point below.
>
> > 1. As noted above, while modeling LLMs as a Markov chain provides a new perspective, there is no new development in Markov chain theory itself. The author's theoretical framework is limited to an extension of existing Markov chain theory, and it remains unclear to what extent this framework fully captures the complex structure and dynamic properties unique to LLMs. Due to the multilayered structure and self-attention mechanisms within LLMs, it is unclear to what extent these can be accurately presented using Markov chains.
>
> We believe there might have been a misunderstanding and would like to clarify that our theoretical analysis focuses on transformer-based architectures: the **fixed context window $K$ only characterizes transformer-based LLM architecture**. Moreover, the class $\mathcal{F}$, defined right before Section 4.1, is a class of transformer models. In conclusion, our theoretical contribution explicitly relies on the transformer architecture and we believe our **results can be highly relevant to the LLM community** (being also validated by experiments on commonly used LLMs).
>
> Kindly note that **the framework from Section 3 does fully capture the inference mechanism of an LLM**, especially its dynamic generation. One simple way to understand why it can be captured explicitly is to recall that the finiteness of the vocabulary and context window allows us to enumerate all possible transitions that an LLM can make. The probabilities of such transitions are estimated by an LLM of choice then define the transition kernel. We would also like to stress out that **Section 4 does consider a multi-layered transformer with self-attention mechanism**. Hence the proofs extensively rely on the characteristics of the LLM's Transformer architecture and are not a simple extension of Markov chain theory (cf Appendix and proofs p.22-37).
>
> > 2. The effect of temperature parameters on convergence rate is analyzed, but the influence of other hyperparameters and model structure, such as depth, number of heads, and layer-specific settings, on convergence and inference performance, is not examined in detail. Consequently, the paper provides insufficient guidance for optimal parameter design in LLMs.
>
> Kindly note that Section 3 is concerned only with understanding and characterizing the inference mechanism of the LLMs, while other dependencies are analyzed later in Section 4. Our Theorem 4.1 and Corollary 4.2 show how the pretraining success depends on the number of heads, hidden dimensionality of attention, and MLPs, as well as the boundness of certain weight matrices, and therefore may be of interest for the practical architecture design of LLMs. The experimental results of Section 5 on state-of-the-art also show the high relevance of our theoretical analysis to the LLM community.
>
> > 3. I believe that Fig.3(d) on line 243 might be a typo for Fig.3(c). Setting that aside, what is the meaning of Fig.3(c)? If the parameter $K$ is only assigned to Llama and GPT, there seems to be no additional information added to Proposition 3.3. Could the authors explain how to evaluate the specific value of $\varepsilon$ in LLMs?
>
> We thank the reviewer for noting this typo and corrected in the revised version of the paper.
>
> This plot shows how the convergence to the stationary distribution depends on the context window size $K$. Intuitively, and as established formally by our theory, for larger $K$, the model has more possible transitions to cover before reaching the stationary distribution. This plot illustrates this behavior for different LLMs based on the value of $K$ and empirically validates our theory (Proposition 3.3). Does this explanation clarify the meaning of Figure 3. c)?
>
> The $\varepsilon$ was fixed to $1e-6$ value for all models as it is impossible to know the lowest probability that a given LLM may assign in principle for a pair of sequences.
>
> Could the reviewer specify what they mean by "$K$ is only assigned to Llama and GPT"? It is also a parameter characterizing our toy model where it is equal to $K=3$ (toy model defined in Section 3.2).
>
> > 4. Fig.4(d) presents the temperature dependence of $\varepsilon$, but how was this obtained? I may have missed something, but I would appreciate clarification from the authors. It is obvious that convergence is faster at high temperatures, so I am curious about the functional form of $\varepsilon(T)$.
>
> This was obtained from a toy model (defined in Section 3.2) by calculating the min value of the transition kernel (i.e., $\varepsilon$) for different temperatures $\tau$.

---

> > ### Author Response · Authors · 2024-11-15
> > **Answer to Reviewer Gi2N (Final Part)**
> >
> > > 5. The probability distribution formula on line 327 seems redundant.
> >
> > We thank the reviewer for noting this typo, which we corrected in the revised version of the paper.
> >
> > > 6. In evaluating the generalization bound, it is assumed that the pre-training data is generated from the Marton coupling. Is this condition essential for obtaining the bound?
> >
> > We thank the reviewer for the question. We assume a Marton coupling to remain the closest possible to the practice as this is closely related to how the pretraining is done in practice where some sequences can be easier to predict than others. It should be noted that this is a **weaker assumption** than what is typically done in the literature [1, 2, 3, 4], and, in particular, it encompasses the case of IID data (the mixing matrix $\Gamma$ becomes the identity).
> >
> > [1] Wolfer and Kontorovich. Minimax learning of ergodic Markov chains. Algorithmic Learning Theory 2019.
> >
> > [2] Zhang et al. What and how does in-context learning learn? Bayesian model averaging, parameterization, and generalization. arXiv 2023.
> >
> > [3] Hu et al. Unveiling the Statistical Foundations of Chain-of-Thought Prompting Methods. arXiv 2024.
> >
> > [4] Pierre Marion. Generalization bounds for neural ordinary differential equations and deep residual networks. NeurIPS 2023.
> >
> > > 7. In Fig.5(right), the context length $N_{icl}$  dependence of the risk is presented for several values of $t_{min}$. While the results for small $N_{icl}$ deviate from the scaling law, the theoretical prediction is $\sqrt{\frac{t_{min}}{N_{icl}}}$. Did the authors attempt to plot the results as a function of $\frac{t_{min}}{N_{icl}}$ or $\frac{N_{icl}}{t_{min}}$ to check for a universal curve, indicating a crossover?
> >
> > We thank the reviewer for the great suggestion! We added the suggested plot (risk as a function $\smash{N_{\mathrm{icl}}/t_{\mathrm{min}}}$) in the right panel of Figure 5 of the revised paper. As predicted by the reviewer, we indeed observe that all of the risks follow a universal curve that agrees remarkably well with our theoretical prediction of $\mathcal{O}(\sqrt{\frac{t_{min}}{N_{icl}}})$ (Theorem 4.4).
> >
> > We hope that all the reviewer's concerns and questions have been addressed and we remain open to future discussions. Given our explanations and additional experiments, we would be grateful if the reviewer could reconsider the evaluation of our work accordingly.

---

> > > ### Author Response · Authors · 2024-11-22
> > > **Looking forward to the reviewer's reply**
> > >
> > > We thank the reviewer Gi2N for their insightful feedback, which has contributed to the enhancement of our paper. We hope that our comments addressed their concerns and clarified our contributions.
> > >
> > > As the author-reviewer discussion period is ending, we would happily read their reply to answer any additional questions. However, we may not have the opportunity to respond to their comments after that.
> > >
> > > We thank them again for their time and valuable review. We look forward to hearing from them.

---

> > > > ### Comment · Reviewer_Gi2N · 2024-11-29
> > > >
> > > > Thank you very much for your detailed response. Many of my questions have been clarified.
> > > >
> > > > *Regarding Fig. 3(c), I would like to ask about the specific language model mentioned in the legend. I assume this plot only represents the context window parameter $K$, and does not refer to any specific language model. Proposition 3.3 shows exponential convergence and that the characteristic scale is proportional to $K$. Initially, I thought this figure might have some significance for actual language models. However, it seems there is no deeper meaning beyond illustrating the relationship with $K$.
> > > >
> > > > *Thank you for re-plotting Fig. 5(c) with a different scale. This version makes it easier to observe the behavior of $(t_{min}/N_{icl})^{1/2}$. Upon closely examining the figure, I expect that the error ${\cal R}(N_{icl},t_{min})$ can be scaled using a scaling function $f(x)$ as ${\cal R}(N_{icl},t_{min})=t_{min}^{-1/2}f(N_{icl}/t_{min})$. At least, it appears to scale roughly within the plotted range. In other words, if the vertical axis is multiplied by $t_{min}^{1/2}$, the data should collapse onto an approximately universal function.
> > > >
> > > > In any case, my misunderstandings have largely been resolved, and I am starting to appreciate the strengths of this paper. I will raise my score.

---

> ### Author Response · Authors · 2024-11-29
> **Thank You!**
>
> We thank the reviewer for their time, their nice words about our work, and for raising their score. This kind of feedback feels encouraging!
>
> **Significance of Figure 3(c)**
>
> We are happy to have clarified the reviewer's concern. Indeed, in this particular case, we only use the context length of the respective actual models to obtain the plot. However, this experiment can be closely related to several safety issues commonly observed with LLMs. For instance, repetitions and loops in LLM generations connect nicely to the notion of the stationary distribution as looping over the repeating sequences in an almost deterministic way is close to how we interpret the convergence to the stationary distribution in our work. Similarly, recent work showed that almost all LLMs learn a feature that captures the relative frequencies of tokens seen during pretraining [1]. Our work (in particular Proposition 3.3) can hint at the fact that this feature should become increasingly used when the generation goes on for a certain number of steps. In parallel, glitch tokens [2] with low relative frequency may force the LLM to converge to the stationary distribution faster as they increase the value of $\varepsilon$. Other fallbacks of LLM, such as repetitions, are also of great interest [3] and could be analyzed using the Markov chain formalization we propose. We hope that these ideas will be explored by the subsequent works if our paper, acting as a first step in this direction, gets a chance to be published.
>
> **Updated Figure 5**
>
> We are happy to read that the reviewer appreciates our updated figure. We agree with them that it better shows the rate $\sqrt{\frac{t_{min}}{N_{icl}}}$ from Theorem 4.4. Hence, the reviewer's suggestion of multiplying the vertical axis is (again) excellent as, according to Theorem 4.4, data should collapse onto an approximately universal function.
>
>
> We thank again the reviewer for their time and precious feedback to improve the paper. If the reviewer thinks our paper sheds new light on important phenomena observed in recent LLMs, we would greatly appreciate their stronger endorsement of our work.
>
> ---
> [1] Sean Ogier. LLMs Universally Learn a Feature Representing Token Frequency / Rarity. 2024.
>
> [2] Martin Fell. A Search for More ChatGPT / GPT-3.5 / GPT-4 "Unspeakable" Glitch Tokens. 2023.
>
> [3]  Ivgi et al. From Loops to Oops: Fallback Behaviors of Language Models
> Under Uncertainty. arXiv 2024

---

### Official Review · Reviewer_ugPy · 2024-11-04

**Soundness:** 3
**Presentation:** 2
**Contribution:** 2
**Rating:** 6
**Confidence:** 3

**Summary:**

In this paper, the authors first enstablish the equivalence between autoregressive models and first-order Markov chains with number of states proportional to $T^K$, where $T$ is the alphabet size and $K$ is the context window, proving results about the stationary distribution of such chains. In the second part of the paper, the authors prove pre-training and in-context generalization bounds for LLMs, validating them with several experiments.

**Strengths:**

The paper is well written. I was not able to check all the proofs but the math seems sound to me and the results are interesting.

**Weaknesses:**

My main concern is that it is not clear to me what is the connection between the first part (equivalence between autoregressive models and Markov chains) and the second part (generalization bounds) of the paper. As it is, the paper looks like two different papers glued together.

**Questions:**

I have a few questions that would make it easier for me to understand the results of the paper and its impact:
1) What is the connection between the first and second part of the paper? Are the results of the first part important for the derivation of the generalization bounds? If so, what are the key steps when the equivalence proved in the first part is useful in the second part? What is the intuition?
2) I struggle to understand the implication of the results of the first part for LLMs. In particular, what does it mean for LLMs to converge to their stationary distribution? Does that mean that the model's loss has converged to some minima? Does that mean that the LLM inference performance cannot be improved further?
3) I was curious about the choice of the TV distance in the risk definition (2). Would it be easy to move to some more popular loss choice for LLMs such as the cross-entropy loss? Intuitively, it shouldn't be too hard since cross-entropy loss is basically KL divergence, which in turn is connected to TV distance through Pinsker's inequality.

---

> ### Author Response · Authors · 2024-11-15
> **Answer to Reviewer ugPy**
>
> We thank the reviewer ugPy for their valuable comments. We are happy to hear that the reviewer found the paper well-written and our theoretical results solid and interesting.
>
> We address the reviewer's concerns point by point below.
>
> > 1. My main concern is that it is not clear to me what is the connection between the first part (equivalence between autoregressive models and Markov chains) and the second part (generalization bounds) of the paper. As it is, the paper looks like two different papers glued together.
>
> We apologize for the confusion and are happy to provide clarification concerning the two parts of the paper. As noted by the reviewer, the first part establishes an equivalence between LLMs and Markov chains acting on sequences (the mentions of autoregressive models were removed following the Reviewer o34n suggestions), while the generalization bounds rely on the Markov chain viewpoint. Following the reviewer's comment, we revised the paper in Section 4 (l311 & l314), to highlight the dependence of the risk definition on the Markov chain formalization. Finally, our experiments conducted in Section 5, show that our theoretical results, based on the Markov chain framework, are validated on state-of-the-art LLMs.
>
> To further clarify the connections between Sections 3 and 4, we propose to the Reviewer to move the presentation of the sample complexity results given by Corollary 4.3 at the beginning of Section 4. We would then unroll the results that led to it. Would such a transition make the link between the two sections clearer?
>
> > 2. I struggle to understand the implication of the results of the first part for LLMs. In particular, what does it mean for LLMs to converge to their stationary distribution? Does that mean that the model's loss has converged to some minima? Does that mean that the LLM inference performance cannot be improved further?
>
> On a high level, it means that calling an LLM on any given prompt for a sufficiently (and very large) number of times will lead to a sequence that reflects the frequencies of transitions between different sequences that the LLMs were trained on, regardless of the input prompt. For instance, if in English the token corresponding to the letter $J$ (assuming per letter tokenization for simplicity) has a very low probability to be followed with an $N$ then, and despite the actual context of the appearance of the latter $J$, the LLM will attribute a low probability to $N$ as it has seen few such transitions in the training corpora. One can see this as a universal vocabulary of the rareness of transitions that the LLM's generation converges to, not immediately related to the training process.
>
> > 3. I was curious about the choice of the TV distance in the risk definition (2). Would it be easy to move to some more popular loss choice for LLMs such as the cross-entropy loss? Intuitively, it shouldn't be too hard since cross-entropy loss is basically KL divergence, which in turn is connected to TV distance through Pinsker's inequality.
>
> We thank the reviewer for suggesting to use the cross-entropy (or equivalently the KL divergence). This intuition of the reviewer is correct as Theorem 4.1, Corollary 4.2, and Corollary 4.3 can be derived too by using the KL divergence instead of the TV distance in the definition of the risk (Eq. 2). Formally, the important sub-result to obtain is akin to Lemma D.6 but with the KL divergence. Assuming that $|\log(\frac{P(z)}{Q(z)})| \leq B$, Lemma D.6 ensures that $d_{TV}(P, Q) \leq \sqrt{2B}$. Similarly, we can show that under the same assumption, we have $d_{KL}(P, Q) \leq B$ and Theorem 4.1, Corollary 4.2 and Corollary 4.3 will follow accordingly by replacing $\sqrt{2B}$ by $B$. Indeed, we have:
> $$0 \leq d_{KL}(P, Q) = |d_{KL}(P, Q)| = |\int P(z) \log(\frac{P(z)}{Q(z)})dz| \leq \int |P(z)| |\log(\frac{P(z)}{Q(z)})|dz.$$
> As $P$ is a probability distribution, we have $\int |P(z)| dz = \int P(z) dz = 1$ and using $|\log(\frac{P(z)}{Q(z)})| \leq B$, we obtain $d_{KL}(P, Q) \leq B$, which concludes the proof. However, as the KL divergence is **not a distance**, Theorem 4.4 cannot be obtained as it relies on the triangular inequality. As Theorem 4.4 is one of our main results and enables us to show that our theory **aligns** with the practice on commonly used LLMs, and that the TV distance is commonly used to compare probability distributions, this motivated us to prefer the TV distance to the KL divergence (or cross-entropy). It should be noted that in our case, Pinsker's inequality is not in the good sense so we cannot use it in our proof. We added these results in the revised paper (Appendix F) and thank again the reviewer for the suggestion.
>
> We hope that all the reviewer's concerns and questions have been addressed. We will be happy to answer any additional questions. Given our explanations and revision to the paper, we would be grateful if the reviewer could reconsider the evaluation of our work accordingly.

---

> > ### Author Response · Authors · 2024-11-22
> > **Looking forward for the reviewer's reply**
> >
> > We thank the reviewer ugPy for their insightful feedback, which has contributed to the enhancement of our paper. We hope that our comments addressed their concerns and clarified our contributions.
> >
> > As the author-reviewer discussion period is ending, we would happily read their reply to answer any additional questions. However, we may not have the opportunity to respond to their comments after that.
> >
> > We thank them again for their time and valuable review. We look forward to hearing from them.

---

> ### Comment · Reviewer_ugPy · 2024-11-23
>
> I thank the authors for their thorough answers. I increased my score to 6.

---

> > ### Author Response · Authors · 2024-11-24
> > **Thank You!**
> >
> > We thank the reviewer for their updated score and valuable review. We are glad that our explanations helped clarify our contributions and their impact. If the reviewer has any remaining concerns, we would happily address them before the end of the discussion period.

---

### Official Review · Reviewer_o34N · 2024-11-05

**Soundness:** 3
**Presentation:** 2
**Contribution:** 2
**Rating:** 3
**Confidence:** 4

**Summary:**

This work studies the learning theoretical properties of auto-regressive model ( which the author claim to be LLM ) on Markov Chains.
In particular, they prove the convergence of MC and the generalization guarantees of learning the parameter $\Theta$. They also carried out numerical experiments on it.

**Strengths:**

The strength is the theoretical results look pretty solid. And the author writes this paper in a very clear fashion. making it very readable. Also the results follow from the uniform concentration bound of MCs, presented in a rather classical learning theoretic (a.k.a. VC style) fashion.  I believe this result is relevant to the learning theory community although I cannot decide how much technical innovations are made here. The reviewer believes that this result adapt previous results by many other learning theory researchers who develop theory on estimating the transition kernel of MC given single path observations.

**Weaknesses:**

The major weakness is the connection to LLMs. The first thing is whether the class F is sufficiently large to include some instances of LLMs. Right now it seems only takes a single input $v_{I}$ instead of $v_{I-1},v_{I-2},\ldots,v_{I-k}$. Though it may not be feasible anymore to learn MC due to the Markovian properties after we make this change.  However, real world language might not be treated as Markovian, which is a very stylized assumption here. The reviewer believes this will affect the impact of this work in the LLM community.

Another huge concern is that $F$ is regardless of the structure of the model itself. It can either be an RNN, Transformer, or a LSTM, and the result continues to hold. It can even be non-NN model. This does not actually reveal why LLM is a powerful machine since it highly depends on the NN as its backbone. Therefore these limitations significantly affect its contribution to the modern LLM theory community.

Hence, the reviewer suggests that instead of connecting such work to LLM, the author might be able to use the Auto-regressive model as the title, which is deemed more proper. And this model might include LLM as a special instance. At the current stage, the reviewer does not think LLM community will put a lot of focus on this work due to its over-general problem setup.

**Questions:**

Overall this work is very well written, which is the reason why the reviewer gives a 5 rather than a 3. The reviewer believes that this work is not very relevant to the LLM community but more relevant to the traditional learning theory (a.k.a. VC theory) community (given that the author changes the way this work is presented by replacing most of the LLMs to autoregressive models and maybe draw connections in a milder way).

---

> ### Author Response · Authors · 2024-11-15
> **Answer to Reviewer o34N (Part 1)**
>
> We thank the reviewer o34N for their detailed feedback. We appreciate that the reviewer found our work theoretically solid and well-written.
>
> We address the reviewer's concerns point by point below.
>
> > 1. The reviewer believes that this result adapt previous results by many other learning theory researchers who develop theory on estimating the transition kernel of MC given single path observations.
>
> We believe that there might have been a misunderstanding regarding the scope of our work. While it is true that our theoretical results leverage the classical learning theoretic framework, our work provides a modelization and guarantees on the inference mechanism of LLMs (defined as in the general reply above). These LLMs' generalization bounds are novel in the literature to the best of our knowledge. We revised the paper to include a definition of what we call an LLM in Section 1 (see revised paper l112-116).
>
> As for the results regarding the estimation of the transition kernel of MC given single path observations, we **only use them as a baseline** in the analysis of Corollary 4.3 to compare the LLM's performance to a frequentist approach but the proofs of Section 3 and 4 **do not rely on it**.
>
> While leveraging prior work, our proof techniques go well beyond simple and straightforward adaptions (cf. the appendix and proofs p.21-36).
>
> > 2. The major weakness is the connection to LLMs. The first thing is whether the class F is sufficiently large to include some instances of LLMs.
>
> We employ the term LLM to define a deep transformer-based model trained on non-iid data whose inference is based on the next-token prediction principle. The latter implies that such a model transitions as a Markov chain from a **sequence of tokens** to a **sequence of tokens** (with appending and deletion of tokens due to the limited context length). Such a definition includes many, if not all, popular LLMs. We hope that this clarification, along with the empirical results validating our theory on existing modern LLMs, makes the scope of our work clearer to the reviewer.
>
> > 3. Right now it seems only takes a single input $v_I$ instead of $v_{I-1}, \ldots, v_{I-k}$. Though it may not be feasible anymore to learn MC due to the Markovian properties after we make this change. However, real world language might not be treated as Markovian, which is a very stylized assumption here. The reviewer believes this will affect the impact of this work in the LLM community.
>
> We believe that there might have been a misunderstanding with our theoretical contributions. Here, the input $v_I$ designs a **sequence of token** (e.g., "I am a "), while the output $v_{I+1}$ is the concatenation of $v_I$ and a token $v$ predicted by the LLM (e.g., "I am a cat"). We note that this **does not rely** on any Markovian assumption of the language and corresponds to the practical setup used in the LLM community. Following the reviewer's comment, we revised the paper to clarify this point in the introduction after the summary of contributions (see revised paper l112-117).
>
> > 4. Another huge concern is that $F$ is regardless of the structure of the model itself. It can either be an RNN, Transformer, or a LSTM, and the result continues to hold. It can even be non-NN model. This does not actually reveal why LLM is a powerful machine since it highly depends on the NN as its backbone. Therefore these limitations significantly affect its contribution to the modern LLM theory community.
>
> We believe that there might have been a misunderstanding as we only analyze models with fixed context windows. Although the fixed vocabulary size is common to all these models for language modeling, the **fixed context window $K$ only characterizes transformer-based LLM architecture**. Hence, **other models** such as RNNs, LSTMs, or even State-Spaces models (SSMs) like Mamba [1,2] **do not fit into our theoretical framework** since the context window is only _implicit_ (as it depends on the model's ability to “remember” information about several successive states). Moreover, the class $\mathcal{F}$, defined right before Section 4.1 (l320-331 and Appendix B), is a class of transformer models (therefore a neural network). Finally, our theoretical contribution explicitly relies on the transformer architecture and we believe this makes our results **highly relevant to the LLM community** (while being also validated by experiments on commonly used LLMs like **Mistral 7B, Llama 2, Llama 3.2, Gemma 2B**).
>
> [1] Gu and Dao. Mamba: Linear-Time Sequence Modeling with Selective State Spaces. Arxiv 2023.
>
> [2] Dao and Gu. Transformers are SSMs: Generalized Models and Efficient Algorithms Through Structured State Space Duality. ICML 2024.

---

> > ### Author Response · Authors · 2024-11-15
> > **Answer to Reviewer o34N (Final Part)**
> >
> > > 5. Hence, the reviewer suggests that instead of connecting such work to LLM, the author might be able to use the Auto-regressive model as the title, which is deemed more proper. And this model might include LLM as a special instance. At the current stage, the reviewer does not think LLM community will put a lot of focus on this work due to its over-general problem setup.
> >
> > We thank the reviewer for their suggestion, which allowed us to clarify our contribution. Following the answer to the previous comment, we revised the paper to remove the mention of auto-regressive models. Once again, we would like to emphasize that our theoretical model actually characterizes LLMs and not any sequence-to-sequence algorithm.
> >
> > We also revised the paper to include a definition of LLMs in the introduction (l112-117) as a deep transformer-based model trained on non-IID data with its inference mechanism being based on the next token prediction principle (with addition and deletion). In addition, we would like to clarify that in Section 3 we study only the inference mechanism of these models regardless of their specific architecture, while in Section 4, we aim to understand the impact of the architecture and the nature of the training data. In particular, all the results of Section 4 extensively rely on the **characteristics of transformer-based LLMs**, and experiments of Section 5 are conducted on the most commonly used and recent LLMs (**Mistral 7B, Gemma 2B, Llama 3.2**, etc.). Hence, we believe our work can be valuable to the LLM community both from a theoretical and practical perspective.
> >
> > > 6. Overall this work is very well written, which is the reason why the reviewer gives a 5 rather than a 3. The reviewer believes that this work is not very relevant to the LLM community but more relevant to the traditional learning theory (a.k.a. VC theory) community (given that the author changes the way this work is presented by replacing most of the LLMs to autoregressive models and maybe draw connections in a milder way).
> >
> > We thank the reviewer for their question. We believe we already answered it in the replies above.
> >
> > We thank the reviewer for acknowledging the quality of our writing and presentation. We hope that the focus of our work on LLMs, our theoretical results tailored to transformer-based LLMs, and our experiments conducted on commonly used LLMs addressed the reviewer's concerns regarding the relevance of our work to the current LLM community. Do these explanations make the relevance of our work clearer for the reviewer?
> >
> > We hope that all the reviewer's concerns and questions have been addressed and we remain open to future discussions. Given our explanations and revisions to the paper, we would be grateful if the reviewer could reconsider the evaluation of our work accordingly.

---

> ### Comment · Reviewer_o34N · 2024-11-15
> **Response to the author.**
>
> We thank the authors for their clarification on the notations and improved presentations. However, the reviewer is still not convinced that such function class is specific to Transformers. In particular, this does not fully leverage the attention+MLP structure of the Transformers. Given the definition in line 324, the reviewer believes that ML researchers **can actually construct so many models that are not Transformers but still satisfy the conditions given by the LLMs in this work**. However these models might be very bad in practice and LLMs won't succeed on them while the theoretical results remain untouched (In particular the reviewer thinks that LSTMs and RNNs does not necessarily take the input as a whole and can perform exactly the same as Transformer here). I also checked the proof and realized that the Transformer architecture is not necessary in the proof either.
>
> Hence, using LLM in the title without analyzing the Transformer specific properties might not be very helpful as contributions to the LLM community. The reviewer believes that this is a significant reason that maybe LLM should be removed from the title here. However, as a pure learning theory paper this work seems to also lack theoretical novelties (like solving some novel mathematical open problems and unusual mathematical techniques). The proof idea seems to be quite standard in the literature or the author does not highlight it in the contribution.
>
> Given that the other reviewers remain mildly negative on the acceptance of this work, the reviewer changes the score from 5 to 3 to clarify their stance. However, the reviewer believes that this work might be interesting ( suppose enough technical innovations are made ) to the learning theory community with the removal of LLMs but some class of autoregressive models as the subject of study. Though, in this case, significant contributions in the technical ideas might need to be highlighted.

---

> > ### Author Response · Authors · 2024-11-26
> > **Extension of the discussion period**
> >
> > Given the extension of the discussion period, we would like to engage in a discussion with the reviewer to understand better how they see us framing this work. Currently, we tend to think that the term “LLM” for a reviewer denotes a user-friendly version of a pre-trained (and fine-tuned through RLHF) language model optimized for conversational tasks. If this is the case, we agree with the reviewer that this model is hard, or almost impossible, to study theoretically. For instance, our theory predicts that an LLM will converge to a stationary distribution likely leading to repeating patterns, yet in deployment, such an LLM will use guardrails ensuring that repetitiveness will not happen, for instance, if Topk (Fan, Lewis, and Dauphin (2018)), Nucleus sampling (Holtzman et al. (2020)) or repetition penalty is used. However, without works like ours, one cannot understand why such repetitions occur in the first place, although numerous works are dedicated to this (Google Scholar search output for “repetition” “llm” given over 4k results, ¾ of them are from 2024).
> >
> > As for the VC theory remark, we feel that the reviewer may employ this term as a metaphor for VC theory being historically perceived as uninformative in providing insightful results for deep neural networks. For this part, we replied earlier by explicitly defining a class of deep transformer-based models that we study and, most importantly, empirical validation of our theory for such models. We hope that the reviewer will change their mind in this regard.
> >
> > Would such a clarification included in the revised version allow us to find common ground with the reviewer in addition to previous replies?

---

> ### Author Response · Authors · 2024-11-16
> **Second answer to Reviewer o34N (part 1)**
>
> We thank the reviewer again for their detailed feedback and the time taken to review our work. However, we believe some of the reviewer comments are based on misunderstandings or inaccuracies. Below, we provide a point-by-point response to their concerns, citing specific parts of the paper to address them comprehensively. After addressing these points in detail, we provide a general comment regarding the change in score.
>
> ---
>
> ### **1. Specificity of the function class to Transformers**
>
> > "The reviewer is still not convinced that such function class is specific to Transformers. In particular, this does not fully leverage the attention+MLP structure of the Transformers."
>
> This statement does not align with the definitions and analysis presented in our paper. Specifically:
>
> - **Definition of the class  $\mathcal{F}$:**
>   As defined in Section 4.1 (lines 324–333) and elaborated in Appendix B (pages 16-18), the function class $\mathcal{F}$ is **explicitly defined as a class of transformer models**. In particular, this section describes that the architecture considered is composed of :
>
> _ Token embeddings.
>
> _ Positional embeddings.
>
> _ Multi-head attention (MHA), followed by a layer normalization.
>
> _ Feed-forward block (with MLP), followed by a layer normalization.
>
> _ Softmax output layer.
>
> - **Link to what we have in practice:**
>
>
> We further add that if the reviewer remains unconvinced about the strong connection between our theoretical framework and what an LLM might look like in practice, they could refer to the implementation of an LLM class in the Python ```transformers``` package to verify that our modeling aligns closely with the practical architectures used in state-of-the-art systems. For example, the following three lines of code illustrate the architecture of a typical transformer-based LLM (gpt2):
>
> ```python
> from transformers import AutoModelForCausalLM
>
> model = AutoModelForCausalLM.from_pretrained("gpt2")
> print(model)
> ```
>
> This outputs:
>
> ```python
> GPT2LMHeadModel(
>   (transformer): GPT2Model(
>     (wte): Embedding(50257, 768)
>     (wpe): Embedding(1024, 768)
>     (drop): Dropout(p=0.1, inplace=False)
>     (h): ModuleList(
>       (0-11): 12 x GPT2Block(
>         (ln_1): LayerNorm((768,), eps=1e-05, elementwise_affine=True)
>         (attn): GPT2SdpaAttention(
>           (c_attn): Conv1D()
>           (c_proj): Conv1D()
>           (attn_dropout): Dropout(p=0.1, inplace=False)
>           (resid_dropout): Dropout(p=0.1, inplace=False)
>         )
>         (ln_2): LayerNorm((768,), eps=1e-05, elementwise_affine=True)
>         (mlp): GPT2MLP(
>           (c_fc): Conv1D()
>           (c_proj): Conv1D()
>           (act): NewGELUActivation()
>           (dropout): Dropout(p=0.1, inplace=False)
>         )
>       )
>     )
>     (ln_f): LayerNorm((768,), eps=1e-05, elementwise_affine=True)
>   )
>   (lm_head): Linear(in_features=768, out_features=50257, bias=False)
> )
> ```
>
> The architecture includes the following components, which reflect the descriptions in Section 4.1 (lines 320–331) and Appendix B of our paper:
>
> _ Token embeddings: Implemented within **(wte)**.
>
> _ Positional embeddings: Implemented within **(wpe)**.
>
> _ Multi-head attention (MHA), followed by a layer normalization: Implemented within the Attention block **(attn)**, and layer normlization **(ln_2)**.
>
> _ Feed-forward block (with MLP), followed by a layer normalization: Implemented within the MLP module **(mlp)**, and layer normlization **(ln_f)**.
>
> _ Softmax output layer: While not explicitly present in the model architecture, the softmax operation is applied to the logits generated by **(lm_head)** to obtain probabilities during inference.
>
> This implementation demonstrates that our modelization aligns closely with the practical architectures used in so-called LLMs.

---

> > ### Author Response · Authors · 2024-11-16
> > **Second answer to Reviewer o34N (part 2)**
> >
> > ### **2. Claim about constructing exotic mdoels**
> >
> > > "the reviewer believes that ML researchers can actually construct so many models that are not Transformers but still satisfy the conditions given by the LLMs in this work."
> >
> > This statement misunderstands the constraints imposed by our theoretical framework.
> >
> > In addition to the constraints discussed in the previous points, the elements of the class $\mathcal{F}$ are fixed-vocabulary models of size $T$ with a fixed context window of size  $K$. This further narrows the class to practical characteristics even more specific to transformer-based LLMs, as the **fixed context window** is a stamp of such models. We do not see how ML researchers could propose an exotic model that satisfies all the conditions imposed by class $\mathcal{F}$ (those outlined in the previous point, plus the fixed context window $K$ and fixed vocabulary size $T$) without effectively being a transformer-based LLM.
> >
> > Even if we hypothetically removed the explicit requirement for the class $\mathcal{F}$ to include transformers (as described in Section 4.1 and Appendix B) and kept only the fixed context window property, models like RNNs and LSTMs would still not fall into this class. As we explained in a previous response that the reviewer seems to have overlooked, RNNs and LSTMs do not implement the fixed context window mechanism explicitly and are thus excluded from this framework. The reviewer’s claim about the applicability to arbitrary models does not hold under the constraints we have established.
> >
> > ---
> >
> > ### **3. Relevance of the Transformer architecture in the proofs**
> >
> > > "The reviewer thinks that LSTMs and RNNs do not necessarily take the input as a whole and can perform exactly the same as Transformers here. I also checked the proof and realized that the Transformer architecture is not necessary in the proof either."
> >
> >
> > We encourage the reviewer to carefully check **all the proofs related to Section 4** (pages 25–36 in the Appendix). These 11 pages of detailed mathematical derivations link our theoretical results directly to the transformer architecture. This extensive body of work demonstrates that our analysis is rigorously grounded in the specific properties of transformers. **Specifically, we encourage the reviewer to carefully review the following key results**: Lemma D.7, Lemma D.8, Proposition D.9, Lemma D.12, Lemma D.15, Corollary D.16, Lemma D.19, Lemma D.20, and Proposition D.21. These results form the backbone of our theoretical analysis, directly supporting the claims made in Section 4. Furthermore, these results are deeply rooted in the specific properties of transformer-based LLM architectures, including (but not limited to) multi-head attention mechanisms and feed-forward MLP layers, which **excludes LSTMs and RNNs**.
> >
> > We also note that comments are directly included in the paper following the derivation of our theoretical results. For instance, after Corollary 4.2, we explicitly state: *"We note that $\overline{B}$ exhibits an exponential dependence on the depth of the transformer, which also amplifies the hidden dimensionality (width) of the embedding layer $r$. This contrasts with the dependency in $m$, the hidden dimensionality of the MLP block, which is linear."* The relevance of the transformer architecture, as well as the dependency on its characteristics, is therefore clearly discussed after each result.
> >
> >
> > ---
> >
> > ### **4. Use of "LLM" in the title**
> >
> > > "Using LLM in the title without analyzing the Transformer specific properties might not be very helpful as contributions to the LLM community."
> >
> > We strongly believe that "LLM" is the correct term to use in the title for the following reasons:
> >
> > - **Focus on LLMs:**
> > Our work explicitly defines LLMs as deep Transformer-based models trained on non-IID data with inference based on next-token prediction (lines 112–117). This definition aligns with the state-of-the-art in natural language processing and distinguishes our analysis from general autoregressive models.
> >
> > - **Theoretical setup close to the practice:**
> > As highlighted in our clarification number $1$, a few simple lines of code using the ```transformers``` package are sufficient to demonstrate that our theoretical framework closely aligns with practical implementations.
> >
> >
> > - **Experimental validation:**
> > Our experiments were conducted on widely used LLMs such as Mistral 7B, Gemma 2B, Llama 3.2 (Section 5). These models are representative of the modern LLM landscape, reinforcing the relevance of our work to the LLM community.
> >
> > Changing the title to "autoregressive models" would misrepresent the scope of our contributions, which are directly targeted at LLMs.

---

> ### Author Response · Authors · 2024-11-16
> **Second answer to Reviewer o34N (final part)**
>
> ### **5. Theoretical novelty**
>
> > "The proof idea seems to be quite standard in the literature or the author does not highlight it in the contribution."
>
> While we utilize classical tools, such as uniform concentration bounds, our work combines these with novel applications tailored to LLMs.
>
> Note that we do not claim to reinvent the foundations of machine learning theory. It is entirely standard to use concentration inequalities to derive generalization bounds. The challenge lies in **incorporating the specific characteristics of the model** under consideration into the proofs.
>
> **We believe we have successfully addressed this challenge** in our paper and have clarified our theoretical contributions —particularly that these bounds are explicitly derived for transformer-based LLMs— in the previous points of this answer.
>
> ---
>
> ### **General Comment on the Score Change**
>
> We note that you have changed your score from 5 to 3. While we respect your decision, we wish to highlight that:
>
> 1. **Inconsistency with previous comments:**
> You initially acknowledged that our work is well-written and rigorous. Given the additional clarifications and revisions we have provided, a decrease in score seems difficult to reconcile with your earlier evaluation.
>
> 2. **Failure to address our clarifications:**
> We have carefully addressed all your feedback, and responded point by point to all your criticisms, both in our initial responses and in the revised manuscript. However, your latest comment ignores many of these clarifications.
>
> 3. **Experimental contributions ignored:**
> The reviewer mentions that our work is purely theoretical, yet disregards our experimental section, which is based on commonly used LLMs. In addition, it appears that the reviewer undervalues theoretical work within the LLM community, while we believe that such work holds significant value. Moreover, our paper combines both theoretical and experimental contributions, making this critique seem unjustified.
>
> 4. **Relevance to the LLM community:**
> We do believe that simply stating "I don’t think it is relevant to the LLM community" without engaging with or acknowledging the major contributions we have outlined is not a sufficient basis to lower the score to 3. Our contributions are theoretical and empirical, tailored specifically to transformer-based LLMs, and are clearly detailed throughout the paper. In addition, note that getting theoretical bounds justifying in-context learning capabilities and scaling laws (observed in papers like [1]) was an open problem in the literature.
>
> 5. **Impact on the review process:**
> Lowering the score after significant revisions and clarifications signals that such efforts are not valued, which may discourage constructive dialogue in future review processes.
>
> We hope this response clarifies any remaining misunderstandings and provides sufficient justification for our work’s contributions to the LLM community. We thank you again for your feedback and your time.
>
> [1] Large Language Models Are Zero-Shot Time Series Forecasters. Gruver et al. NeurIPS 2023

---

### Author Response · Authors · 2024-11-15
**General Comment**

We thank all the reviewers for thoroughly and carefully reading our paper. We are happy to hear that they acknowledged the **novelty and quality** of our study (Reviewers o34N, ugPy, Gi2N), its **solid and interesting theoretical foundations** (Reviewers o34N, ugPy, Gi2N), and the **clarity of the writing** (Reviewers o34N, ugPy, Gi2N).

### **General comment**
Before giving individual replies to all reviewers, we would like to provide three important clarifications to clear out any potential misunderstanding.

- **1)** In our work, we employ the term LLM to define a deep transformer-based model trained on non-iid data whose inference is based on the next-token prediction principle. The latter implies that such a model transitions from **a sequence of tokens** to **a sequence of tokens**. Hence, in Section 3, the Markov chain formalization transitions between states that are **sequences of tokens** and **not single tokens**. This defines a Markov chain on a state space of size $|\mathcal{V}_K^*| = T(T^{K} - 1)/(T-1)$. The vast majority of existing LLMs fall into our definition suggesting that our results apply to them. We also note that this **does not rely** on any Markovian assumption on the language.

We propose to add this definition in Section 1 and avoid using the term auto-regressive models which raised some confusion.

-  **2)** We would like to highlight that our **experimental results**, not acknowledged in the reviews, **validate our theoretical findings** using Mistral 7B, Gemma 2B, Llama 3.2, 2 versions of Llama 2, which are all very recent models and commonly used in the LLM community. We believe that this shows the high relevance of this work for the broad LLM community. We would be more than happy to answer any questions on the experimental section.

-  **3)** All our theoretical results are obtained by considering **a generic transformer-based LLM** (multi-layer, multi-head Transformers typically used in GPTs, Llamas, etc. [1]) **without any simplifying assumption on the architecture** which is rarely done in the literature (e.g., most theoretical work focus on linear attention or simplified one-layer transformers [2, 3 ,4]). In this setup, we consider a challenging setting with **non-IID data** which was also scarcely explored. Although our work does not provide a definite and complete answer, we believe it makes a significant contribution to this field of study. We made these choices to remain as close as possible to the practice of the LLM community (see point 2). This comes with many technical challenges and to derive our proofs, we **extensively rely on the characteristics of the transformer architecture**. We believe that in addition to our experiments on commonly used LLMs, our results and the techniques used can be valuable both to the LLM and the Machine Learning communities.

We hope that the reviewers will take these clarifications into account in their evaluation of our work. We remain open to continuing this constructive discussion for the length of the rebuttal period.


We strongly believe that the paper benefited from the reviews, and we respectfully invite the reviewers to reconsider their evaluation score if they think we addressed their concerns.

### **Updates on the revised paper**

We list below the changes made to the paper (in blue) as suggested by the reviewers:

- Additional definition of LLM following point 1 (l 112-117).
- Explicit formulation of the size of the state space $\mathcal{V}_K^*$ of the Markov chain in Proposition 3.1 (l 177).
- Risk definition, with a more direct link to the Markov chain equivalence. (l311 & l314).
- Updated Figure 5 (p.10) with the evolution of the risk as a function of $\frac{N_{icl}}{t_{min}}$
- Additional results with the KL divergence in the risk instead of the TV distance (Appendix F).
- Replaced the term “auto-regressive models”, which has given rise to some confusion.

[1] Brown et al. Language models are few-shot learners. NeurIPS 2020

[2] Furuya et al. Transformers are Universal In-context Learners. arXiv 2024

[3] Zhang et al. Trained Transformers Learn Linear Models In-Context. JMLR 2024

[4] Oswald et al. Transformers Learn In-Context by Gradient Descent. ICML 2023

[5] Kim et al. Transformers are Minimax Optimal Nonparametric In-Context Learners. ICML 2024 Workshop

---

### Author Response · Authors · 2024-12-03
**Wrap-up of the rebuttal**

As the rebuttal period approaches its end, we would like to thank the reviewers for their contribution to improving our manuscript. We are pleased with the fact that **two reviewers (ugPy and Gi2N) chose to raise their scores**. To summarize, we would like to recapitulate the following
1. No reviewer questioned the novelty of our contributions, which, together with the importance of the studied topic, experiments on recent up-to-date LLMs and lack of technical flaws leaves **no particular unresolved issues**. As it appears now, the scores for our paper are rather borderline but we hope that the reviewers will take into account what is written above to endorse our paper stronger.
2. We would like to reemphasize that our answers to reviewer o34N provide a factual justification for our claims. We hope that the reviewer will acknowledge this post-rebuttal as they haven’t engaged in any further discussion so far.
3. The reviewer o34N’s motivation to decrease their score seems out-of-place as we do not see any negative reaction from reviewers’ ugPy and Gi2N justifying lowering their score. In fact, both of them raised their scores as mentioned above, and highlighted the fact our thorough answers resolved their concerns.

---

### Meta-Review · Area_Chair_hz2G · 2024-12-20

**Metareview:**

The paper brings a learning theory viewpoint to the standard equivalence between language models (LLMs or others, of window size K) and Markov chains (of state size |V|^K). It later connects the existence of a stationary distribution of Markov chains to the the inference power of LLMs.

Main concerns after author feedback remained around whether the theoretical framework truly captures transformer-specific properties of LLMs. The work suffers from over-claiming in this aspect, even affecting the title of the paper. The main equivalence in Proposition 3.1 (and results of Section 3) have nothing to do with transformers but holds for any neural network which (in an autoregressive fashion) predicts the next token based on previous K tokens.

As suggested by Reviewer ugPy, it would be better to decouple this into two papers, separating the (in my view not novel results of Section 3) and the more interesting results of Section 4, or at least deemphasize Section 3 as being only a warm-up for the main results in Section 4.
The author feedback phase led to constructive discussion and improvements in clarity, but nevertheless couldn't fully resolve the main concerns.

**Additional Comments On Reviewer Discussion:**

The author feedback phase was productive as acknowledged by the reviewers. Some of the concerns however remained if the work is ready for the high bar of ICLR.

---

### Decision · Program_Chairs · 2025-01-22

Reject